# CMT: Mid-Training for Efficient Learning of Consistency, Mean Flow, and Flow Map Models

**Zheyuan Hu**[1,*]    **Chieh-Hsin Lai**[1,*]    **Yuki Mitsufuji**[1,2]    **Stefano Ermon**[3]

[1]Sony AI    [2]Sony Group Corporation    [3]Stanford University
*Equal Contribution
zyhu2001@gmail.com   chieh-hsin.lai@sony.com

## Abstract

Flow map models such as Consistency Models (CM) and Mean Flow (MF) enable few-step generation by learning the long jump of the ODE solution of diffusion models, yet training remains unstable, sensitive to hyperparameters, and costly. Initializing from a pre-trained diffusion model helps, but still requires converting infinitesimal steps into a long-jump map, leaving instability unresolved. We introduce *mid-training*, the first concept and practical method that inserts a lightweight intermediate stage between the (diffusion) pre-training and the final flow map training (i.e., post-training) for vision generation. Concretely, *Consistency Mid-Training* (CMT) is a compact and principled stage that trains a model to map points along a solver trajectory from a pre-trained model, starting from a prior sample, directly to the solver-generated clean sample. It yields a trajectory-consistent and stable initialization. This initializer outperforms random and diffusion-based baselines and enables fast, robust convergence without heuristics. Initializing post-training with CMT weights further simplifies flow map learning. Empirically, CMT achieves state-of-the-art two-step FIDs of 1.97 (CIFAR-10), 1.32 (ImageNet $64 \times 64$), and 1.84 (ImageNet $512 \times 512$), using up to 98% less training data and GPU time than CMs. On ImageNet $256 \times 256$, it attains 1-step FID 3.34 with $\sim$50% less training than MF from scratch (FID 3.43). On MSCOCO T2I, CMT reaches the best FID with $\sim$47% less training. This establishes CMT as a principled, efficient, and general framework for training flow map models. Code and models are available at https://github.com/sony/cmt.

## 1 Introduction

Diffusion models (Ho et al., 2020; Song & Ermon, 2019) have become a cornerstone of modern generative modeling, yet their practical application is often hindered by a significant computational burden during inference. This latency arises because sampling is equivalent to solving a probability flow ordinary differential equation (PF-ODE) (Song et al., 2021), a process that requires many iterative steps. To circumvent this limitation, a promising direction focuses on directly learning the solution (integration) map of the PF-ODE, which is also referred to as a *flow map model*.

Because the PF–ODE flow map lacks a closed form, recent methods learn surrogate maps by enforcing proper-

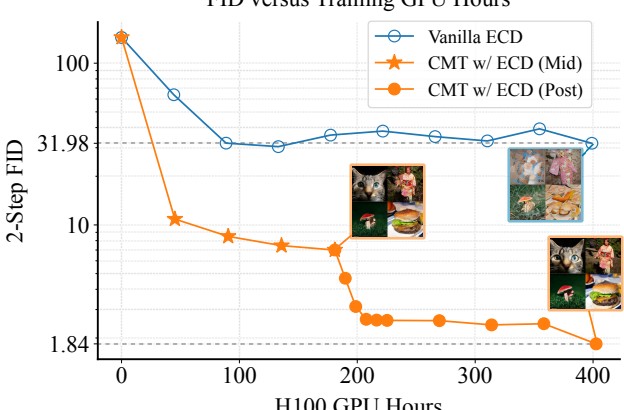

**FID versus Training GPU Hours**

Figure 1: **FID vs. training time for vanilla ECD (Geng et al., 2025b) and CMT (ours) on ImageNet** $512 \times 512$**.** With the proposed mid-training, our CMT w/ ECD (as post-trained flow map) achieves SOTA two step FID of 1.84 using only 400 H100 GPU hours (mid- and post-training combined). Under the same budget, vanilla ECD still produces unrecognizable images, and even to reach a reasonable two step FID of 3.38 it requires 4643.99 hours. Overall, CMT reduces the total training cost of flow map models by **91.4%** while achieving SOTA performance.

ties that any exact flow must satisfy: e.g., Consistency Models (CM) (Song et al., 2023) impose cross–noise-level self-consistency and Mean Flow (MF) (Geng et al., 2025a) matches time averages along trajectories. However, these objectives (Song et al., 2023; Song & Dhariwal, 2024; Kim et al., 2024; Geng et al., 2025a; Lu & Song, 2025; Sabour et al., 2025) supervise against *stop-gradient, network-dependent pseudo-targets* that drift with training dynamics. The lack of a true, time-invariant regression target injects bias, yields unstable optimization signals, and slows convergence. While recent works observed that initializing from pre-trained diffusion weights can mitigate instability (Geng et al., 2025b; Lu & Song, 2025), this does not address the root cause. Fundamentally, a flow map must learn large integrated jumps of the trajectory, whereas diffusion models capture only the *infinitesimal* movements. This mismatch renders diffusion-based initialization fragile: flow map training then depends on brittle heuristics (e.g., time weightings and sampling schedules) yet still suffers from flow map learning's instability and converges slowly (Geng et al., 2025b). In particular, recent studies (Zhu, 2025) have observed that post-training MF, even when initialized from a well-trained large-scale diffusion model (Ma et al., 2024), is prone to divergence and requires careful configuration tuning.

We address the instability and high cost of training few-step flow maps by introducing *mid-training* for vision generation, conceptually inspired by mid-training in large language models (Groeneveld et al., 2024). In our setting, mid-training is a brief intermediate stage that bridges pre-training (e.g., diffusion model) and flow map post-training. We instantiate this idea as *Consistency Mid-Training* (CMT), a lightweight procedure that leverages trajectories generated by a pre-trained model to produce a trajectory-aware initialization. Concretely, CMT trains a model to map any point along a trajectory determined by a pre-trained model, from a prior sample directly to the clean endpoint of exactly that same trajectory in a single step. Mid-training with CMT requires no architectural changes, converges quickly, adds only modest cost, and avoids fragile heuristics such as stop gradients, time sampling, and weighting schedules. This trajectory-aligned initializer provides a better starting point for flow map post-training than either random or diffusion-based weight transfer, while also simplifying engineering practices. Most importantly, it significantly reduces the total training cost (in both time and required training data) and improves training stability.

Theoretically, we show that CMT reduce the gradient discrepancy between the oracle and practical flow map losses, providing a stronger and trajectory-aligned initializer for the flow map post-training. Empirically, we evaluate on pixel-space benchmarks (CIFAR-10, FFHQ $64 \times 64$, AFHQv2 $64 \times 64$, ImageNet $64 \times 64$) and latent-space high-resolution models (ImageNet $256 \times 256/512 \times 512$) as well as MSCOCO T2I (Lin et al., 2014). Initializing flow map models with CMT improves post-training stability, speeds up convergence, and boosts final quality. CMT achieves new SOTA two-step FIDs of 1.97 (CIFAR-10), 1.32 (ImageNet $64 \times 64$), 1.84 (ImageNet $512 \times 512$), 2.34 (AFHQv2 $64 \times 64$), and 2.75 (FFHQ $64 \times 64$), while reducing training budget (images processed, equivalently backprop steps) and GPU time by up to 98% versus baselines without mid-training. On ImageNet $512 \times 512$, CMT reaches FID 1.84 with 91.4% less training than the baseline (FID 3.38; Figure 1); on ImageNet $256 \times 256$, it attains FID 3.34 with ~50% less training than MF from scratch (FID 3.43). On MSCOCO T2I, CMT achieves the best FID with ~47% less training.

CMT applies to both CM and MF, demonstrating broad applicability across ODE-based flow map generators. To our knowledge, this work presents the first systematic investigation of mid-training for few-step flow map models in vision generation, establishing CMT as an effective approach that significantly reduces training cost while achieving state-of-the-art quality.

## 2 PRELIMINARIES AND RELATED WORK

### 2.1 DIFFUSION MODELS AND FLOW MATCHING

Diffusion models define a forward process that perturbs clean data $\mathbf{x}_0 \sim p_{\text{data}}$ into $\mathbf{x}_t = \alpha_t \mathbf{x}_0 + \sigma_t \boldsymbol{\epsilon}$, where $\boldsymbol{\epsilon} \sim \mathcal{N}(\mathbf{0}, \mathbf{I})$ and $t \in [0, T]$. Equivalently, $\mathbf{x}_t \sim p_t(\mathbf{x}_t | \mathbf{x}_0) = \mathcal{N}(\cdot; \alpha_t \mathbf{x}_0, \sigma_t^2 \mathbf{I})$, which induces marginals $p_t(\mathbf{x}_t) = \int p_t(\mathbf{x}_t | \mathbf{x}_0) p_{\text{data}}(\mathbf{x}_0) d\mathbf{x}_0$. Two closely related training approaches are standard.

**EDM** (Karras et al., 2022) trains a denoiser $\mathbf{D}_{\boldsymbol{\theta}}(\mathbf{x}_t, t)$ with a preconditioned parametrization by minimizing $\mathcal{L}_{\text{DM}}(\boldsymbol{\theta}) = \mathbb{E}_t \mathbb{E}_{\mathbf{x}_0, \boldsymbol{\epsilon}}[\, w(t) \|\mathbf{D}_{\boldsymbol{\theta}}(\mathbf{x}_t, t) - \mathbf{x}_0\|_2^2]$. At optimum, $\mathbf{D}(\mathbf{x}_t, t) = \mathbb{E}[\mathbf{x}_0 | \mathbf{x}_t]$. EDM uses $\alpha_t = 1$, $\sigma_t = t$ for $t \in [0, T]$, so for large $T$, the prior $p_{\text{prior}}$ approaches $\mathcal{N}(\mathbf{0}, T^2 \mathbf{I})$.

**Flow Matching** (Lipman et al., 2023) fits a vector field $\mathbf{v}_{\boldsymbol{\theta}}(\mathbf{x}_t, t)$ to the conditional velocity of the perturbation: $\mathcal{L}_{\mathrm{FM}}(\boldsymbol{\theta}) = \mathbb{E}_t \, \mathbb{E}_{\mathbf{x}_0, \boldsymbol{\epsilon}} \left[ w(t) \left\| \mathbf{v}_{\boldsymbol{\theta}}(\mathbf{x}_t, t) - \left( \alpha_t' \mathbf{x}_0 + \sigma_t' \boldsymbol{\epsilon} \right) \right\|_2^2 \right]$. At optimum, $\mathbf{v}(\mathbf{x}_t, t) = \mathbb{E}[\alpha_t' \mathbf{x}_0 + \sigma_t' \boldsymbol{\epsilon} | \mathbf{x}_t]$. A common choice $\alpha_t = 1 - t$, $\sigma_t = t$ for $t \in [0, 1]$ yields a unit-Gaussian prior.

**Their Relationship.** The parametrizations are equivalent; the marginal optimal velocity and denoiser satisfy $\mathbf{v}(\mathbf{x}_t, t) = \left( \alpha_t' - \alpha_t \frac{\sigma_t'}{\sigma_t} \right) \mathbf{D}(\mathbf{x}_t, t) + \frac{\sigma_t'}{\sigma_t} \mathbf{x}_t$, so one can translate between $\mathbf{v}_{\boldsymbol{\theta}}$ and $\mathbf{D}_{\boldsymbol{\theta}}$ given the scheduler. Sampling integrates the PF-ODE, $\frac{\mathrm{d}\mathbf{x}_t}{\mathrm{d}t} = \mathbf{v}(\mathbf{x}_t, t)$, starting from $\mathbf{x}_T \sim p_{\mathrm{prior}}$ (Gaussian in both views) down to $t = 0$. Either $\mathbf{v}_{\boldsymbol{\theta}} \approx \mathbf{v}$ or $\mathbf{D}_{\boldsymbol{\theta}} \approx \mathbf{D}$ can be used to realize the drift.

## 2.2 FEW-STEP FLOW MAP GENERATIVE MODELING

In this section, we propose a unified view that connects existing formulations of flow map models. Numerical integration of the PF-ODE can be slow, as it requires simulating a system across many small time steps. Few-step models offer a more efficient alternative by directly learning the solution to the PF-ODE's integral: the *flow map*, $\boldsymbol{\Psi}_{t \to s}(\cdot)$. This map takes an initial state $\mathbf{x}_t$ at time $t$ and jumps directly to its final destination at time $s$:

$$\boldsymbol{\Psi}_{t \to s}(\mathbf{x}_t) := \mathbf{x}_t + \int_t^s \mathbf{v}(\mathbf{x}_u, u) \mathrm{d}u, \quad \mathbf{v}(\mathbf{x}_u, u) = \left( \alpha_u' - \alpha_u \frac{\sigma_u'}{\sigma_u} \right) \mathbf{D}(\mathbf{x}_u, u) + \frac{\sigma_u'}{\sigma_u} \mathbf{x}_u. \quad (1)$$

**Special Flow Map: Consistency Models (CM).** The CM family adapts EDM's framework and learn a few-step denoiser $\mathbf{f}_{\boldsymbol{\theta}}(\cdot, t)$ that approximates the flow map to the origin, $\boldsymbol{\Psi}_{t \to 0}(\cdot)$, for any $t \in (0, T]$. Training relies on the *consistency property*: any two points along the same PF-ODE trajectory should map the same origin. We propose a *principled re-interpretation* of the CM family objective (Song et al., 2023; Song & Dhariwal, 2024; Geng et al., 2025b; Lu & Song, 2025):

$$\mathcal{L}_{\mathrm{oracle\text{-}CM}}(\boldsymbol{\theta}) := \mathbb{E}_t \mathbb{E}_{\mathbf{x}_t \sim p_t} \left[ w(t) d(\mathbf{f}_{\boldsymbol{\theta}}(\mathbf{x}_t, t), \boldsymbol{\Psi}_{t \to 0}(\mathbf{x}_t)) \right], \quad (2)$$

with $d$ a point-wise distance (e.g., squared $\ell_2$ or perceptual (Zhang et al., 2018)). At optimum, $\mathbf{f}(\mathbf{x}_t, t) = \boldsymbol{\Psi}_{t \to 0}(\mathbf{x}_t)$ (Proposition F.1). Since $\boldsymbol{\Psi}_{t \to 0}$ is unavailable, CM uses a *stop-gradient* surrogate from the previous step: $\boldsymbol{\Psi}_{t \to 0}(\mathbf{x}_t) \approx \mathbf{f}_{\boldsymbol{\theta}^-}(\mathbf{x}_{t-\Delta t}, t - \Delta t)$ with $\Delta t > 0$, where $\mathbf{x}_{t-\Delta t}$ comes from (i) *Consistency Distillation (CD)*: a one-step solver with a pre-trained diffusion teacher, which calls the teacher during training; or (ii) *Consistency Training (CT)*: the analytic estimate $\mathbf{x}_{t-\Delta t} = \alpha_{t-\Delta t} \mathbf{x}_0 + \sigma_{t-\Delta t} \boldsymbol{\epsilon}$ using the same $(\mathbf{x}_0, \boldsymbol{\epsilon})$ as in $\mathbf{x}_t = \alpha_t \mathbf{x}_0 + \sigma_t \boldsymbol{\epsilon}$, requiring no teacher calls. Both approaches improve performance by initializing from pre-trained diffusion weights (Lu & Song, 2025; Geng et al., 2025b). In the CT setting, the CM surrogate loss is

$$\mathcal{L}_{\mathrm{CM}}(\boldsymbol{\theta}) := \mathbb{E}_{t, \mathbf{x}_t} \left[ w(t) d \left( \mathbf{f}_{\boldsymbol{\theta}}(\mathbf{x}_t, t), \mathbf{f}_{\boldsymbol{\theta}^-}(\mathbf{x}_{t-\Delta t}, t - \Delta t) \right) \right]. \quad (3)$$

Recent variants (e.g., ECT (Geng et al., 2025b)) refine initialization, time steps, $w(t)$, and $d(\cdot, \cdot)$.

**General Flow Map.** Consistency Trajectory Model (CTM) (Kim et al., 2024) was the first to learn the general flow map $\boldsymbol{\Psi}_{t \to s}$ for arbitrary $t > s$ via $\mathbf{G}_{\boldsymbol{\theta}}(\mathbf{x}_t, t, s)$, minimizing

$$\mathcal{L}_{\mathrm{oracle\text{-}CTM}}(\boldsymbol{\theta}) := \mathbb{E}_{t > s} \mathbb{E}_{\mathbf{x}_t \sim p_t} \left[ w(t) d(\mathbf{G}_{\boldsymbol{\theta}}(\mathbf{x}_t, t, s), \boldsymbol{\Psi}_{t \to s}(\mathbf{x}_t)) \right]. \quad (4)$$

As $\boldsymbol{\Psi}_{t \to s}$ is inaccessible, CTM uses a stop-gradient target evaluated at $\mathbf{G}_{\boldsymbol{\theta}}$ itself, similar to CM.

More recently, MF (Geng et al., 2025a) builds on the flow matching formulation by modeling the average drift $\mathbf{h}_{\boldsymbol{\theta}}(\mathbf{x}_t, t, s) \approx \mathbf{h}(\mathbf{x}_t, t, s) := \frac{1}{t-s} \int_s^t \mathbf{v}(\mathbf{x}_u, u) \, \mathrm{d}u$ over an interval $[s, t]$, also following the principled Equation (4). MF constructs a surrogate target by differentiating $(t - s)\mathbf{h}(\mathbf{x}_t, t, s) = \int_s^t \mathbf{v}(\mathbf{x}_u, u) \, \mathrm{d}u$ w.r.t. $t$, yielding the MF training loss:

$$\mathcal{L}_{\mathrm{MF}}(\boldsymbol{\theta}) := \mathbb{E}_{t > s} \mathbb{E}_{\mathbf{x}_t \sim p_t} \left[ w(t) \| \mathbf{h}_{\boldsymbol{\theta}}(\mathbf{x}_t, t, s) - \mathbf{h}_{\boldsymbol{\theta}^-}^{\mathrm{tgt}}(\mathbf{x}_t, t, s) \|_2^2 \right], \quad (5)$$

where the regression target is applied with stop-gradient as $\mathbf{h}_{\boldsymbol{\theta}^-}^{\mathrm{tgt}}(\mathbf{x}_t, t, s) := \mathbf{v}(\mathbf{x}_t, t) - (t - s)\left( \mathbf{v}(\mathbf{x}_t, t)\partial_{\mathbf{x}}\mathbf{h}_{\boldsymbol{\theta}^-} + \partial_t \mathbf{h}_{\boldsymbol{\theta}^-} \right)$. In practice, the oracle $\mathbf{v}(\mathbf{x}_t, t)$ is approximated either by (i) a pre-trained diffusion model (*distillation*), or (ii) the conditional velocity $\alpha_t' \mathbf{x}_0 + \sigma_t' \boldsymbol{\epsilon}$ (*training from scratch*). CTM and MF share the same framework with equivalent losses up to a constant (see Appendix A and Equation (11)), differing only in parameterization and backbone (CTM with EDM, MF with flow matching). We therefore use MF as the representative flow map model $\boldsymbol{\Psi}_{t \to s}$.

## 2.3 RELATED WORK

**Diffusion Models and ODE Samplers.** The PF-ODE enables deterministic ODE sampling (Song et al., 2021; Lipman et al., 2023; Liu et al., 2023; Karras et al., 2022). DDIM is a first-order (Euler-type) PF-ODE discretization that reduces steps without retraining (Song et al., 2020). DPM-Solver methods use a log-SNR parameterization with 2nd/3rd-order exponential integrators to solve the PF-ODE in very few NFEs (Lu et al., 2022; 2025), and parallel schemes can further optimize the discretization grid (Sabour et al., 2024; Nguyen et al., 2024).

**Few-Step Generative Models.** Because stepwise ODE integration is slow, distillation approaches train a student to imitate a multi-step teacher via long jumps (Salimans & Ho, 2022; Luhman & Luhman, 2021; Zheng et al., 2023). CM instead learn a direct flow map via consistency property; subsequent work improves training and extends to continuous time (Song et al., 2023; Song & Dhariwal, 2024; Geng et al., 2025b; Lu & Song, 2025). CTM learn maps between arbitrary times but rely on adversarial losses (Kim et al., 2024; Lai et al., 2023). Later work removes the adversarial component with new parameterizations and losses while improving few-step fidelity (Frans et al., 2025; Boffi et al., 2024; Sabour et al., 2025). MF (Geng et al., 2025a) trains from scratch without adversarial loss, at the cost of expensive Jacobian–vector products. Despite recent advances (Lai et al., 2025), training flow map models remains computationally expensive and often unstable.

## 3 CONSISTENCY MID-TRAINING FOR FAST, GENERAL FLOW MAP LEARNING

### 3.1 PROPOSED PIPELINE FOR FLOW MAP LEARNING

Despite recent advances, large-scale flow map training remains costly, unstable, and configuration-sensitive. The key challenge is the lack of an oracle regression target $\boldsymbol{\Psi}_{t\to s}$: current methods rely on stop-gradients of imperfect models, yielding poor supervision and large deviations from the true flow. To address this, we introduce a compact mid-training stage between pre-training and flow map post-training. Specifically, our pipeline incorporates the proposed CMT as a mid-training step, providing a general and cost-efficient framework for flow map learning:

**Stage 1: Pre-Training.** It aims to obtain a deterministic ODE sampler that transports samples from $p_{\text{prior}}$ to $p_{\text{data}}$, consistent with the marginals of the forward noising process $\mathcal{N}(\cdot; \alpha_t \mathbf{x}_0, \sigma_t^2 \mathbf{I})$. A practical choice is an off-the-shelf pre-trained diffusion model with its PF-ODE solver, as many such models are available (Peebles & Xie, 2023; Karras et al., 2024b; Ma et al., 2024). Alternatively, one may use a lightweight few-step flow map model supporting deterministic sampling (e.g., MF). We call these variants collectively the *teacher sampler*.

**Stage 2: Mid-Training (CMT).** Efficiently learn a lightweight, trajectory-aligned proxy of the target flow map with minimal computation and stable convergence, without ad-hoc heuristics. CMT's loss is designed to match the objectives of post-training while using fixed, explicit regression targets supplied by the teacher (see Equation (7) or Equation (8) below). Operationally, CMT learns to jump directly between points on the teacher-generated trajectory of a pre-trained model. Because the targets are fixed and high quality, CMT trains stably and yields a trajectory-aligned initializer.

**Stage 3: Post-Training.** Learn the final few-step flow-map model. Compared to random initialization or initialization from pre-trained diffusion models proposed by literature (Geng et al., 2025b; Lu & Song, 2025), the CMT initializer is trajectory-aligned, making post-training more stable, simpler, and faster (as supported by our theoretical analysis in Theorem 5.1 and Appendix F). CMT offers a general recipe for significantly cost-efficient flow map learning.

In what follows, we detail the mid-training stage with CMT, first instantiating it for CM ($\boldsymbol{\Psi}_{t\to 0}$) and then extending it to the general flow map via MF ($\boldsymbol{\Psi}_{t\to s}$), with pseudocode in Algorithm 1.

### 3.2 CMT FOR LEARNING CONSISTENCY FUNCTION

Here, we focus on CM as the flow map post-training stage. To obtain a trajectory aligned initializer for this flow map and to motivate the design of the CMT's mid-training loss, we revisit the CM oracle objective $\mathcal{L}_{\text{oracle-CM}}$.

We first propose a reinterpretation of $\mathcal{L}_{\text{oracle-CM}}$ from a reverse time generative perspective, under which the objective becomes transparent. Every point $\mathbf{x}_t \sim p_t$ along a PF-ODE trajectory is uniquely determined by its terminal state $\mathbf{x}_T$. Hence, one may sample a single terminal state $\mathbf{x}_T \sim p_{\text{prior}}$ and trace its entire trajectory backward. Training then reduces to mapping every point on this reverse path to its single consistent origin in the data distribution $p_{\text{data}}$. This yields the following equivalent formulation of the oracle loss; the proof is provided in Appendix F.1.

**Theorem 3.1.** *If $p_{\text{prior}}$ matches the diffused marginal $p_T$[1], the oracle loss can be expressed as*

$$\mathcal{L}_{\text{oracle-CM}}(\boldsymbol{\theta}) = \mathbb{E}_t \, \mathbb{E}_{\mathbf{x}_T \sim p_{\text{prior}}} \Big[ w(t) d(\mathbf{f}_{\boldsymbol{\theta}}\big(\boldsymbol{\Psi}_{T \to t}(\mathbf{x}_T), t\big), \boldsymbol{\Psi}_{T \to 0}(\mathbf{x}_T)) \Big]. \tag{6}$$

Building on the reverse-time formulation in Equation (6), we now introduce the training objective of the proposed CMT. On the interval $[0, T]$, we fix a time grid $0 = t_0 < t_1 < \cdots < t_M = T$ with $M$ discretization steps. Given a sample $\mathbf{x}_T \sim p_{\text{prior}}$, we obtain a discrete reference trajectory $\{\hat{\mathbf{x}}_{t_i}\}_{i=0}^{M}$ by running a numerical ODE solver with the pre–trained diffusion model $\mathbf{D}_{\phi}$ (in EDM formulation) as the teacher sampler, anchored at $\hat{\mathbf{x}}_{t_M} = \mathbf{x}_T$. The goal of CMT is for $\mathbf{f}_{\boldsymbol{\theta}}$ to match any intermediate state $\hat{\mathbf{x}}_{t_i}$ back to its clean origin $\hat{\mathbf{x}}_{t_0}$. Training proceeds by minimizing the following loss:

$$\mathcal{L}_{\text{CMT-CM}}(\boldsymbol{\theta}) := \mathbb{E}_i \mathbb{E}_{\mathbf{x}_T \sim p_{\text{prior}}} \Big[ d(\mathbf{f}_{\boldsymbol{\theta}}\big(\hat{\mathbf{x}}_{t_i}, t_i\big), \hat{\mathbf{x}}_{t_0}) \Big]. \tag{7}$$

This objective is a discrete approximation of the oracle loss $\mathcal{L}_{\text{oracle-CM}}$, since the solver-generated points approximate the true flow map, i.e., $\hat{\mathbf{x}}_{t_i} \approx \boldsymbol{\Psi}_{T \to t_i}(\mathbf{x}_T)$.

Since the starting states $\mathbf{x}_T \sim p_{\text{prior}}$ are randomly sampled, the set of possible trajectories can be arbitrarily many. Yet, once a particular $\mathbf{x}_T$ is fixed, the corresponding trajectory is uniquely determined. In principle, CMT can thus be trained with arbitrarily many distinct trajectories, avoiding the overfitting issues that arise in standard supervised tasks.

### 3.3 CMT FOR LEARNING GENERAL FLOW MAP

We now focus on the MF parameterization for the general flow map learning. MF aims to learn the average drift, defined as $\mathbf{h}(\mathbf{x}_t, t, s) := \frac{1}{t-s} \int_s^t \mathbf{v}(\mathbf{x}_u, u) \, \mathrm{d}u$, which aggregates the ODE velocity over the interval $[s, t]$. We observe that this quantity can also be expressed through the flow map. Let $\mathbf{x}_T$ denote the initial state at time $T$ on the same PF-ODE trajectory as $\mathbf{x}_t$. Then

$$\mathbf{h}(\mathbf{x}_t, t, s) = \frac{1}{t-s} \Big( \int_s^T \mathbf{v}(\mathbf{x}_u, u) \, \mathrm{d}u - \int_t^T \mathbf{v}(\mathbf{x}_u, u) \, \mathrm{d}u \Big) = \frac{1}{t-s} \big( \boldsymbol{\Psi}_{T \to t}(\mathbf{x}_T) - \boldsymbol{\Psi}_{T \to s}(\mathbf{x}_T) \big).$$

Motivated by this decomposition, CMT allows to construct a teacher–reference trajectory $\{\hat{\mathbf{x}}_{t_i}\}$ from a prior sample $\mathbf{x}_T \sim p_{\text{prior}}$ using two possible teacher samplers. The first employs a numerical ODE solver applied to the PF-ODE of a pre–trained flow matching model $\mathbf{v}_{\phi}$. Alternatively, since MF supports deterministic sampling, we may use a smaller and lightweight MF model to perform multi–step deterministic generation. Although not optimal, this model is much easier to train and still yields a valid teacher trajectory. In both cases, the resulting trajectory provides a feasible approximation of the oracle states, $\boldsymbol{\Psi}_{T \to t_i}(\mathbf{x}_T) \approx \hat{\mathbf{x}}_{t_i}$.

The CMT loss for MF then encourages the average drift parametrization $\mathbf{h}_{\boldsymbol{\theta}}$ to align with the finite differences between successive reference states:

$$\mathcal{L}_{\text{CMT-MF}}(\boldsymbol{\theta}) = \mathbb{E}_{i > j} \, \mathbb{E}_{\mathbf{x}_T \sim p_{\text{prior}}} \Big[ \big\| \mathbf{h}_{\boldsymbol{\theta}}(\hat{\mathbf{x}}_{t_i}, t_i, t_j) - \frac{\hat{\mathbf{x}}_{t_i} - \hat{\mathbf{x}}_{t_j}}{t_i - t_j} \big\|_2^2 \Big]. \tag{8}$$

Crucially, in both Equation (7) and Equation (8), our formulation reduces training to a standard regression problem with a fixed target, either $\hat{\mathbf{x}}_0$ or $\frac{\hat{\mathbf{x}}_{t_i} - \hat{\mathbf{x}}_{t_j}}{t_i - t_j}$. The CMT loss for MF generalizes the CM case. In fact, if we fix $t_j = 0$ in $\mathcal{L}_{\text{CMT-MF}}$, the loss reduces to learning a mapping from every point on the trajectory directly to the clean data, thereby recovering the CM formulation.

---

[1]This holds for sufficiently large $T$ as in EDM, or with appropriate noise schedules as in flow matching. Empirically, $p_T$ (data-dependent) and $p_{\text{prior}}$ (data-free) perform identically, so we adopt $p_{\text{prior}}$ in all experiments.

## 4 EXPERIMENTAL RESULTS

### 4.1 EXPERIMENTAL SETUPS

Table 1: Sample quality on unconditional CIFAR-10 32×32 and class-conditional ImageNet 64×64.

| Unconditional CIFAR-10 32×32 | | | Class-Conditional ImageNet 64×64 | | |
|---|---|---|---|---|---|
| **METHOD** | **NFE** (↓) | **FID** (↓) | **METHOD** | **NFE** (↓) | **FID** (↓) |
| **Diffusion Models** | | | **Diffusion Models (*Auto-Guidance)** | | |
| EDM (Karras et al., 2022) | 35 | 2.01 | RIN (Jabri et al., 2023) | 1000 | 1.23 |
| | | | EDM2 (Karras et al., 2024b) | 63 | 1.33 |
| **Joint Training** | | | EDM2* (Karras et al., 2024a) | 63 | **1.01** |
| CTM (Kim et al., 2024) | 1 | **1.87** | **Joint Training** | | |
| DMD (Yin et al., 2024b) | 1 | 3.77 | DMD2 (Yin et al., 2024a) | 1 | **1.28** |
| SiD (Zhou et al., 2024) | 1 | 1.92 | SiD (Zhou et al., 2024) | 1 | 1.52 |
| **Diffusion Distillation** | | | CTM (Kim et al., 2024) | 1 / 2 | 1.92 / 1.73 |
| DFNO (Zheng et al., 2023) | 1 | 3.78 | **Auto-Guidance Diffusion Distillation** | | |
| 2-Rectified Flow (Liu et al., 2023) | 1 | 4.85 | AYF (Sabour et al., 2025) | 1 / 2 | 2.98 / **1.25** |
| TRACT (Berthelot et al., 2023) | 1 / 2 | 3.78 / 3.32 | ECD (Geng et al., 2025b) | 1 / 2 | 2.24 / 1.50 |
| PD (Salimans & Ho, 2022) | 1 / 2 | 8.34 / 5.58 | CMT (w/ ECD) (Ours) | 1 / 2 | **1.78** / 1.32 |
| **Flow Map Models** | | | **Flow Map Models** | | |
| CD (Song et al., 2023) | 1 / 2 | 3.55 / 2.93 | CD (Song et al., 2023) | 1 / 2 | 6.20 / 4.70 |
| iCT (Song & Dhariwal, 2024) | 1 / 2 | 2.83 / 2.46 | iCT (Song & Dhariwal, 2024) | 1 / 2 | 4.02 / 3.20 |
| iCT-deep (Song & Dhariwal, 2024) | 1 / 2 | 2.51 / 2.24 | iCT-deep (Song & Dhariwal, 2024) | 1 / 2 | 3.25 / 2.77 |
| ECT (Geng et al., 2025b) | 1 / 2 | 3.60 / 2.11 | ECT (Geng et al., 2025b) | 1 / 2 | 2.49 / 1.67 |
| sCT (Lu & Song, 2025) | 1 / 2 | 2.85 / 2.06 | sCT (Lu & Song, 2025) | 1 / 2 | 2.04 / 1.48 |
| sCD (Lu & Song, 2025) | 1 / 2 | 3.66 / 2.52 | sCD (Lu & Song, 2025) | 1 / 2 | 2.44 / 1.66 |
| Stable CT (Wang et al., 2025) | 1 / 2 | 2.92 / 2.02 | MultiStep-CD (Heek et al., 2024) | 1 / 2 | 3.20 / 1.90 |
| VCT (Silvestri et al., 2025) | 1 / 2 | 3.26 / 2.02 | Stable CT (Wang et al., 2025) | 1 / 2 | 2.42 / 1.55 |
| TCM (Lee et al., 2025) | 1 / 2 | **2.46** / 2.05 | VCT (Silvestri et al., 2025) | 1 / 2 | 4.93 / 3.07 |
| IMM (Zhou et al., 2025) | 1 / 2 | 3.20 / 1.98 | TCM (Lee et al., 2025) | 1 / 2 | 2.20 / 1.62 |
| MF (Geng et al., 2025b) | 1 | 2.92 | | | |
| CMT (w/ ECT) (Ours) | 1 / 2 | 2.74 / **1.97** | CMT (w/ ECT) (Ours) | 1 / 2 | **2.02** / 1.48 |

**Datasets & Setup.** We evaluate on CIFAR10 at 32×32 (Krizhevsky et al., 2009), AFHQv2 at 64×64, FFHQ at 64×64 (Karras et al., 2022), and ImageNet (Deng et al., 2009) at 64×64, 256×256, and 512×512. The low-resolution *unconditional* datasets (CIFAR10, AFHQv2, FFHQ) follow EDM/ECT/VCT protocols (Karras et al., 2022; Geng et al., 2025b; Silvestri et al., 2025). For ImageNet 64×64 and 512×512, we adopt EDM2 (Karras et al., 2024b), training the 512×512 case in the latent space of Stable Diffusion (SD) autoencoders. For ImageNet 256×256, we follow MF and SiT (Geng et al., 2025a; Ma et al., 2024), also in the SD latent space. Detailed experimental setup is provided in Appendix B.

**Teachers and Solvers for CMT's Mid-Training.** Across datasets, CMT employs different teacher–solver pairs based on availability: EDM w/ DPM-Solver++ (Lu et al., 2022; 2025) on {CIFAR10, AFHQv2, FFHQ}; EDM2 w/ DPM-Solver++ on {ImageNet 64×64, 512×512}; and MF-B/4 on ImageNet 256×256. DPM-Solver++ uses 16 solver steps and MF-B/4 uses 8 with fixed discretization. We adopt these solvers with their default log-SNR sampling schedulers due to their strong theoretical foundations and keep all other settings as simple as possible. For mid-training of EDM/EDM2-related settings, we apply a learned perceptual loss to align CMT's predictions with the teacher's high-fidelity outputs, specifically using LPIPS (Zhang et al., 2018) in pixel space and ELatentLPIPS (Kang et al., 2024) in latent space. We use squared $\ell^2$ loss for MF. We discuss CMT loss function selection motivation in Appendix D.

**Post-Training of Flow Map Model.** After mid-training, we respectively train a flow map with: ECT on {CIFAR-10, AFHQv2, FFHQ}, ECT/ECD on ImageNet 64×64, MF on ImageNet 256×256, and ECD on ImageNet 512×512 for stability in very high dimensions (Lu & Song, 2025). We remark that ECT and MF serve as strong representatives of flow map models: ECT builds on EDM's backbone, while MF builds on FM's backbone, both of which are widely used in practice. These post-training methods are chosen due to public availability and representativeness on respective datasets.

Table 2: Comparison between various CMs given the identical 51.2 million training images budget on AFHQv2 64×64 and FFHQ 64×64. Our CMT achieve the best 1-step and 2-step FIDs.

| Unconditional AFHQv2 64×64 | | | Unconditional FFHQ 64×64 | | |
|---|---|---|---|---|---|
| **METHOD** | **NFE ($\downarrow$)** | **FID ($\downarrow$)** | **METHOD** | **NFE ($\downarrow$)** | **FID ($\downarrow$)** |
| iCT (Song & Dhariwal, 2024) | 1 / 2 | 5.40 / 2.92 | iCT (Song & Dhariwal, 2024) | 1 / 2 | 5.80 / 4.02 |
| ECT (Geng et al., 2025b) | 1 / 2 | 3.89 / 2.61 | ECT (Geng et al., 2025b) | 1 / 2 | 5.99 / 4.39 |
| VCT (Silvestri et al., 2025) | 1 / 2 | 3.84 / 2.71 | VCT (Silvestri et al., 2025) | 1 / 2 | 5.47 / 4.16 |
| CMT (w/ ECT) (Ours) | 1 / 2 | **3.28 / 2.34** | CMT (w/ ECT) (Ours) | 1 / 2 | **3.89 / 2.75** |

**Metrics.** We report FID (Heusel et al., 2017), data cost in millions of images (Mimgs) for data efficiency, and training A100 GPU (80GB) time for convergence speed. Specifically, the data cost is computed via batch size per iteration × total iterations, where each batch is randomly drawn from the entire dataset at each iteration, i.e., the data cost equals the number of backpropagated inputs.

## 4.2 MID-TRAINING WITH CMT IMPROVES FLOW MAP POST-TRAINING

In this section, we benchmark CMT against baselines across datasets. Because ECT and related distillation methods (e.g., the distilled variant of MF) typically start post-training from the weights of a pre-trained diffusion model, the most direct and fair evaluation of our mid-training strategy is to compare ECT vs. CMT (w/ ECT) and MF vs. CMT (w/ MF). Beyond these direct comparisons, we also report broader results against other baselines in terms of both FID and training cost.

**CIFAR-10 and ImageNet 64×64.** Table 1 shows that CMT (w/ ECT) attains SOTA with 2-step FID=1.97 on CIFAR-10, surpassing the teacher EDM's 2.01 with 35 steps and outperforming prior CMs and flow maps. On ImageNet 64×64, CMT (w/ ECT) achieves the best FIDs among all CMs and flow maps; our 2-step FID=1.48 is a satisfactory choice over EDM's 1.33 with 63 NFEs. Further, we follow AYF (Sabour et al., 2025) to distill a strong EDM2 with Auto-Guidance to surpass the vanilla flow map model, improving the 1/2-step FID to 1.78/1.32 for CMT (w/ ECD).

For **CIFAR-10**, the total budget is 51.2 Mimgs (38.4 Mimgs mid-training, 12.8 Mimgs post-training). Under the same 51.2 Mimgs, CMT beats ECT and VCT. sCT uses more budget; under the same 51.2 Mimgs, sCT's 1-step FID is 3.09 vs. our 2.74, demonstrating better data efficiency. Stable CT requires 153.6 Mimgs yet still trails our 51.2 Mimgs results. TCM reaches comparable SOTA but at 332.8 Mimgs. For **ImageNet 64×64**, CMT (w/ ECT) uses only 12.8 Mimgs (6.4 mid, 6.4 post), whereas ECT and Stable CT use 102.4 Mimgs, sCT 819.2 Mimgs, and TCM 143.36 Mimgs. Compared to sCT, we save up to 98% training images. Because CMT has lower per-iteration cost than sCT (no expensive JVP), we also cut GPU time by 98% while achieving SOTA. Meanwhile, CMT (w/ ECD) uses only 19.2 Mimgs (6.4 mid, 12.8 post), while vanilla ECD and AYF use 102.4 Mimgs.

Overall, CMT is both SOTA and highly data-efficient. Notably, CMT with ECT/ECD as the post-training flow map consistently outperforms vanilla ECT/ECD under equal or lower budgets, underscoring the importance of our mid-training. Additional CIFAR-10 evidence appears in Section 4.3.

**AFHQv2 and FFHQ 64×64.** Table 2 compares CMs under a 51.2 Mimgs budget. Since AFHQv2 and FFHQ are also unconditional, we directly transfer the CIFAR-10 hyperparameters. CMT achieves the best 1-step and 2-step FIDs, and with ECT as post-training again outperforms ECT at the same budget, highlighting both hyperparameter robustness and the critical role of CMT across datasets.

**ImageNet 512×512.** Table 3 reports results along with training costs (Mimgs) for flow map models and CMT, and their comparison with diffusion models. For post-training, we use ECD, making CMT directly comparable to vanilla ECD. CMT substantially outperforms vanilla ECD, again confirming the critical role of mid-training. Overall, CMT achieves the best 2-step FID=1.84 and a competitive 1-step FID=3.46 at dramatically 93% lower cost than previous sCD. The same advantage holds for GPU time, since sCD requires costly JVP computations per iteration. Random samples generated by the trained CMT (w/ ECD) are shown in Figure 3.

**Remark: Simplicity and Stability of CMT on CM-family Experiments.** In mid-training, CMT learns a flow map proxy via an explicit regression target, avoiding stop-gradients, custom time sampling, and handcrafted weights $w(t)$, leading to stable training. Although reference trajectories require a diffusion ODE solver, few-step (∼16) methods such as DPM-Solver++ suffice, and the multistep scheme reuses past states $\hat{x}_{t_k}$ to build later ones, keeping overhead low. On ImageNet

Table 3: Sample quality on class-conditional ImageNet 512×512 of diffusion models and flow map models. The cost comparisons for flow map models and CMT are measured under millions of training images (Mimgs).

| METHOD | NFE (↓) | FID (↓) | METHOD | NFE (↓) | FID (↓) | Cost (↓) |
|---|---|---|---|---|---|---|
| **Diffusion Models (\*Auto-Guidance)** | | | **Flow Map Models** | | | |
| RIN (Jabri et al., 2023) | 1000 | 3.95 | ECT (Geng et al., 2025b) | 1 / 2 | 9.98 / 6.28 | 204.8 |
| EDM2 (Karras et al., 2024b) | 63×2 | 1.81 | ECD (Geng et al., 2025b) | 1 / 2 | 8.47 / 3.38 | 409.6 |
| EDM2\* (Karras et al., 2024a) | 63×2 | **1.25** | sCT (Lu & Song, 2025) | 1 / 2 | 4.29 / 3.76 | 204.8 |
| DiT (Peebles & Xie, 2023) | 250×2 | 3.04 | sCD (Lu & Song, 2025) | 1 / 2 | **2.28** / 1.88 | 409.6 |
| Large-DiT (Zhang et al., 2023) | 250×2 | 2.52 | AYF (Sabour et al., 2025) | 1 / 2 | 3.32 / 1.87 | 102.4 |
| SiT (Ma et al., 2024) | 250×2 | 2.62 | CMT (w/ ECD) (Ours) | 1 / 2 | 3.38 / **1.84** | **28.8** |

Table 4: Comparison between CMT and MF on ImageNet 256×256.

| Method | Pre-Training | Mid-Training | Post-Training | Total Time (↓) | FID (↓) |
|---|---|---|---|---|---|
| MF-XL/2 (Rand. Init.) | 0 | 0 | 1520 hours | 1520 hours | 3.43 |
| MF-XL/2 (SiT Init.) | >1520 hours | 0 | 357 hours | >1520 hours | 4.52 |
| CMT-XL/2 | 38 hours | 135 hours | 587 hours | **760 hours** | **3.34** |

$64 \times 64$ and $512 \times 512$, this yields clear gains: CMT outperforms sCT/sCD while cutting mid- and post-training data and GPU time by 93%–98% (see Appendices B and C).

CMT initialization simplifies post-training of ECT/ECD models on {CIFAR-10, ImageNet 64×64/512×512, AFHQv2, FFHQ}, removing ad hoc tricks such as $\Delta t$ annealing, loss reweighting, custom time sampling, EMA variants, and nonlinear learning-rate schedules. The resulting pipeline consistently outperforms ECT/ECD baselines and converges faster with minimal engineering.

**ImageNet 256×256: CMT Enables Flexible Teacher Samplers Beyond Diffusion.** We test whether CMT's mid-training can use a non-diffusion teacher sampler, even if its quality is low. We compare performance on a larger MF-XL/2 model under three settings: (1) **MF-XL/2 (Rand. Init.)**: post-train only (vanilla MF with random initialization); (2) **MF-XL/2 (SiT Init.)**: pre-train with SiT (Zhu, 2025) followed by post-training with its weights as initialization; (3) **CMT-XL/2**: train a small MF-B/4 for quick convergence, use it in mid-training as a teacher sampler to generate ODE trajectories for XL/2 with random initialization, then post-train MF-XL/2 initialized from the mid-trained model (see Appendix B.4).

Table 4 reports the pre-training, mid-training, and post-training time together with the final FID. In particular, CMT-XL/2 cuts total training time by 50% compared to the other two settings, while achieving even better FID. Even though MF-B/4 is a

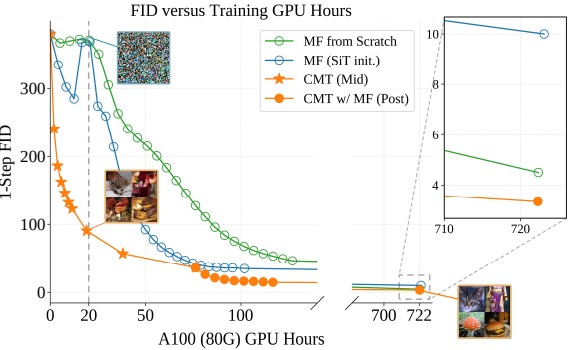

Figure 2: **FID vs. training time for vanilla MF and CMT (ours) on ImageNet** 256×256. We perform mid-training starting from a randomly-initialized XL/2 model, where CMT of XL/2 size learns to match the deterministic sampler of a weaker, smaller teacher MF-B/4. The resulting mid-trained weights of CMT-XL/2 are then used to initialize MF-XL/2 post-training. This initialization produces semantically meaningful samples early and drives significantly faster convergence. With CMT's pipeline, training reaches lower FID in only half the GPU hours compared to MF trained from scratch. MF initialized from SiT also converges fast, but requires more than 1520 hours of pre-training, which exceeds the cost of training MF itself.

weak teacher (1 / 2 / 8-step FID = 24.47 / 14.96 / 13.44), using it in CMT substantially accelerates MF-XL/2 training, which converges faster and achieves better FID than vanilla MF. By contrast, SiT pre-training at XL/2 scale requires very long training and its weights lead to unstable MF post-training (Zhu, 2025), making it impractical as a pre-training or mid-training teacher. We also provide qualitative comparisons in Figure 2. After 20 GPU hours, both MF-XL/2 (Rand. Init.) and MF-XL/2 (SiT Init.) still produce noise, whereas CMT already generates semantically meaningful images.

Table 5: Comparison between MF (initialized with random weights or pretrained SiT) and CMT (MF initialized with CMT mid-trained weights) on the MS-COCO text-to-image (T2I) dataset.

| Method | Pre-Training | Mid-Training | Post-Training | Total Time ($\downarrow$) | 1/2-Step FID ($\downarrow$) |
|---|---|---|---|---|---|
| MF (Rand. Init.) | 0 | 0 | 60 hours | 60 hours | 15.97 / 5.42 |
| MF (SiT Init.) | 22 hours | 0 | 38 hours | 60 hours | 15.55 / 5.26 |
| CMT | 22 hours | 1 hours | 9 hours | **32 hours** | **15.12 / 5.01** |

Eventually, CMT attains superior FID with only half the total GPU hours compared to training MF from scratch.

These results show that diffusion initialization alone is insufficient and that mid-training is crucial. Using B/4 for pre-training but XL/2 for mid-training further demonstrates that CMT's mid-training is architecture-agnostic, since it directly learns a map aligned with the teacher sampler.

**MS-COCO Text-to-Image (T2I): CMT Accelerates and Improves T2I Generation.** We follow the setup of U-ViT (Bao et al., 2023) and train MF with the MM-DiT architecture (Esser et al., 2024) on MS-COCO (Lin et al., 2014) from scratch, and we use the validation set for FID evaluation. We consider three MF configurations: (1) **MF (Rand. Init.)**: MF is trained only in the post-training stage, starting from random initialization; (2) **MF (SiT Init.)**: we first pre-train a SiT model and then post-train MF initialized from the SiT weights; (3) **CMT**: we pre-train SiT, use it as the teacher sampler in the CMT mid-training stage to generate ODE trajectories for an MF model initialized from SiT, and finally post-train MF starting from this mid-trained initializer. Detailed hyperparameters and training settings are provided in Appendix B.6, and largely follow the ImageNet $256 \times 256$ configuration, since MS-COCO shares the same resolution.

Table 5 shows that CMT also performs well for T2I, with significantly faster convergence: it achieves the best FID while reducing total training time by about $47\%$. CMT's 2-step FID (5.01) is also close to the teacher SiT's 50-step FID (4.73). We further note that the relatively large 1-step FIDs observed across all variants may be attributable to limitations of the dataset rather than to CMT itself: within a fixed backbone, CMT consistently improves performance and convergence speed over alternative initialization schemes. CMT should therefore be viewed as a general recipe for achieving faster and more stable convergence, not as a modification of the backbone architecture. Together with the ImageNet $256 \times 256$ results, these experiments demonstrate that both a smaller MF and a SiT diffusion model can serve as CMT's teacher and yield strong performance even in T2I generation, highlighting the generality and effectiveness of CMT across problem settings and model architectures.

## 4.3 Empirical Analysis of the Proposed Mid-Training Scheme

**Importance of Mid-Training.** We further assess the importance of our mid-training stage on CIFAR-10 using ECT as the flow map model of $\mathbf{\Psi}_{t \to 0}$. To fairly isolate CMT's mid-training, post-training and vanilla ECT share the same hyperparameters (constant $\Delta t = 1/256$, learning rate $10^{-4}$), with all other settings held fixed. We evaluate:

- **Vanilla ECT:** Under a 51.2Mimgs budget, we obtain the final 1-step / 2-step FID as 3.54 / 2.12.
- **CMT$^{\text{short}}$:** Short mid-training for 1.28Mimgs + long post-training for 49.92Mimgs. We obtain the final 1-step / 2-step FID as 3.42 / 2.11.
- **CMT$^{\text{long}}$:** Long mid-training for 25.6Mimgs + short post-training for 25.6Mimgs. We obtain the final 1-step / 2-step FID as 3.30 / 2.04.

These results show that longer mid-training outperforms shorter mid-training, which in turn improves over no mid-training, further confirming the importance of CMT 's mid-training stage.

**Ablation Study on Potential Alternatives for Post-Training Initialization.** Although we have demonstrated the efficiency of CMT as a mid-training scheme, other variants are possible. We focus on learning the flow map $\mathbf{\Psi}_{t \to 0}$ and highlight two straightforward alternatives. Following our mid-training design principles, these variants aim to be stable and easy to train (e.g., stop-gradient-free regression targets) while using fewer ad-hoc tricks and hyperparameters:

$$\mathcal{L}_{\text{var}}^{(1)}(\boldsymbol{\theta}) := \mathbb{E}_{t, p_t} \left[ \|\mathbf{f}_{\boldsymbol{\theta}}(\mathbf{x}_t, t) - \hat{\mathbf{x}}_0(\mathbf{x}_t)\|_2^2 \right], \quad \mathcal{L}_{\text{var}}^{(2)}(\boldsymbol{\theta}) := \mathbb{E}_{\mathbf{x}_T \sim p_{\text{prior}}} \left[ \|\mathbf{f}_{\boldsymbol{\theta}}(\mathbf{x}_T, T) - \hat{\mathbf{x}}_0(\mathbf{x}_T)\|_2^2 \right],$$

where $\hat{\mathbf{x}}_0(\mathbf{x}_t)$ denotes the estimate at $t{=}0$ obtained by running the solver with the pre-trained diffusion model starting from the forward-perturbed sample $\mathbf{x}_t$, and $\hat{\mathbf{x}}_0(\mathbf{x}_T)$ is obtained similarly by running backward from a prior sample $\mathbf{x}_T$. We refer to $\mathcal{L}_{\text{var}}^{(1)}(\boldsymbol{\theta})$ as *Slow CMT*: it follows the same principle as CMT but uses only the endpoints of the ODE trajectory, ignoring intermediate states, so generating the same amount of training data requires substantially more ODE-solver inference. The loss $\mathcal{L}_{\text{var}}^{(2)}(\boldsymbol{\theta})$ is the standard *knowledge distillation (KD)* objective of Luhman & Luhman (2021).

We compare CMT's mid-training with KD and Slow CMT on CIFAR-10, training all three under the same settings and then post-training ECT for flow-map learning for 12.8Mimgs with each initialization. Comparing CMT with KD, CMT's 1-step / 2-step FID (2.74 / 1.97) is significantly better than KD's (3.54 / 2.19), confirming the benefit of learning intermediate steps. Comparing CMT with Slow CMT, their FIDs are similar (2.74 / 1.97 vs. 2.75 / 1.98), but Slow CMT's mid-training costs roughly $3\times$ more GPU time because its regression targets cannot reuse intermediate ODE states.

## 5 THEORETICAL ANALYSIS

In mid–training, CMT learns a reliable proxy of the flow map, which then serves as a well–aligned initialization for post–training. To quantify this effect, we analyze the CM flow map $\boldsymbol{\Psi}_{t\to 0}$ in the post–training stage; the same reasoning applies to general flow maps $\boldsymbol{\Psi}_{t\to s}$ such as MF.

We compare how well the surrogate CM objective tracks an oracle objective under three initialization schemes: CMT mid–training ($\boldsymbol{\theta}_{\text{CMT}}$), a pre–trained diffusion model ($\boldsymbol{\theta}_{\text{DM}}$), and random initialization ($\boldsymbol{\theta}_{\text{rand}}$). For simplicity, we consider the squared distance $d(\mathbf{x}, \mathbf{y}) = \|\mathbf{x} - \mathbf{y}\|_2^2$ and a uniform weight $w(t) \equiv 1$. Let $\mathcal{L}_{\text{oracle-CM}}(\boldsymbol{\theta})$ and $\mathcal{L}_{\text{CM}}(\boldsymbol{\theta})$ denote the oracle and surrogate CM objectives (see Equations (2) and (3)). We define the *gradient bias* $\mathcal{B}(\boldsymbol{\theta}) := \left\|\nabla_{\boldsymbol{\theta}}\mathcal{L}_{\text{oracle-CM}}(\boldsymbol{\theta}) - \nabla_{\boldsymbol{\theta}}\mathcal{L}_{\text{CM}}(\boldsymbol{\theta})\right\|_2^2$, and evaluate $\mathcal{B}(\boldsymbol{\theta})$ at the initial parameters $\boldsymbol{\theta} \in \{\boldsymbol{\theta}_{\text{CMT}}, \boldsymbol{\theta}_{\text{DM}}, \boldsymbol{\theta}_{\text{rand}}\}$. A smaller $\mathcal{B}(\boldsymbol{\theta})$ means that SGD steps on $\mathcal{L}_{\text{CM}}$ closely follow those on the oracle objective. The following informal result summarizes the worst–case bias for each scheme.

**Theorem 5.1** (Informal Bias Comparison). *Fix a tolerance $\varepsilon > 0$ and a small time step $\Delta t$. Then:*

(i) **CMT Initialization.** *If $\mathcal{L}_{\text{CMT-CM}}(\boldsymbol{\theta}_{\text{CMT}}) < \varepsilon$, then $\mathcal{B}(\boldsymbol{\theta}_{\text{CMT}}) = \mathcal{O}(\varepsilon + \Delta t^2)$.*

(ii) **Diffusion Initialization.** *If $\mathcal{L}_{\text{DM}}(\boldsymbol{\theta}_{\text{DM}}) < \varepsilon$, then $\mathcal{B}(\boldsymbol{\theta}_{\text{DM}}) = \mathcal{O}\left(\varepsilon + \Delta t^2 + \mathbb{E}_t[\sigma_t^2/\alpha_t^2]\right) + \mathbb{E}_{t,\mathbf{x}_t}\left[\|\boldsymbol{\Psi}_{t\to 0}(\mathbf{x}_t) - \mathbb{E}[\mathbf{x}_0|\mathbf{x}_t]\|_2^2\right].$*

(iii) **Random Initialization.** *For random weights $\boldsymbol{\theta}_{\text{rand}}$, $\mathcal{B}(\boldsymbol{\theta}_{\text{rand}}) = \mathcal{O}(1)$.*

This comparison explains why CMT's mid–training provides a particularly robust starting point: it already yields a good proxy to the oracle flow map, so the CM gradient is nearly unbiased up to $\mathcal{O}(\varepsilon + \Delta t^2)$. In contrast, diffusion–based initialization (Geng et al., 2025b) necessarily incurs additional bias from the forward noising process and from the mismatch between the PF–ODE solution $\boldsymbol{\Psi}_{t\to 0}(\mathbf{x}_t)$ and the posterior mean $\mathbb{E}[\mathbf{x}_0|\mathbf{x}_t]$, while random initialization can be arbitrarily far from the oracle. We present the complete and rigorous version of this result in Theorem F.1, which also covers initialization from a trained consistency distillation model. In that case, the bias contains an additional, uncontrolled discrepancy term, and in practice its training often requires extra ad–hoc stabilization, further limiting robustness compared to CMT. We refer to Appendix F for the full bias–variance analysis and the resulting excess–risk guarantees.

## 6 CONCLUSION

We introduced CMT, an efficient mid-training stage that learns a trajectory-consistent initialization for flow map models from teacher sampler trajectories. This simple, architecture-agnostic step stabilizes optimization, removes reliance on stop-gradient targets and ad hoc time weighting, and accelerates convergence. With CMT as initialization, flow map models such as Consistency Models and Mean Flow attain SOTA two-step FIDs across pixel and latent benchmarks while reducing training data budget and GPU time by up to 98%. The approach makes training of flow map models more efficient and practical, and in principle, it applies to a broad class of ODE-based generative models.

## STATEMENT ON THE USE OF LARGE LANGUAGE MODELS

This work made use of large language models to assist with proofreading and improving the clarity of the writing. All research ideas, theoretical development, experiments, and coding were carried out solely by the authors.

## STATEMENT ON REPRODUCIBILITY

We provide all necessary code and models to reproduce CMT's results on CIFAR-10, AFHQv2, and FFHQ, together with the corresponding checkpoints, at the https://github.com/sony/cmt (also included in the abstract). Comprehensive details of our experimental configurations are further provided in Appendix B to ensure faithful reproducibility.

## STATEMENT ON ETHICS

As with other generative models, CMT can inadvertently produce harmful or inappropriate content (e.g., violent, deepfakes, or derogatory/NSFW materials). These risks are mitigated by enforcing comprehensive safety policies with automated content screening and moderation pipelines that prevent such outputs.

## ACKNOWLEDGMENT

We are grateful to our colleagues *Christian Simon*, *Murata Naoki*, *Yuhta Takida*, *Nguyen Bac*, and *Uesaka Toshimitsu* from Sony for helpful feedback.

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

TABLE OF CONTENTS

## A   RELATIONSHIP OF FLOW MAP MODELS

The principled objective for learning the general flow map $\boldsymbol{\Psi}_{t \to s}$ is to train a neural network $\mathbf{G}_{\boldsymbol{\theta}}(\mathbf{x}_t, t, s)$ by minimizing Equation (4) (Lai et al., 2025):

$$\mathcal{L}_{\text{oracle-CTM}}(\boldsymbol{\theta}) := \mathbb{E}_{t>s}\, \mathbb{E}_{\mathbf{x}_t \sim p_t}\left[ w(t)\, d\big(\mathbf{G}_{\boldsymbol{\theta}}(\mathbf{x}_t, t, s),\, \boldsymbol{\Psi}_{t \to s}(\mathbf{x}_t)\big)\right].$$

In CTM, the network is parameterized in a form inspired by an Euler step:

$$\mathbf{G}_{\boldsymbol{\theta}}(\mathbf{x}_t, t, s) := \frac{s}{t}\mathbf{x}_t + \frac{t-s}{t}\mathbf{g}_{\boldsymbol{\theta}}(\mathbf{x}_t, t, s).$$

Since the true flow map $\boldsymbol{\Psi}_{t \to s}$ cannot be accessed directly, CTM constructs a surrogate target using its own outputs, in the spirit of a stop-gradient approximation (as in CM). Concretely, it replaces the oracle with an intermediate reference:

$$\boldsymbol{\Psi}_{t \to s}(\mathbf{x}_t) \approx \mathbf{G}_{\boldsymbol{\theta}^-}(\boldsymbol{\Psi}_{t \to u}(\mathbf{x}_t), u, s), \quad \text{for} \quad t > u > s,$$

where $\boldsymbol{\Psi}_{t \to u}(\mathbf{x}_t)$ is obtained either by applying a few-step solver to a pre-trained diffusion model (distillation), or by using CTM's own parameterization $\mathbf{g}_{\boldsymbol{\theta}}(\mathbf{x}_t, t, t)$ to generate a self-teacher trajectory.

In contrast, MF takes a different perspective: instead of directly predicting $\boldsymbol{\Psi}_{t \to s}(\mathbf{x}_t)$, it parameterizes the network to approximate the *average drift* along the trajectory:

$$\mathbf{h}_{\boldsymbol{\theta}}(\mathbf{x}_t, t, s) \approx \mathbf{h}(\mathbf{x}_t, t, s) := \frac{1}{s-t}\int_t^s \mathbf{v}(\mathbf{x}_u, u)\, \mathrm{d}u.$$

Conceptually, CTM and MF share the same underlying framework, but differ in how the learned function is parameterized. The relation can be written as

$$\boldsymbol{\Psi}_{t \to s}(\mathbf{x}_t) = \frac{s}{t}\mathbf{x}_t + \frac{t-s}{t}\underbrace{\left[\mathbf{x}_t + \frac{t}{t-s}\int_t^s \mathbf{v}(\mathbf{x}_u, u)\, \mathrm{d}u\right]}_{\approx\, \mathbf{g}_{\boldsymbol{\theta}}}$$

$$= \mathbf{x}_t + (s-t)\underbrace{\left[\frac{1}{s-t}\int_t^s \mathbf{v}(\mathbf{x}_u, u)\, \mathrm{d}u\right]}_{\approx\, \mathbf{h}_{\boldsymbol{\theta}}}.$$

Thus, CTM can be seen as approximating the first form via $\mathbf{g}_{\boldsymbol{\theta}}$, while MF approximates the second via $\mathbf{h}_{\boldsymbol{\theta}}$. Their backbone choices also differ: CTM builds on EDM, whereas MF builds on flow matching.

Let

$$\mathbf{g}_{\boldsymbol{\theta}}(\mathbf{x}_t, t, s) := \mathbf{x}_t - t\, \mathbf{h}_{\boldsymbol{\theta}}(\mathbf{x}_t, t, s),$$

and take the distance to be the squared norm $d(\mathbf{x}, \mathbf{y}) := \|\mathbf{x} - \mathbf{y}\|^2$. Substituting into Equation (4), we can expand the loss term as

$$
\begin{aligned}
& d\big(\mathbf{G}_{\boldsymbol{\theta}}(\mathbf{x}_t, t, s), \boldsymbol{\Psi}_{t \to s}(\mathbf{x}_t)\big) \\
&= \|\mathbf{G}_{\boldsymbol{\theta}}(\mathbf{x}_t, t, s) - \boldsymbol{\Psi}_{t \to s}(\mathbf{x}_t)\|^2 \\
&= \left\| \left( \frac{s}{t}\mathbf{x}_t + \frac{t-s}{t}\, \mathbf{g}_{\boldsymbol{\theta}}(\mathbf{x}_t, t, s) \right) - \left( \frac{s}{t}\mathbf{x}_t + \frac{t-s}{t}\left[ \mathbf{x}_t + \frac{t}{t-s}\int_t^s \mathbf{v}(\mathbf{x}_u, u)\, \mathrm{d}u \right] \right) \right\|^2 \\
&= \left( \frac{t-s}{t} \right)^2 \left\| \mathbf{g}_{\boldsymbol{\theta}}(\mathbf{x}_t, t, s) - \left( \mathbf{x}_t + \frac{t}{t-s}\int_t^s \mathbf{v}(\mathbf{x}_u, u)\, \mathrm{d}u \right) \right\|^2 && (9) \\
&= \left( \frac{t-s}{t} \right)^2 \left\| (\mathbf{x}_t - t\mathbf{h}_{\boldsymbol{\theta}}(\mathbf{x}_t, t, s)) - \left( \mathbf{x}_t + \frac{t}{t-s}\int_t^s \mathbf{v}(\mathbf{x}_u, u)\, \mathrm{d}u \right) \right\|^2 \\
&= \left( \frac{t-s}{t} \right)^2 \left\| (\mathbf{x}_t - t\mathbf{h}_{\boldsymbol{\theta}}(\mathbf{x}_t, t, s)) - \left( \mathbf{x}_t + \frac{t}{t-s}\int_t^s \mathbf{v}(\mathbf{x}_u, u)\, \mathrm{d}u \right) \right\|^2 \\
&= (t-s)^2 \left\| \mathbf{h}_{\boldsymbol{\theta}}(\mathbf{x}_t, t, s) - \left( \frac{1}{s-t}\int_t^s \mathbf{v}(\mathbf{x}_u, u)\, \mathrm{d}u \right) \right\|^2 && (10)
\end{aligned}
$$

Equations (9) and (10) show that the two parameterizations are tightly connected. In particular,

$$
\frac{1}{t^2}\left\| \mathbf{g}_{\boldsymbol{\theta}}(\mathbf{x}_t, t, s) - \left( \mathbf{x}_t + \frac{t}{t-s}\int_t^s \mathbf{v}(\mathbf{x}_u, u)\, \mathrm{d}u \right) \right\|^2 = \left\| \mathbf{h}_{\boldsymbol{\theta}}(\mathbf{x}_t, t, s) - \left( \frac{1}{s-t}\int_t^s \mathbf{v}(\mathbf{x}_u, u)\, \mathrm{d}u \right) \right\|^2 .
$$
$$(11)$$

Hence, the CTM and MF training losses are mathematically related, differing only by a multiplicative constant. Moreover, in both cases setting $s = 0$ recovers the CM scenario, where each state is mapped directly to the clean data. Based on this observation, we will focus our theoretical analysis on $\boldsymbol{\Psi}_{s \to 0}$ mostly (Appendix F), noting that the same arguments extend naturally to the general $\boldsymbol{\Psi}_{s \to t}$, including the MF case.

## B    EXPERIMENTAL DETAILS

### B.1    SUMMARY OF THE CMT PIPELINE AND ALGORITHMS

---

**Algorithm 1** CMT Pipeline for Fast Flow Map Learning (CM or MF)

---

**Input:**
  flow map type `flow_map_type` $\in \{\text{CM}, \text{MF}\}$, time grid $0 = t_0 < t_1 < \cdots < t_M = T$,
  pre-trained teacher sampler weights $\phi$, student flow map weights $\theta$,
  mid-training learning rate $\eta_{\text{mid}}$, post-training learning rate $\eta_{\text{post}}$.

**Stage 1: Pre-training (teacher sampler).**
**if** teacher sampler not already available **then**
    **if** `flow_map_type` $=$ CM **then**
        Train diffusion model $\mathbf{D}_\phi$ with a standard EDM-style objective.
        Construct PF-ODE sampler (e.g. DPM-Solver++) from $\mathbf{D}_\phi$.
    **else if** `flow_map_type` $=$ MF **then**
        Train a flow-matching model $\mathbf{v}_\phi$ or a smaller MF model $\mathbf{h}_\phi^{\text{teacher}}$ in the usual way.
        Construct a deterministic sampler (e.g., a Heun solver with the PF-ODE determined by
$\mathbf{v}_\phi$), or an MF multi-step solver based on $\mathbf{h}_\phi^{\text{teacher}}$.
    **end if**
**end if**
Freeze $\phi$ and treat the resulting sampler as the teacher sampler `TeacherSampler`$(\phi; \cdot)$.

**Stage 2: CMT's Mid-Training.**
Initialize $\theta$ (optionally $\theta \leftarrow \phi$ when architectures match).
**repeat**
    Sample prior noise $\mathbf{x}_T \sim p_{\text{prior}}$.
    Generate teacher reference trajectory
        $\{\hat{\mathbf{x}}_{t_i}\}_{i=0}^M \leftarrow$ `TeacherSampler`$(\phi; \mathbf{x}_T)$.
    **if** `flow_map_type` $=$ CM **then**                    ▷ Special flow map $\mathbf{\Psi}_{t \to 0}$
        Compute CMT loss as in Equation (7)
            $\mathcal{L}_{\text{CMT-CM}}(\theta) = \sum_{i=0}^M d\big(\mathbf{f}_\theta(\hat{\mathbf{x}}_{t_i}, t_i), \hat{\mathbf{x}}_{t_0}\big).$
    **else if** `flow_map_type` $=$ MF **then**                ▷ General flow map $\mathbf{\Psi}_{t \to s}$
        Compute CMT loss as in Equation (8)
            $\mathcal{L}_{\text{CMT-MF}}(\theta) = \sum_{i>j} \big\| \mathbf{h}_\theta(\hat{\mathbf{x}}_{t_i}, t_i, t_j) - \frac{\hat{\mathbf{x}}_{t_i} - \hat{\mathbf{x}}_{t_j}}{t_i - t_j} \big\|_2^2.$
    **end if**
    Update CMT parameters
        $\theta \leftarrow \theta - \eta_{\text{mid}} \nabla_\theta \mathcal{L}_{\text{CMT-CM}}(\theta)$; or $\theta \leftarrow \theta - \eta_{\text{mid}} \nabla_\theta \mathcal{L}_{\text{CMT-MF}}(\theta)$.
**until** CMT converges (trajectory-aligned initializer $\theta$ obtained)

**Stage 3: Post-Training of Flow Map Model.**

Use the converged CMT mid-trained weights $\theta$ as the initialization of the flow map model.

**if** `flow_map_type` $=$ CM **then**                ▷ CM-style post-training for $\mathbf{\Psi}_{t \to 0}$, e.g. ECT/ECD
    **repeat**
        Sample training batch (data or noise) and times $t$.
        Compute CM post-training loss $\mathcal{L}_{\text{CM}}(\theta)$ as in Equation (3).
        Update $\theta \leftarrow \theta - \eta_{\text{post}} \nabla_\theta \mathcal{L}_{\text{CM}}(\theta)$.
    **until** post-training converges
**else if** `flow_map_type` $=$ MF **then**                        ▷ MF post-training for general $\mathbf{\Psi}_{t \to s}$
    **repeat**
        Sample training batch and pairs $(t, s)$ with $t > s$.
        Compute MF post-training loss $\mathcal{L}_{\text{MF}}(\theta)$ as in Equation (5).
        Update $\theta \leftarrow \theta - \eta_{\text{post}} \nabla_\theta \mathcal{L}_{\text{MF}}(\theta)$.
    **until** post-training converges
**end if**

**Output:** Learned few-step flow map $\mathbf{\Psi}_{t \to 0}$ (CM) or $\mathbf{\Psi}_{t \to s}$ (MF) with parameters $\theta$.

---

## B.2 CIFAR-10, AFHQV2, AND FFHQ

We use the variance-preserving (VP) formulation and DDPM++ model structure in Score-SDE (Song et al., 2021), which is also adopted in the teacher EDM diffusion model (Karras et al., 2022).

For the CMT mid-training stage, we utilize a third-order DPM-solver++ (Lu et al., 2025) with 16 NFEs to generate the ODE trajectory, achieving an FID of 2.14/2.25/2.99 compared to FIDs of 1.97/1.96/2.39 on CIFAR-10/AFHQv2/FFHQ, respectively, under an abundant 79 NFEs. The good FID under just 16 steps ensures the sample quality while making CMT fast since the ODE-solver across steps cannot be parallelized without additional care Shih et al. (2023). We use the same batch size of 128, 0.2 dropout rate, and RAdam optimizer (Liu et al., 2020) as the ECT stage later. We almost keep the same hyperparameters as the latter ECT, but make the following changes. We choose a 2e-4 learning rate for mid-training, which linearly decays to zero until the end of optimization. The EMA $\beta = 0.999$ since CMT is stable to ensure faster convergence. The loss metric for CMT is LPIPS (Zhang et al., 2018), and we use the simplest unit weighting.

For the ECT stage, we adopt the same hyperparameters as the original ECT setting (Geng et al., 2025b) on CIFAR-10 but keep the $\Delta t$ fixed to 1/4096, 1/1024, and 1/512 on CIFAR-10, AFHQv2, and FFHQ, respectively. We use the same 1e-4 learning rate but decay it linearly to zero until the end of optimization. This simplifies the complicated $\Delta t$ annealing trick in ECT. The choice of $\Delta t$ in our setting is quite straightforward. We search for the smallest $\Delta t$ that will not trigger a loss spike during the first several iterations.

### B.3 IMAGENET 64×64

#### B.3.1 EXPERIMENTAL DETAILS

We use the EDM2-XL (Karras et al., 2024b) model setting.

For the CMT mid-training stage, we use a third-order DPM-solver++ (Lu et al., 2025) with 16 NFEs to generate the ODE trajectory. We do not use classifier-free guidance (CFG) to accelerate the trajectory generation. Our 16-NFE FID is 1.56 compared with the EDM2's best 1.33 under 63 NFEs, ensuring a good teacher for mid-training. We use the same hyperparameters, including batch size, dropout rate, Adam optimizer (Kingma & Ba, 2015), etc., as the ECT stage later, but make the following modifications. We choose a 7e-4 learning rate for mid-training, which linearly decays to zero until the end of optimization. The EMA $\beta = 0.9999$. The loss metric for CMT is LPIPS (Zhang et al., 2018), and we use the simplest unit weighting. We train for 6.4 Mimgs.

For the ECT stage, we primarily adopt the same hyperparameters as the original ECT setting (Geng et al., 2025b) on ImageNet 64×64 with the XL size and a batch size of 128, while simplifying the following hyperparameters. We keep the $\Delta t$ fixed to 1/512 instead of the original ECT's complex annealing trick. We use an initial learning rate of 1e-4 decaying linearly to zero at the end of optimization, which is simpler than the quadratic decay in the original ECT and EDM2. Furthermore, we just use a simple vanilla EMA with $\beta = 0.9999$ instead of the power function post-hoc EMA in ECT and EDM2. This simplifies various tricks in ECT. We conduct ECT for another 6.4 Mimgs.

For ECD with the Auto-Guidance (Karras et al., 2024a) augmented EDM2, we start from the mid-trained CMT checkpoint. We keep the $\Delta t$ fixed to 1/256 and the learning rate fixed to 1e-4. We conduct ECD for 12.8 Mimgs.

#### B.3.2 COST DETAILS

We compare training cost and data budget for ECT, ECD, sCT, AYF, and CMT here.

ECT and CMT require the standard EDM2 diffusion pre-training, with a batch size of 2048 and a total of 327680 iterations. Hence, the total training data budget is $671088640 \approx 671.1$ Mimgs. AYF uses a batch size of 2048 and a total of 524288 iterations, leading to about 1073.7 Mimgs diffusion pre-training cost. sCT requires a TrigFlow diffusion pre-training, with a batch size of 2048 and a total of 540000 iterations. Hence, the total training data budget is $1105920000 \approx 1105.9$ Mimgs.

The total pre-training, mid-training, and post-training data budget costs of all methods are summarized in Table 6. Our 98% (CMT w/ ECT over sCT) and 81.25% (CMT w/ ECD over AYF) training

data budget reduction includes both mid-training and post-training. In other words, we compare CMT's mid-training + post-training total budget with other methods' post-training budget. sCT's TrigFlow-based EDM2 is just reproducing vanilla EDM2, and the teacher diffusion quality is almost the same. And we are focusing on flow map learning, but not the diffusion model pre-training part. Thus, for EDM2-related experiments, including ImageNet 64×64 and 512×512, we focus on comparing the mid-training + post-training costs.

| Method | Pre-Training | Mid-Training | Post-Training | FID ($\downarrow$) |
|---|---|---|---|---|
| ECT (Geng et al., 2025b) | 671.1 | 0 | 102.4 | 2.49 / 1.67 |
| ECD (Geng et al., 2025b) | 671.1 | 0 | 102.4 | 2.24 / 1.50 |
| sCT (Lu & Song, 2025) | 1105.9 | 0 | 819.2 | 2.04 / 1.48 |
| AYF (Sabour et al., 2025) | 1073.7 | 0 | 102.4 | 2.98 / 1.25 |
| CMT (w/ ECT) *(Ours)* | 671.1 | 6.4 | 6.4 | 2.02 / 1.48 |
| CMT (w/ ECD) *(Ours)* | 671.1 | 6.4 | 12.8 | 1.78 / 1.32 |

Table 6: ImageNet 64×64: Pre-, mid-, and post-training data costs (in Mimgs).

Furthermore, we summarize CMT's time reduction and speedup compared to the baselines in Table 7, where we compare the A100 (80G) GPU time for training ECT, ECD, sCT, and CMT. We compare with ECT and ECD since they are the post-training methods in our CMT. Meanwhile, we compare with the competitive sCT. We conduct experiments to measure per-iteration time for every method, and compute the total training time as total iterations × per-iteration time.

| Method | Baseline | Baseline (hrs) | Ours (hrs) | Reduction | Speedup |
|---|---|---|---|---|---|
| CMT w/ ECT | ECT | 1280 | 180 | 85.9% | 7.11× |
| CMT w/ ECT | sCT | 13312 | 180 | 98.6% | 73.96× |
| CMT w/ ECD | ECD | 1664 | 308 | 81.5% | 5.40× |
| CMT w/ ECD | sCD | 16000 | 308 | 98.1% | 51.95× |

Table 7: ImageNet 64×64: Comparison of training time reduction and speedup for four cases: (1) CMT w/ ECT & ECT; (2) CMT w/ ECT & sCT; (3) CMT w/ ECD & ECD; and (4) CMT w/ ECD & sCD.

We emphasize that the reported best performances of ECT and sCT are achieved by initializing their flow map models from pre-trained diffusion models, which have similar generation quality as the teacher model that CMT uses for trajectory creation. Hence, in our comparison, the teacher model's training cost is excluded. Overall, we observe that flow map training with CMT (including both mid- and post-training stages) achieves an 80%–98% reduction in training time compared to training a flow map model alone.

### B.4 IMAGENET 256×256

We follow SiT (Ma et al., 2024) and Mean Flow (Geng et al., 2025a) for this setting.

Regarding MF from scratch, we directly use the default setting in the original MF paper Geng et al. (2025a) and follow the PyTorch Paszke et al. (2019) implementation Zhu (2025). The efficient forward-mode JVP Shi et al. (2024) is used to maximize MF training efficiency.

Regarding MF initialized by SiT, we follow Zhu (2025) for a two-stage post-training, starting with MF without CFG for stability and then switching to the default MF training with CFG. However, we found that this approach still diverges at some point during optimization and cannot be mitigated by changing the random seed and restarting. Furthermore, if one directly tunes the MF initialized by SiT with CFG, then the optimization directly diverges, and the gradient explodes at the very beginning. These observations all point to the instability of SiT initialized MF, i.e., the diffusion initialization.

Regarding CMT, the post-training MF hyperparameter is kept the same as vanilla MF except that we reduce the batch size from 256 to 64. Since CMT can stabilize training by providing a better initialization, there is no need to use a large batch size to stabilize training as in MF from scratch. For pre-training a tiny and efficient MF-B/4, we use the MF-B/4 training hyperparameters in the

original MF paper Geng et al. (2025a) but change the CFG-related hyperparameters the same as the post-training. For mid-training, we generate the reference ODE trajectory with the pre-trained MF-B/4 with eight uniform steps between 0 and 1. We also use a constant learning rate of 1e-4 and do not use any weighting trick, and use squared $\ell_2$ loss. We use four random samples to generate trajectories, and each sample provides 28 pairs of $(\hat{\mathbf{x}}_{t_i}, \hat{\mathbf{x}}_{t_j})$. With this batch size, we conduct mid-training for 200k iterations. We found that the key is to use the same CFG scale for all the stages. MF with various CFG scales during training has a different ODE trajectory. Therefore, it is imperative to match the pre-, mid-, and post-training stage CFG scale. Otherwise, one would obtain inferior results due to the trajectory mismatch during different stages.

## B.5   IMAGENET 512×512

### B.5.1   EXPERIMENTAL DETAILS

**ELatentLPIPS**. We follow the standard approach to train a VGG (Simonyan & Zisserman, 2015) for ELatentLPIPS. We train VGG for 100 epochs with SGD. The initial learning rate is 0.1 and decays at the 30th, 60th, and 90th epochs with a 0.1 decay rate. The batch size is 256. The resulting VGG achieves 95% top1 accuracy on the train set and 64% validation top1 accuracy. Then, this VGG should have been fine-tuned on the BAPPS (Zhang et al., 2018) data to learn human perception. However, we do not take this step to ensure a fair comparison with other baselines, i.e., we keep the training data as ImageNet only and do not rely on additional data that other CMs do not.

**CMT**. We mainly transfer our ImageNet 64×64 hyperparameters since they all follow the EDM2 setting. We highlight the difference below. We do not use dropout to stabilize the training. We use the XXL model size.

**CMT Mid-Training**.  We use a third-order DPM-solver++ (Lu et al., 2025) with 16 NFEs to generate the ODE trajectory. We do not use classifier-free guidance (CFG) to accelerate the trajectory generation. We choose a 2e-4 learning rate for mid-training, which linearly decays to zero until the end of optimization. The EMA $\beta = 0.999$. The loss metric for CMT is ELatentLPIPS (Kang et al., 2024), and we use the simplest unit weighting. We train for 12.8 Mimgs with a batch size of 128.

**CMT Post-Training's ECD**. We use ECD as post-training to distill the EDM2 Auto-Guidance (Karras et al., 2024a) model of Size XXL. We keep the $\Delta t$ fixed to 1/1024 instead of the original ECD's complex annealing trick. We use a constant learning rate of 1e-4. The batch size is 128. The total training budget is 12.8 Mimgs.

### B.5.2   COST DETAILS

Similar to the ImageNet 64×64 case, we compare various methods' training data budget cost and training time. Table 8 shows the training data budget, where we achieve 93% lower cost than the sCD and 71% lower cost than the AYF. We report H100 GPU training time in Table 9, where we used a better GPU for this higher-dimensional generation task with a larger model. Table 9 demonstrates that CMT (including both mid- and post-training stages) achieves an 75%–92.8% reduction in training time compared to training a flow map model alone.

| Method | Pre-Training | Mid-Training | Post-Training | FID ($\downarrow$) |
|---|---|---|---|---|
| ECT (Geng et al., 2025b) | 939.5 | 0 | 204.8 | 9.98 / 6.28 |
| ECD (Geng et al., 2025b) | 939.5 | 0 | 409.6 | 8.47 / 3.38 |
| sCT (Lu & Song, 2025) | 770.0 | 0 | 204.8 | 4.29 / 3.76 |
| sCD (Lu & Song, 2025) | 770.0 | 0 | 409.6 | 2.28 / 1.88 |
| AYF (Sabour et al., 2025) | 2147.5 | 0 | 102.4 | 3.32 / 1.87 |
| CMT (w/ ECD) *(Ours)* | 939.5 | 12.8 | 16 | 3.38 / 1.84 |

Table 8: ImageNet 512×512: Pre-, mid-, and post-training data costs (in Mimgs).

| Method | Baseline | Baseline (hrs) | Ours (hrs) | Reduction | Speedup |
|---|---|---|---|---|---|
| CMT w/ ECD | ECT | 1611.18 | 403.63 | 75.0% | 3.99× |
| CMT w/ ECD | sCT | 2339.88 | 403.63 | 82.7% | 5.80× |
| CMT w/ ECD | ECD | 4643.99 | 403.63 | 91.3% | 11.51× |
| CMT w/ ECD | sCD | 5591.74 | 403.63 | 92.8% | 13.85× |

Table 9: ImageNet 512×512: Comparison of training time reduction and speedup for four cases: (1) CMT w/ ECD & ECT; (2) CMT w/ ECD & sCT; (3) CMT w/ ECD & ECD; and (4) CMT w/ ECD & sCD.

### B.6 MS-COCO TEXT-TO-IMAGE (T2I)

The SiT teacher model is trained with REPA (Yu et al., 2025) for 50k iterations, achieving a 50-step FID of 4.73 using a second-order Heun ODE solver, and the CFG scale is 2. All other models are also trained with REPA. The SiT teacher additionally serves as the pre-training stage.

Regarding MF from scratch, we mainly reuse the default setting in our ImageNet 256×256 MF experiment. But we set the MF's effective CFG scale to 2 and the CFG trigger interval as $[0, 1]$ since the SiT teacher uses the same CFG hyperparameters. We specifically set $\kappa = 0.5$ and $\omega = 1$. Regarding MF initialized by SiT and CMT with MF, we use the same MF training configuration for post-training.

For mid-training in CMT, we generate the reference ODE trajectory with the pre-trained SiT with an 8-step Heun solver. We also use a constant learning rate of 1e-4 and do not use any weighting trick, and use squared $\ell_2$ loss. We use four random samples to generate trajectories, and each sample provides 28 pairs of $(\hat{\mathbf{x}}_{t_i}, \hat{\mathbf{x}}_{t_j})$, inducing a total $28 \times 4 = 112$ batch size per iteration.

## C TRAINING SPEED AND MEMORY COST

### C.1 EMPIRICAL RUNTIME COMPARISON

We report the running speed of CMT, CT, and CD. For ImageNet 512×512, we used a single H100 GPU, while for other datasets, we tested on a single A100 GPU with 80 GB (81920 MiB) of memory. We chose the simple ECT and ECD as representatives for comparison. CMs with additional tricks may incur larger costs. We adopt a second-order Heun or a first-order Euler solver in CD. The training hyperparameters, especially the batch size, are kept the same as in the main results. For easy speed comparison, we normalize our method's speed to 1 unit, where a larger number means a lower speed.

| Dataset | Batch | CMT | CT (↓) | CD-Euler (↓) | CD-Heun (↓) |
|---|---|---|---|---|---|
| CIFAR-10 | 128 | 1 | 0.79 | 0.98 | 1.17 |
| AFHQv2 & FFHQ | 128 | 1 | 0.85 | 1.05 | 1.25 |
| ImageNet 64×64 | 32 | 1 | 0.80 | 0.92 | 1.04 |
| ImageNet 512×512 | 16 | 1 | 0.68 | 0.83 | 0.98 |

The memory costs of all methods are similar, as they all involve one backpropagation step. But CT has the smallest memory cost, CMT is the second, and CD has the largest memory cost. This is because CT does not require any additional teacher network, while CMT requires one unguided teacher. Lastly, CD requires two additional types of nets: guided and unguided teachers.

### C.2 ANALYSIS OF COMPUTATIONAL COST IN CM

**Empirical Runtime Comparison with CMT, CD, and CT.** The important factor in CMT's wall-clock time is the number of ODE solver steps (NFEs): steps along the trajectory are sequential, so higher NFE directly increases wall time. While solver-parallelization may ease this bottleneck (Shih et al., 2023), we target fast training under prevailing practice by operating in the low-NFE regime.

Throughout, we fix NFE $= 16$, which we found to be a sweet spot: it provides sufficiently accurate supervision while keeping CMT only slightly slower than CT. We use the third-order multistep DPM-Solver++ (Lu et al., 2025) for CM-style teachers, as it's stable and effective at low NFEs. The multistep scheme also reuses previously computed states $\hat{\mathbf{x}}_{t_k}$ to construct later states $\hat{\mathbf{x}}_{t_i}$, further reducing overhead. Two error sources matter: (i) the *distillation fit* error due to nonzero training loss and (ii) the *teacher discretization* error from small NFE; empirically, (i) dominates.

With NFE $= 16$, DPM-Solver++ attains FID typically within $0.2$ of the best large-NFE setting across all datasets, indicating diminishing returns beyond 16. Consequently, CMT with 16 NFEs preserves speed while maintaining competitive quality. Concretely, CMT attains training efficiency comparable to CD (which requires teacher inference) and continuous CT or MF requiring student JVP. Further, CMT's cost is also close to that of discrete CT (Lu & Song, 2025). Per iteration, CMT is only 15%–25% slower than discrete CT (Geng et al., 2025b), where we used Easy CT and Easy CD (Geng et al., 2025b)'s framework for time evaluation. However, CMT converges faster in far fewer iterations than these alternatives in the entire training loop; thus, the wall-clock runtime is lower. This advantage is clear on ImageNet $64 \times 64$ and $512 \times 512$, where CMT outperforms sCT/sCD while reducing training data and GPU cost by 93%–98%. We also cut 50% of the training GPU time compared with vanilla MF while outperforming in FID.

Overall, because it provides a stronger proxy of the flow-map trajectory during mid-training, it substantially accelerates the subsequent flow-map post-training (e.g., CT).

If teacher trajectories are pre-generated, training with CMT reduces to a single backpropagated student evaluation per pair, which is the fastest regime and is used by prior distillation work (Zheng et al., 2023). However, pre-generation requires extra preparation and storage; to keep the setup simple and comparable, all our experiments run the ODE solver on-the-fly during training.

**Theoretical NFEs Comparison with CMT , CD, and CT.** We compare CMT, CD, and CT by teacher function evaluations (NFEs), student forwards, and student backpropagations. Costs are normalized per training pair, where a pair is one input–target term in the loss.

In CMT, each teacher trajectory $\{\hat{\mathbf{x}}_{t_i}\}_{i=0}^M$ yields $M$ pairs $(\hat{\mathbf{x}}_{t_i}, t_i) \mapsto \hat{\mathbf{x}}_0$ for $i = 1, \ldots, M$. Let $M \geq k$ be the number of steps from $t_M$ to $t_0$, $k$ the multistep order, and $s$ the NFE cost per bootstrap step used to initialize the first $k-1$ history points (e.g., $s=1$ for Euler, $s=2$ for Heun). An explicit $k$-step solver then incurs one new teacher evaluation per step thereafter. Hence

$$\text{NFEs}_{\text{traj}} = s(k-1) + \big(M - (k-1)\big) = M + (s-1)(k-1),$$

and the *per-pair* teacher cost for CMT is

$$\text{Teacher NFEs per pair (CMT)} = 1 + \frac{(s-1)(k-1)}{M}.$$

In CD and CT, each pair corresponds to a single sampled time $t$. If $q$ denotes teacher NFEs for the one-step teacher update used inside CD (e.g., $q=1$ Euler, $q=2$ Heun), then CD has Teacher NFEs per pair $= q$; CT has none.

The above yields the following per-pair cost summary:

$$\textbf{CMT:} \quad \text{Teacher NFEs} = 1 + \frac{(s-1)(k-1)}{M}, \quad \text{Student} = 1\,\text{fwd} + 1\,\text{bwd},$$

$$\textbf{CD:} \quad \text{Teacher NFEs} = q, \quad\quad\quad\quad\quad\quad\quad \text{Student} = 1\,\text{fwd} + 1\,\text{bwd},$$

$$\textbf{CT:} \quad \text{Teacher NFEs} = 0, \quad\quad\quad\quad\quad\quad\quad\; \text{Student} = 2\,\text{fwd} + 1\,\text{bwd}.$$

We now instantiate the parameters to match our experimental setup. With $M = 16$ and $k \in \{2, 3\}$,

$$\text{CMT teacher NFEs per pair} = \begin{cases} 1, & s=1 \text{ (Euler warm-up)}, \\ 1 + \frac{k-1}{16} \in [1.06, 1.12], & s=2 \text{ (Heun warm-up)}. \end{cases}$$

Thus, relative to CD: CMT matches CD when $q=1$ and $s=1$; it is only $+6\% \sim 12\%$ higher when $q=1$, $s=2$, $k \in \{2, 3\}$; and it is cheaper than CD when $q=2$ for these $M, k$. These accounting predictions align closely with the empirical measurements reported earlier in this subsection, further supporting the efficiency of CMT .

In short, we analyze the cost per input–target pair. In CMT, one teacher-generated trajectory yields many pairs by matching intermediate states to the clean target, whereas CD and CT generate each pair independently from a sampled time. Per pair, CMT needs one teacher call plus one student forward and one backpropagation; CD needs one teacher call plus the same student cost; CT needs no teacher call but two student forwards and one backpropagation. Hence, CMT is roughly as costly as CD and only slightly slower than CT, consistent with our empirical findings (Appendix C.1). Meanwhile, CMT achieves near–unit teacher cost per pair while using a single student forward, making it a lightweight and effective choice for the mid-training stage.

**Theoretical NFEs Comparison with CMT and Slow CMT Variant.** Mathematically, both the CMT loss $\mathcal{L}_{\text{CMT-CM}}$ in Equation (7) and the Slow CMT variant $\mathcal{L}_{\text{var}}^{(1)}$ are discrete-time approximations to the same oracle loss $\mathcal{L}_{\text{oracle-CM}}$ in Equation (5). However, they differ crucially in how the teacher-generated supervision targets are constructed. In CMT, we first draw $\mathbf{x}_T \sim p_{\text{prior}}$, run the teacher ODE solver once from $T$ down to $0$ to obtain a full trajectory $\{\hat{\mathbf{x}}_{t_i}\}_{i=0}^{M}$, and then use the single endpoint $\hat{\mathbf{x}}_{t_0}$ as a shared target for all intermediate states $(\hat{\mathbf{x}}_{t_i}, t_i)$ on that trajectory. Thus, one teacher solve yields $M$ supervised pairs $(\hat{\mathbf{x}}_{t_i}, t_i) \mapsto \hat{\mathbf{x}}_{t_0}$. In contrast, in Slow CMT (based on $\mathcal{L}_{\text{var}}^{(1)}$), we sample $(\mathbf{x}_t, t)$ and, for each such pair, independently run the teacher solver from $\mathbf{x}_t$ to $t = 0$ to obtain $\hat{\mathbf{x}}_0(\mathbf{x}_t)$. Each supervised pair $(\mathbf{x}_t, t) \mapsto \hat{\mathbf{x}}_0(\mathbf{x}_t)$ therefore requires its own teacher integration. The two objectives coincide in the continuous-time limit, but their discrete implementations use teacher trajectories in fundamentally different ways.

Because of this difference, the mid-training cost of Slow CMT is significantly higher for a fixed number of supervised pairs. Let $M$ denote the number of discretization steps on $[0, T]$, and let $\text{NFEs}_{\text{traj}}$ be the teacher function evaluations required for one trajectory. In CMT, a single teacher trajectory is amortized over $M$ pairs, so the teacher cost per training pair is $\text{NFEs}_{\text{traj}}/M = \mathcal{O}(1)$. In Slow CMT, each training pair requires a fresh teacher solve, so the teacher cost per pair is $\mathcal{O}(\text{NFEs}_{\text{traj}})$. Consequently, to train on the same number of supervised pairs, Slow CMT would require roughly a factor of $M$ more teacher evaluations (and thus GPU time) than CMT; conversely, under a fixed teacher-compute budget, Slow CMT can only afford about $1/M$ as many pairs. In our experimental configuration, this leads to an overhead of roughly $3\times$ in wall-clock GPU time, which is what we refer to in the paper when stating that the Slow CMT mid-training stage is about three times more expensive. We did not perform an exhaustive sweep that precisely equalizes GPU hours between the two variants during the review period, but the existing experiments already show that CMT achieves better FID at substantially lower teacher cost, which is exactly the mid-training efficiency advantage we aim to demonstrate.

## D   DISCUSSION ON CMT LOSS METRIC

**Loss Metric: LPIPS**. The metric is crucial to CMT's performance. The LPIPS metric (Zhang et al., 2018) measuring perceptual similarity is known to align more closely with human vision. Optimizing LPIPS loss helps models generate images that are perceptually similar to the original because it penalizes differences in a feature space that aligns better with human visual processing, rather than $L_2$ loss in pixel space. The features are derived from VGG (Simonyan & Zisserman, 2015), which is initially pretrained for ImageNet classification in torchvision (Paszke et al., 2019), and then fine-tuned on human perceptual judgments (BAPPS dataset) to better reflect human judgments of image similarity. CMT generates high-quality supervision signals using high-order multistep ODE solvers, providing accurate and stable labels. To encourage the student model to closely match the teacher's output, we employ the LPIPS loss. Moreover, since CMT provides fixed and stable labels, the training process becomes inherently robust, obviating the need for additional robust loss functions such as the Huber loss used in iCT (Song & Dhariwal, 2024). This also eliminates the burden of tuning extra hyperparameters associated with such losses. While minimizing the $L_2$ loss yields outputs with low pixel-wise error relative to the teacher's predictions, it often results in blurry images that fail to capture perceptual fidelity. This is because $L_2$ loss penalizes deviations uniformly across all pixels, disregarding the spatial and structural cues that are critical for human visual perception.

**Why ECT/iCT Uses Huber/$L_1$ Loss?** The optimization objectives and training dynamics of ECT/iCT and CMT differ fundamentally. CMT leverages high-quality, fixed teacher labels generated via accurate numerical solvers, providing a reliable supervision signal throughout training. In contrast,

ECT and iCT rely on self-generated guidance, where the model learns from its own predictions. In such self-training settings, the use of perceptual losses like LPIPS may introduce additional bias, as the supervision signal is inherently noisy and evolving.

**Latent LPIPS Loss**. LPIPS operates exclusively in pixel space, whereas latent CM faces the challenge of lacking a refined metric and must resort to traditional $L_2$. ELatentLPIPS (Kang et al., 2024) trains an LPIPS metric in the autoencoder-dependent latent space. The idea is still to first train a VGG (Simonyan & Zisserman, 2015) and then fine-tune it on BAPPS (Zhang et al., 2018) in the latent space. For our latent space experiments, we train a VGG following their setup on the ImageNet datasets, but do not fine-tune on the additional BAPPS dataset to ensure a fair comparison with other methods. In other words, CMT do not resort to additional datasets while using the same train set as other baselines.

**CMT with MF uses Squared $\ell_2$ Loss**. This is because MF is a general flow map requiring mapping between any time steps, but not just the initial time corresponding to the clean data. Hence, the label in CMT with MF can be noisy data within the trajectory, rendering the LPIPS loss inapplicable. Hence, we resort to the common squared $\ell_2$ loss.

## E    SAMPLES GENERATED BY CMT ON IMAGENET $512\times512$

To illustrate the visual quality of CMT, we show two step samples generated by CMT(with ECD) trained on $512\times512$ in Figure 3.

## F    THEORETICAL ANALYSIS OF CMT

### F.1    ORACLE LOSS AND CMT'S APPROXIMATION

**The Minimizer of Equation (2).**    We first show that the optimizer of Equation (2) recovers the oracle flow map $\Psi_{t\to0}$.

**Proposition F.1** (Oracle CM minimizer). *Assume: (i) $d : \mathbb{R}^D \times \mathbb{R}^D \to [0, \infty)$ satisfies $d(\mathbf{y}, \mathbf{z}) \geq 0$ and $d(\mathbf{y}, \mathbf{z}) = 0$ iff $\mathbf{y} = \mathbf{z}$; (ii) $\mathbf{f}_{\boldsymbol{\theta}}(\mathbf{x}_t, t) \, \mathbb{E}\|\mathbf{f}_{\boldsymbol{\theta}}(\mathbf{x}_t, t)\|_2^2 < \infty$. Then any minimizer of*

$$\mathcal{L}_{\text{oracle-CM}}(\boldsymbol{\theta}) = \mathbb{E}_t \mathbb{E}_{\mathbf{x}_t \sim p_t} \Big[ w(t) d\big(\mathbf{f}_{\boldsymbol{\theta}}(\mathbf{x}_t, t), \Psi_{t\to0}(\mathbf{x}_t)\big) \Big]$$

*satisfies*

$$\mathbf{f}_{\boldsymbol{\theta}^*}(\mathbf{x}_t, t) = \Psi_{t\to0}(\mathbf{x}_t) \quad \text{for a.e. } \mathbf{x}_t \sim p_t \text{ and } t.$$

*If, in addition, $d(\cdot, \mathbf{z})$ is strictly convex for each fixed $\mathbf{z}$, then this minimizer is unique (a.e.).*

*Proof.* Let $\text{Unif}[0, T]$ denote the time distribution of $t$. The integrand $w(t) d\big(\mathbf{f}_{\boldsymbol{\theta}}(\mathbf{x}_t, t), \Psi_{t\to0}(\mathbf{x}_t)\big)$ is a nonnegative measurable function. Hence

$$\mathcal{L}_{\text{oracle-CM}}(\boldsymbol{\theta}) \geq 0 \quad \text{for all } \boldsymbol{\theta}.$$

Choosing $\mathbf{f}_{\boldsymbol{\theta}}(\mathbf{x}_t, t) \equiv \Psi_{t\to0}(\mathbf{x}_t)$ makes the integrand identically zero (since $d(\mathbf{z}, \mathbf{z}) = 0$), so the infimum of the objective is 0 and is attained by this choice. It remains to show that any other minimizer must agree with $\Psi_{t\to0}$ almost surely.

Suppose $\mathbf{f}_{\boldsymbol{\theta}^\star}$ is a minimizer and define the set

$$A := \{(t, \mathbf{x}_t) : \mathbf{f}_{\boldsymbol{\theta}^\star}(\mathbf{x}_t, t) \neq \Psi_{t\to0}(\mathbf{x}_t)\}.$$

On $A$ we have $d\big(\mathbf{f}_{\boldsymbol{\theta}^\star}(\mathbf{x}_t, t), \Psi_{t\to0}(\mathbf{x}_t)\big) > 0$ by assumption (ii). Since $w(t) > 0$ for $\text{Unif}[0, T]$-a.e. $t$, if $\text{Unif}[0, T] \times p_t(A) > 0$ then by Tonelli or Fubini theorem

$$\mathcal{L}_{\text{oracle-CM}}(\boldsymbol{\theta}^\star) = \mathbb{E}\big[w(t) d\big(\mathbf{f}_{\boldsymbol{\theta}^\star}(\mathbf{x}_t, t), \Psi_{t\to0}(\mathbf{x}_t)\big)\big] \geq \mathbb{E}\big[w(t) \mathbf{1}_{\text{CMT}}(t, \mathbf{x}_t) c\big] > 0$$

for some $c > 0$, contradicting minimality (the minimum value is 0). Therefore $\text{Unif}[0, T] \times p_t(A) = 0$, i.e., $\mathbf{f}_{\boldsymbol{\theta}^\star} = \Psi_{t\to0}$ holds $\text{Unif}[0, T] \times p_t$-a.e. If $d(\cdot, \mathbf{z})$ is strictly convex (e.g., squared $\ell_2$), pointwise equality (a.e.) is the only way to achieve the minimum, giving uniqueness (a.e.). $\square$

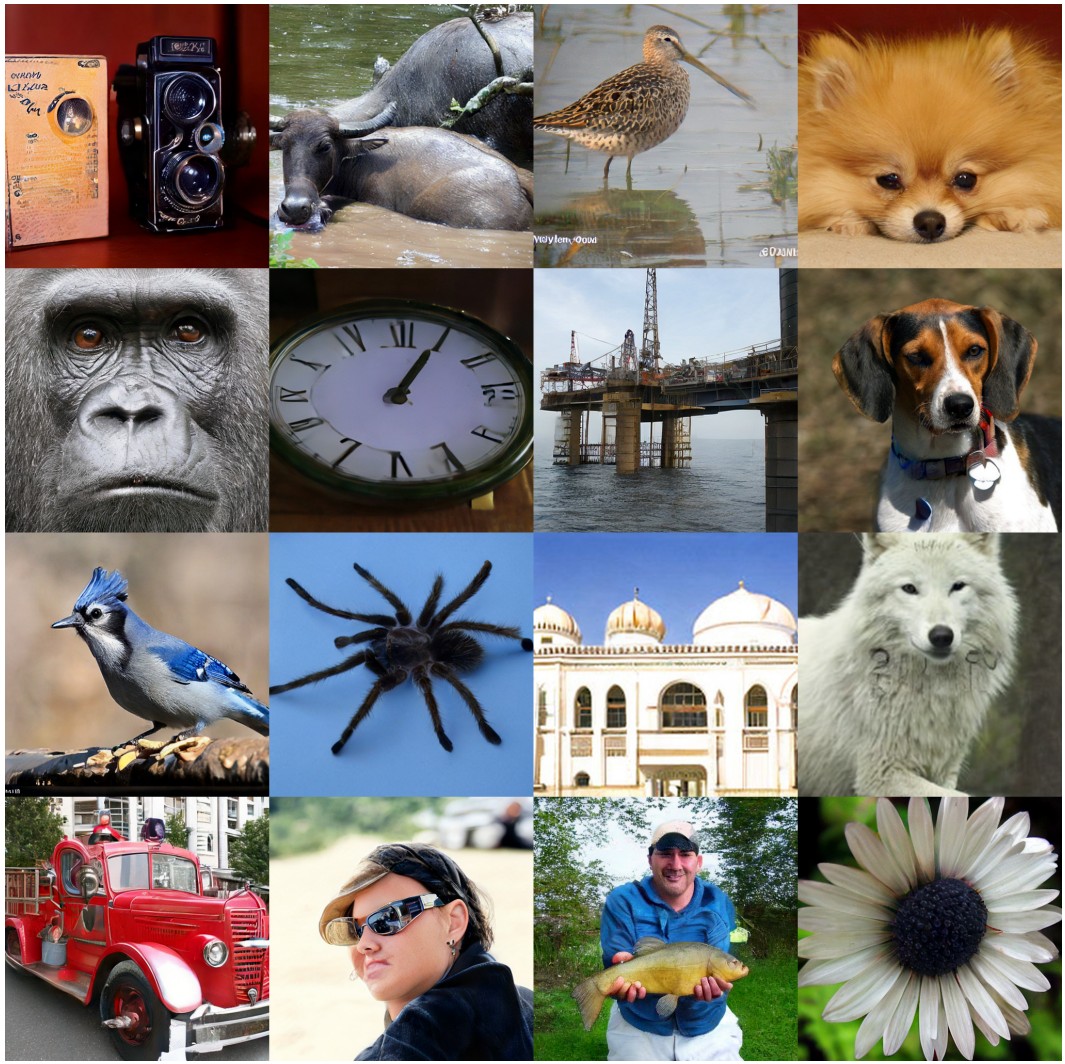

Figure 3: **Two-Step Generated Images by CMT.** Using the trained CMT (w/ ECD) on $512{\times}512$, we achieve the best two-step FID of 1.84, at 93% lower cost than previous sCD.

**CMT's Loss is Equivalent to Equation (2).** We now prove Theorem 3.1, which shows that the CMT objective is, up to a discrete-time approximation, equivalent to minimizing the oracle CM flow map loss in Equation (2). We assume that the terminal distribution $p_{\text{prior}}$ coincides with $p_T$. Under this assumption, we will show that the following result holds:

$$\mathcal{L}_{\text{oracle-CM}}(\boldsymbol{\theta}) = \mathbb{E}_t \mathbb{E}_{p_T(\mathbf{x}_T)} \left[ d\left( \mathbf{f}_{\boldsymbol{\theta}}(\boldsymbol{\Psi}_{T \to t}(\mathbf{x}_T), t), \boldsymbol{\Psi}_{T \to 0}(\mathbf{x}_T) \right) \right].$$

*Proof.* We can exploit the semi-group property of the solution map to express the intermediate distribution $p_t$ as:

$$p_t = \boldsymbol{\Psi}_{0 \to t} \sharp p_{\text{data}} = \boldsymbol{\Psi}_{T \to t} \sharp p_{\text{prior}} = \int \delta(\mathbf{x}_t - \boldsymbol{\Psi}_{T \to t}(\mathbf{x}_T)) p_T(\mathbf{x}_T) \mathrm{d}\mathbf{x}_T.$$

Using this as a change of variables in Equation (2), we obtain:

$$\begin{aligned}
\mathcal{L}_{\text{oracle-CM}}(\boldsymbol{\theta}) &= \mathbb{E}_t \mathbb{E}_{p_t(\mathbf{x}_t)} \left[ d\left( \mathbf{f}_{\boldsymbol{\theta}}(\boldsymbol{\Psi}_{T \to t}(\mathbf{x}_T), t), \boldsymbol{\Psi}_{T \to 0}(\mathbf{x}_T) \right) \right] \\
&= \mathbb{E}_t \int d\left( \mathbf{f}_{\boldsymbol{\theta}}(\boldsymbol{\Psi}_{T \to t}(\mathbf{x}_T), t), \boldsymbol{\Psi}_{T \to 0}(\mathbf{x}_T) \right) p_t(\mathbf{x}_t) \mathrm{d}\mathbf{x}_t \\
&= \mathbb{E}_t \int \int d\left( \mathbf{f}_{\boldsymbol{\theta}}(\boldsymbol{\Psi}_{T \to t}(\mathbf{x}_T), t), \boldsymbol{\Psi}_{T \to 0}(\mathbf{x}_T) \right) \delta(\mathbf{x}_t - \boldsymbol{\Psi}_{T \to t}(\mathbf{x}_T)) p_T(\mathbf{x}_T) \mathrm{d}\mathbf{x}_T \mathrm{d}\mathbf{x}_t \\
&= \mathbb{E}_t \int \int d\left( \mathbf{f}_{\boldsymbol{\theta}}(\boldsymbol{\Psi}_{T \to t}(\mathbf{x}_T), t), \boldsymbol{\Psi}_{T \to 0}(\mathbf{x}_T) \right) p_T(\mathbf{x}_T) \delta(\mathbf{x}_t - \boldsymbol{\Psi}_{T \to t}(\mathbf{x}_T)) \mathrm{d}\mathbf{x}_t \mathrm{d}\mathbf{x}_T \\
&= \mathbb{E}_t \int d\left( \mathbf{f}_{\boldsymbol{\theta}}(\boldsymbol{\Psi}_{T \to t}(\mathbf{x}_T), t), \boldsymbol{\Psi}_{T \to 0}(\mathbf{x}_T) \right) p_T(\mathbf{x}_T) \mathrm{d}\mathbf{x}_T \\
&= \mathbb{E}_t \mathbb{E}_{p_T(\mathbf{x}_T)} \left[ d\left( \mathbf{f}_{\boldsymbol{\theta}}(\boldsymbol{\Psi}_{T \to t}(\mathbf{x}_T), t), -\boldsymbol{\Psi}_{T \to 0}(\mathbf{x}_T) \right) \right].
\end{aligned}$$

$\square$

Therefore, the CMT loss approximates the oracle objective $\mathcal{L}_{\text{oracle}}$ by leveraging a pre-trained diffusion model to estimate the solution map $\boldsymbol{\Psi}_{T \to t}(\mathbf{x}_T)$. This allows for a tractable surrogate to the otherwise intractable oracle loss.

### F.2 Initialization Schemes for Flow Map Model Training

Let $\varepsilon > 0$. We investigate four initialization schemes for the post training stage of CM flow map $\boldsymbol{\Psi}_{t \to 0}$ learning.

**CMT.** There exists $\boldsymbol{\theta}_{\text{CMT}}$ such that

$$\mathcal{L}_{\text{CMT}}(\boldsymbol{\theta}_{\text{CMT}}) := \mathbb{E}_t \mathbb{E}_{\mathbf{x}_T \sim p_{\text{prior}}} \left\| \mathbf{f}_{\boldsymbol{\theta}_{\text{CMT}}}\left( \texttt{Solver}_{T \to t}(\mathbf{x}_T), t \right) - \texttt{Solver}_{T \to 0}(\mathbf{x}_T) \right\|_2^2 < \varepsilon,$$

where $\texttt{Solver}_{t \to u}(\mathbf{x}_t)$ denotes the result of running the ODE solver from $t$ back to $u$ using the drift of a pre trained diffusion model in the PF ODE.

**Diffusion Model (DM).** Let $\mathbf{D}_{\boldsymbol{\theta}}$ denote the clean prediction of a diffusion model. There exists $\boldsymbol{\theta}_{\text{DM}}$ such that

$$\mathcal{L}_{\text{DM}}(\boldsymbol{\theta}_{\text{DM}}) := \mathbb{E}_t \mathbb{E}_{\mathbf{x}_t \sim p_t} \left\| \mathbf{D}_{\boldsymbol{\theta}_{\text{DM}}}(\mathbf{x}_t, t) - \mathbf{x}_0 \right\|_2^2 < \varepsilon.$$

**General Consistency Distillation (gCD).** We define a general consistency distillation loss that employs a "soft label" for teacher supervision (Kim et al., 2024). Let $u \in [0, T]$ be fixed and let $\boldsymbol{\theta}_{\text{gCD}}$ denote the student parameters. We consider

$$\mathcal{L}_{\text{gCD}}(\boldsymbol{\theta}_{\text{gCD}}; u) := \mathbb{E}_t \mathbb{E}_{\mathbf{x}_t \sim p_t} \left[ \left\| \mathbf{f}_{\boldsymbol{\theta}_{\text{gCD}}}(\mathbf{x}_t, t) - \mathbf{f}_{\boldsymbol{\theta}_{\text{gCD}}}\left( \texttt{Solver}_{t \to u}(\mathbf{x}_t), u \right) \right\|_2^2 \right] < \varepsilon.$$

The loss $\mathcal{L}_{\text{gCD}}$ includes two important special cases. First, when $u = t - \Delta t$, it reduces to the conventional consistency distillation objective (Song et al., 2023), where the solver is applied for a single step. Second, when $u = 0$, it resembles knowledge distillation (Luhman & Luhman, 2021). In this case, by construction of the consistency model parametrization,

$$\mathbf{f}_{\boldsymbol{\theta}_{\text{gCD}}}\left( \texttt{Solver}_{t \to 0}(\mathbf{x}_t), 0 \right) = \texttt{Solver}_{t \to 0}(\mathbf{x}_t).$$

**Random Initialization.** We assume that a randomly initialized parameter $\boldsymbol{\theta}_{\text{rand.}}$ satisfies

$$\mathbb{E}_t \mathbb{E}_{\mathbf{x}_t \sim p_t} \left\| \mathbf{f}_{\boldsymbol{\theta}_{\text{rand.}}}(\mathbf{x}_t, t) \right\|_2^2 < R,$$

for some constant $R > 0$.

### F.3 PREREQUISITES

**Key Assumptions.** We first present the summary of assumptions in our individual propositions.

**Assumption A** (Data Distribution). *The data distribution $p_{\text{data}}$ has bounded support and finite second moments:*

$$m := \mathbb{E}_{p_{\text{data}}} \|\mathbf{x}_0\|_2^2 < \infty.$$

**Assumption B** (Smoothness). *For $\boldsymbol{\theta} = \boldsymbol{\theta}_{\text{CMT}}, \boldsymbol{\theta}_{\text{DM}}$, or $\boldsymbol{\theta}_{\text{gCD}}$, we assume the following conditions hold:*

*(i) Bounded Value and Jacobian:*

$$\|\mathbf{f}_{\boldsymbol{\theta}}\|_2, \ \|\nabla_{\boldsymbol{\theta}} \mathbf{f}_{\boldsymbol{\theta}}(\mathbf{x}_t, t)\|_F \leq R, \quad \text{for some constant } R < \infty.$$

*(ii) There exist $\text{Lip}(\mathbf{f}_{\boldsymbol{\theta}}) > 0$ such that for all $\mathbf{x}, \mathbf{y} \in \mathbb{R}^D$ and $s, t \in [0, T]$,*

$$\|\mathbf{f}_{\boldsymbol{\theta}}(\mathbf{x}, t) - \mathbf{f}_{\boldsymbol{\theta}}(\mathbf{y}, s)\| \leq \text{Lip}(\mathbf{f}_{\boldsymbol{\theta}}) \left( \|\mathbf{x} - \mathbf{y}\| + |t - s| \right).$$

**Assumption C** (Oracle Flow Map and Solver). *We assume the exact flow $\boldsymbol{\Psi}_{s \to t}$ and the solver $\text{Solver}_{s \to t}$, using the teacher drift, satisfy the following conditions:*

*(i) Finite targets: $C_{\boldsymbol{\Psi}} := \sup_t \mathbb{E}_{\mathbf{x}_t \sim p_t} \|\boldsymbol{\Psi}_{t \to 0}(\mathbf{x}_t)\|_2^2 < \infty.$*

*(ii) The exact flow is Lipschitz in state: for some $\text{Lip}(\boldsymbol{\Psi}) \geq 1$,*

$$\|\boldsymbol{\Psi}_{s \to t}(\mathbf{x}) - \boldsymbol{\Psi}_{s \to t}(\mathbf{y})\| \leq \text{Lip}(\boldsymbol{\Psi})\|\mathbf{x} - \mathbf{y}\|;$$

*(iii) The solver is Lipschitz in state and time: for some $\text{Lip}(\text{Solver}) \geq 1$*

$$\|\text{Solver}_{t \to u}(\mathbf{x}) - \text{Solver}_{s \to u}(\mathbf{y})\| \leq \text{Lip}(\text{Solver}) \left( \|\mathbf{x} - \mathbf{y}\| + |t - s| \right)$$

*(iv) The solver $\text{Solver}_{s \to t}$ is a zero-stable, global order-p solver with $p \geq 1$:*

$$\sup_{\mathbf{x}_s \sim p_s} \|\text{Solver}_{s \to t}(\mathbf{x}_s) - \boldsymbol{\Psi}_{s \to t}(\mathbf{x}_s)\| = \mathcal{O}(\Delta t^p), \qquad s \geq t.$$

**Some Lemmas.** We summarize some auxiliary tools that we will use later.

**Lemma F.1.** *Let $\mathbf{x}_0 \in \mathbb{R}^D$ be square–integrable and let $\mathbf{x}_t$ be any random variable on the same probability space. For any (deterministic) decoder $\mathbf{F}$ such that $\mathbf{F}(\mathbf{x}_t)$ is square–integrable,*

$$\mathbb{E}\left[ \|\mathbf{x}_0 - \mathbf{F}(\mathbf{x}_t)\|_2^2 \right] = \mathbb{E}\left[ \text{Tr} \, \text{Var}(\mathbf{x}_0 | \mathbf{x}_t) \right] + \mathbb{E}\left[ \|\mathbb{E}[\mathbf{x}_0 | \mathbf{x}_t] - \mathbf{F}(\mathbf{x}_t)\|_2^2 \right].$$

*Proof.* Write the conditional mean (posterior mean) as

$$\boldsymbol{\mu}(\mathbf{x}_t) := \mathbb{E}[\mathbf{x}_0 | \mathbf{x}_t],$$

and the zero–mean conditional residual as

$$\mathbf{e} := \mathbf{x}_0 - \boldsymbol{\mu}(\mathbf{x}_t), \qquad \text{so that} \quad \mathbb{E}[\mathbf{e} | \mathbf{x}_t] = \mathbf{0}.$$

Then for any $\mathbf{F}$,

$$\mathbf{x}_0 - \mathbf{F}(\mathbf{x}_t) = \underbrace{(\mathbf{x}_0 - \boldsymbol{\mu}(\mathbf{x}_t))}_{=\mathbf{e}} + (\boldsymbol{\mu}(\mathbf{x}_t) - \mathbf{F}(\mathbf{x}_t)).$$

Expand the squared norm and take expectations over $(t, \mathbf{x}_t)$:

$$\mathbb{E}\left[\|\mathbf{x}_0 - \mathbf{F}(\mathbf{x}_t)\|_2^2\right] = \mathbb{E}\left[\|\mathbf{e}\|_2^2\right] + \mathbb{E}\left[\|\boldsymbol{\mu}(\mathbf{x}_t) - \mathbf{F}(\mathbf{x}_t)\|_2^2\right] + 2\mathbb{E}\left[\langle \mathbf{e}, \boldsymbol{\mu}(\mathbf{x}_t) - \mathbf{F}(\mathbf{x}_t)\rangle\right].$$

The cross term vanishes by the tower property and the fact that $\boldsymbol{\mu}(\mathbf{x}_t) - \mathbf{F}(\mathbf{x}_t)$ is $\sigma(\mathbf{x}_t)$–measurable:

$$\begin{aligned}
\mathbb{E}\left[\langle \mathbf{e}, \boldsymbol{\mu}(\mathbf{x}_t) - \mathbf{F}(\mathbf{x}_t)\rangle\right] &= \mathbb{E}\left[\mathbb{E}\left[\langle \mathbf{e}, \boldsymbol{\mu}(\mathbf{x}_t) - \mathbf{F}(\mathbf{x}_t)\rangle|\mathbf{x}_t\right]\right] \\
&= \mathbb{E}\left[\langle \mathbb{E}[\mathbf{e}|\mathbf{x}_t], \boldsymbol{\mu}(\mathbf{x}_t) - \mathbf{F}(\mathbf{x}_t)\rangle\right] \\
&= \mathbb{E}\left[\langle \mathbf{0}, \boldsymbol{\mu}(\mathbf{x}_t) - \mathbf{F}(\mathbf{x}_t)\rangle\right] = 0.
\end{aligned}$$

For the first term, use the definition of conditional covariance:

$$\mathrm{Var}(\mathbf{x}_0|\mathbf{x}_t) = \mathbb{E}\left[\mathbf{e}\mathbf{e}^\top|\mathbf{x}_t\right],$$

whose trace equals the conditional mean squared residual:

$$\mathrm{Tr}\,\mathrm{Var}(\mathbf{x}_0|\mathbf{x}_t) = \mathrm{tr}\,\mathbb{E}\left[\mathbf{e}\mathbf{e}^\top|\mathbf{x}_t\right] = \mathbb{E}\left[\|\mathbf{e}\|_2^2|\mathbf{x}_t\right].$$

Taking expectations over $\mathbf{x}_t$ yields

$$\mathbb{E}\left[\|\mathbf{e}\|_2^2\right] = \mathbb{E}\left[\mathrm{Tr}\,\mathrm{Var}(\mathbf{x}_0|\mathbf{x}_t)\right].$$

Combining the pieces gives

$$\mathbb{E}\left[\|\mathbf{x}_0 - \mathbf{F}(\mathbf{x}_t)\|_2^2\right] = \mathbb{E}\left[\mathrm{Tr}\,\mathrm{Var}(\mathbf{x}_0|\mathbf{x}_t)\right] + \mathbb{E}\left[\|\boldsymbol{\mu}(\mathbf{x}_t) - \mathbf{F}(\mathbf{x}_t)\|_2^2\right],$$

which is the claimed identity. $\qquad\square$

**Lemma F.2.** *Let the Assumption C hold. Then*

$$\left\|\mathtt{Solver}_{s\to t}\big(\mathtt{Solver}_{t\to s}(\mathbf{x}_t)\big) - \mathbf{x}_t\right\| = \mathcal{O}\left(\Delta t^p\right).$$

*Proof.* Let $\boldsymbol{\Phi}_{t\to s} := \mathtt{Solver}_{t\to s}$ be a numerical solver (using the teacher drift) on a uniform grid with step size $\Delta t$. For any $t$ and $\mathbf{x}_t$,

$$\begin{aligned}
\left\|\boldsymbol{\Phi}_{s\to t}\big(\boldsymbol{\Phi}_{t\to s}(\mathbf{x}_t)\big) - \mathbf{x}_t\right\| &\leq \underbrace{\left\|\boldsymbol{\Phi}_{s\to t}\big(\boldsymbol{\Phi}_{t\to s}(\mathbf{x}_t)\big) - \boldsymbol{\Psi}_{s\to t}\big(\boldsymbol{\Phi}_{t\to s}(\mathbf{x}_t)\big)\right\|}_{\text{backward global error}} \\
&\quad + \underbrace{\left\|\boldsymbol{\Psi}_{s\to t}\big(\boldsymbol{\Phi}_{t\to s}(\mathbf{x}_t)\big) - \boldsymbol{\Psi}_{s\to t}\big(\boldsymbol{\Psi}_{t\to s}(\mathbf{x}_t)\big)\right\|}_{\text{propagation of forward error}} \\
&\leq C\Delta t^p + L\left\|\boldsymbol{\Phi}_{t\to s}(\mathbf{x}_t) - \boldsymbol{\Psi}_{t\to s}(\mathbf{x}_t)\right\| \\
&\leq (1 + L)C\Delta t^p.
\end{aligned}$$

Therefore,

$$\left\|\mathtt{Solver}_{s\to t}\big(\mathtt{Solver}_{t\to s}(\mathbf{x}_t)\big) - \mathbf{x}_t\right\| = \mathcal{O}\left(\Delta t^p\right).$$

$\qquad\square$

In the proofs we repeatedly use the following inequality, derived from the triangle inequality and the Cauchy–Schwarz inequality, without stating it explicitly. This inequality allows us to convert bounds in the $\ell_2$ norm into bounds in the squared $\ell_2$ norm.

**Lemma F.3.** *Let $N$ be an integer, and $\{a_i\}_{i=1}^N$ be a sequence of real numbers. Then*

$$\left(\sum_{i=1}^N a_i\right)^2 \leq N \sum_{i=1}^N a_i^2.$$

### F.4 ANALYSIS OF GRADIENT BIAS

We focus on the setting where the distance function is the squared $\ell_2$ norm,

$$d(\mathbf{x}, \mathbf{y}) := \|\mathbf{x} - \mathbf{y}\|_2^2,$$

and the weight function is uniform, $w(t) \equiv 1$. The extension to more general choices of distance or weighting follows in the same way. Throughout this section we work with the CM flow map $\mathbf{\Psi}_{t \to 0}$; analogous statements for other flow maps, such as the CTM family, can be derived by following the same arguments presented here.

For convenience, we rewrite Equation (2) in the simplified form

$$\ell_{\mathrm{oracle}}(\boldsymbol{\theta}; \boldsymbol{\xi}) := \left\| \mathbf{f}_{\boldsymbol{\theta}}(\mathbf{x}_t, t) - \mathbf{\Psi}_{t \to 0}(\mathbf{x}_t) \right\|_2^2, \tag{12}$$

and define the CM training loss as

$$\ell_{\mathrm{CM}}(\boldsymbol{\theta}; \boldsymbol{\xi}) := \left\| \mathbf{f}_{\boldsymbol{\theta}}(\mathbf{x}_t, t) - \mathbf{f}_{\boldsymbol{\theta}^-}(\mathbf{x}_{t-\Delta t}, t - \Delta t) \right\|_2^2, \tag{13}$$

where $\boldsymbol{\xi} = (t, \mathbf{x}_t) \sim \mathrm{Unif}[0, T](t) \times p_t$ denotes the training sample, with $\mathrm{Unif}[0, T](t)$ representing the time sampling distribution (for example, uniform over $[0, T]$ or any chosen weighting).

We then introduce the expected objectives

$$\bar{\ell}_{\mathrm{oracle}}(\boldsymbol{\theta}) := \mathbb{E}_{\boldsymbol{\xi}}\big[\ell_{\mathrm{oracle}}(\boldsymbol{\theta}; \boldsymbol{\xi})\big], \qquad \bar{\ell}_{\mathrm{CM}}(\boldsymbol{\theta}) := \mathbb{E}_{\boldsymbol{\xi}}\big[\ell_{\mathrm{CM}}(\boldsymbol{\theta}; \boldsymbol{\xi})\big],$$

which represent the oracle target loss and the CM training loss, respectively. Finally, we define the squared gradient bias

$$\mathcal{B}(\boldsymbol{\theta}) := \left\| \nabla_{\boldsymbol{\theta}} \bar{\ell}_{\mathrm{oracle}}(\boldsymbol{\theta}) - \nabla_{\boldsymbol{\theta}} \bar{\ell}_{\mathrm{CM}}(\boldsymbol{\theta}) \right\|_2^2.$$

**Theorem F.1** (Bias Comparisons). *Assume that Assumptions A to C hold with $p \geq 1$. Then the following bias comparisons are valid for the four different initialization schemes ($\boldsymbol{\theta} = \boldsymbol{\theta}_{\mathrm{CMT}}$, $\boldsymbol{\theta}_{\mathrm{DM}}$, $\boldsymbol{\theta}_{\mathrm{gCD}}$, or random initialization $\boldsymbol{\theta}_{\mathrm{rand.}}$) of flow map model training:*

*(i)* **CMT** *:*

$$\mathcal{B}(\boldsymbol{\theta}_{\mathrm{CMT}}) = \mathcal{O}\left(\varepsilon + \Delta t^2 + \Delta t^p\right).$$

*(ii)* **Diffusion Model***:*

$$\mathcal{B}(\boldsymbol{\theta}_{\mathrm{DM}}) = \mathcal{O}\left(\varepsilon + \Delta t^2 + \mathbb{E}_t\left[\tfrac{\sigma_t^2}{\alpha_t^2}\right]\right) + \mathbb{E}_{\mathbf{x}_t, t}\left[\left\|\mathbf{\Psi}_{t \to 0}(\mathbf{x}_t) - \mathbb{E}[\mathbf{x}_0 | \mathbf{x}_t]\right\|_2^2\right].$$

*(iii)* **General Consistency Distillation***: For a fixed $u \in [0, T]$, assume in addition that*

$$\delta_u := \mathbb{E}_{\mathbf{x}_u \sim p_u}\left\| \mathbf{f}_{\boldsymbol{\theta}_{\mathrm{gCD}}}(\mathbf{x}_u, u) - \mathbf{\Psi}_{u \to 0}(\mathbf{x}_u) \right\|^2 < \infty.$$

*Then*

$$\mathcal{B}(\boldsymbol{\theta}_{\mathrm{gCD}}) = \mathcal{O}\left(\varepsilon + \Delta t^2 + \delta_u\right).$$

*(iv)* **Random Initialization***:*

$$\mathcal{B}(\boldsymbol{\theta}_{\mathrm{rand.}}) = \mathcal{O}(1).$$

*Proof.* Taking the gradient of $\bar{\ell}_{\mathrm{oracle}}$, we obtain the unbiased oracle CM gradient as:

$$\nabla_{\boldsymbol{\theta}} \bar{\ell}_{\mathrm{oracle}}(\boldsymbol{\theta}) = \mathbb{E}_t \mathbb{E}_{\mathbf{x}_t \sim p_t(\mathbf{x}_t)}\left[\nabla_{\boldsymbol{\theta}} \mathbf{f}_{\boldsymbol{\theta}}(\mathbf{x}_t, t) \cdot \left(\mathbf{f}_{\boldsymbol{\theta}}(\mathbf{x}_t, t) - \mathbf{\Psi}_{t \to 0}(\mathbf{x}_t)\right)\right].$$

Likewise, the CM gradient is

$$\nabla_{\boldsymbol{\theta}} \bar{\ell}_{\mathrm{CM}}(\boldsymbol{\theta}) = \mathbb{E}_t \mathbb{E}_{\mathbf{x}_t \sim p_t(\mathbf{x}_t)}\left[\nabla_{\boldsymbol{\theta}} \mathbf{f}_{\boldsymbol{\theta}}(\mathbf{x}_t, t) \cdot \left(\mathbf{f}_{\boldsymbol{\theta}}(\mathbf{x}_t, t) - \mathbf{f}_{\boldsymbol{\theta}^-}(\mathbf{x}_{t-\Delta t}, t - \Delta t)\right)\right].$$

CM is approximating $\mathbf{\Psi}_{t \to 0}(\mathbf{x}_t)$ with $f_{\boldsymbol{\theta}^-}(\mathbf{x}_{t-\Delta t}, t - \Delta t)$. CMT and diffusion differ in the initialization of $\boldsymbol{\theta}$. The one-point bias can be bounded:

$$\left\| \nabla_{\boldsymbol{\theta}} \bar{\ell}_{\mathrm{CM}}(\boldsymbol{\theta}) - \nabla_{\boldsymbol{\theta}} \bar{\ell}_{\mathrm{Oracle}}(\boldsymbol{\theta}) \right\|_2$$

$$\leq \mathbb{E}_{\boldsymbol{\xi}}\left[\left\| \nabla_{\boldsymbol{\theta}} \mathbf{f}_{\boldsymbol{\theta}}(\mathbf{x}_t, t) \cdot \left(\mathbf{f}_{\boldsymbol{\theta}^-}(\mathbf{x}_{t-\Delta t}, t - \Delta t) - \mathbf{\Psi}_{t \to 0}(\mathbf{x}_t)\right) \right\|_2\right]$$

$$\leq \mathbb{E}_{\boldsymbol{\xi}}\left[\left\| \nabla_{\boldsymbol{\theta}} \mathbf{f}_{\boldsymbol{\theta}}(\mathbf{x}_t, t)\right\|_2 \cdot \left\|\mathbf{f}_{\boldsymbol{\theta}^-}(\mathbf{x}_{t-\Delta t}, t - \Delta t) - \mathbf{\Psi}_{t \to 0}(\mathbf{x}_t)\right\|_2\right]$$

$$\leq G \cdot \mathbb{E}_{\boldsymbol{\xi}}\left[\left\|\mathbf{f}_{\boldsymbol{\theta}^-}(\mathbf{x}_{t-\Delta t}, t - \Delta t) - \mathbf{\Psi}_{t \to 0}(\mathbf{x}_t)\right\|_2\right],$$

Namely, the deviation with the gradient of the oracle loss is upper bounded as

$$\|\nabla_{\boldsymbol{\theta}}\ell_{\text{CM}}(\boldsymbol{\theta}) - \nabla_{\boldsymbol{\theta}}\ell_{\text{Oracle}}(\boldsymbol{\theta})\|_2 \leq G \cdot \mathbb{E}_{\boldsymbol{\xi}}\Big[\|\mathbf{f}_{\boldsymbol{\theta}^-}(\mathbf{x}_{t-\Delta t}, t - \Delta t) - \boldsymbol{\Psi}_{t\to 0}(\mathbf{x}_t)\|_2\Big].$$

In the following, we individually derive the upper bound for different initialization scenarios. Denote $t' := t - \Delta t$ for notational simplicity.

**Case 1. CMT :** We denote $\boldsymbol{\Phi}_{t\to s}(\mathbf{x}_t) := \texttt{Solver}_{t\to s}(\mathbf{x}_t)$. Given a sample $\mathbf{x}_t \sim p_t$ and time $t$, define $\widehat{\mathbf{x}}_t := \boldsymbol{\Phi}_{T\to t}(\boldsymbol{\Phi}_{t\to T}(\mathbf{x}_t))$.

For CMT initialization at $\boldsymbol{\theta} = \boldsymbol{\theta}_{\text{CMT}}$, we have

$$\|\mathbf{f}_{\boldsymbol{\theta}_{\text{CMT}}^-}(\mathbf{x}_{t'}, t') - \boldsymbol{\Psi}_{t\to 0}(\mathbf{x}_t)\|_2$$
$$= \|\mathbf{f}_{\boldsymbol{\theta}_{\text{CMT}}}(\mathbf{x}_{t'}, t') - \boldsymbol{\Psi}_{t\to 0}(\mathbf{x}_t)\|_2$$
$$\leq \|\mathbf{f}_{\boldsymbol{\theta}_{\text{CMT}}}(\mathbf{x}_t, t) - \mathbf{f}_{\boldsymbol{\theta}_{\text{CMT}}}(\mathbf{x}_{t'}, t')\|_2 + \|\mathbf{f}_{\boldsymbol{\theta}_{\text{CMT}}}(\mathbf{x}_t, t) - \boldsymbol{\Phi}_{T\to 0}(\boldsymbol{\Phi}_{t\to T}(\mathbf{x}_t))\|$$
$$+ \|\boldsymbol{\Phi}_{T\to 0}(\boldsymbol{\Phi}_{t\to T}(\mathbf{x}_t)) - \boldsymbol{\Psi}_{t\to 0}(\mathbf{x}_t)\|$$
$$=: (\text{I}) + (\text{II}) + (\text{III}).$$

For (I), by Lipschitzness and the forward parameterization,

$$(\text{I}) \leq \text{Lip}(\mathbf{f}_{\boldsymbol{\theta}_{\text{CMT}}})\big(\|\mathbf{x}_{t'} - \mathbf{x}_t\| + |t' - t|\big), \qquad \mathbb{E}\|\mathbf{x}_{t'} - \mathbf{x}_t\|_2^2 = \mathcal{O}(\Delta t^2),$$

since $\alpha_{t'} - \alpha_t = \mathcal{O}(\Delta t)$ and $\beta_{t'} - \beta_t = \mathcal{O}(\Delta t)$. Hence $\mathbb{E}[(\text{I})^2] = \mathcal{O}(\Delta t^2)$.

For (III), since $\boldsymbol{\Psi}_{t\to 0}(\mathbf{x}_t) = \boldsymbol{\Psi}_{T\to 0}(\boldsymbol{\Psi}_{t\to T}(\mathbf{x}_t))$, with Lemma F.2 we have $\|\widehat{\mathbf{x}}_t - \mathbf{x}_t\| = \mathcal{O}(\Delta t^p)$. Thus,

$$\mathbb{E}[(\text{III})^2] = \mathcal{O}(\Delta t^{2p}).$$

For (II), we first insert $\widehat{\mathbf{x}}_t = \boldsymbol{\Phi}_{T\to t}(\boldsymbol{\Phi}_{t\to T}(\mathbf{x}_t))$:

$$(\text{II}) \leq \|\mathbf{f}_{\boldsymbol{\theta}_{\text{CMT}}}(\mathbf{x}_t, t) - \mathbf{f}_{\boldsymbol{\theta}_{\text{CMT}}}(\widehat{\mathbf{x}}_t, t)\| + \|\mathbf{f}_{\boldsymbol{\theta}_{\text{CMT}}}(\widehat{\mathbf{x}}_t, t) - \boldsymbol{\Phi}_{T\to 0}(\boldsymbol{\Phi}_{t\to T}(\mathbf{x}_t))\| =: (\text{IIa}) + (\text{IIb}).$$

From Lemma F.2 and Lipschitzness, $\mathbb{E}[(\text{IIa})^2] = \mathcal{O}(\Delta t^{2p})$. For (IIb), define

$$g(\mathbf{x}_T, t) := \|\mathbf{f}_{\boldsymbol{\theta}_{\text{CMT}}}(\boldsymbol{\Phi}_{T\to t}(\mathbf{x}_T), t) - \boldsymbol{\Phi}_{T\to 0}(\mathbf{x}_T)\|_2^2.$$

Then

$$(\text{IIb})^2 = g(\boldsymbol{\Phi}_{t\to T}(\mathbf{x}_t), t)$$

When the CMT's training expectation is taken with $\mathbf{x}_T \sim p_{\text{prior}}$ and $t \sim \text{Unif}[0, T]$, we compare $\mathbb{E}_{(\mathbf{x}_T, t)\sim p_{\text{prior}}\times\text{Unif}}[g(\mathbf{x}_T, t)]$ to $\mathbb{E}_{(\mathbf{x}_t, t)}[g(\boldsymbol{\Phi}_{t\to T}(\mathbf{x}_t), t)]$. Using the coupling $\mathbf{x}_T^\star \sim p_{\text{prior}}$, $\mathbf{x}_t = \boldsymbol{\Psi}_{T\to t}(\mathbf{x}_T^\star)$, standard stability of $\boldsymbol{\Phi}$ and $\mathbf{f}$ yields a Lipschitz constant $\text{Lip}(g)$ (in $\mathbf{x}_T$) such that

$$\big|\mathbb{E}g(\boldsymbol{\Phi}_{t\to T}(\mathbf{x}_t), t) - \mathbb{E}g(\mathbf{x}_T^\star, t)\big| \leq \text{Lip}(g)\mathbb{E}\|\boldsymbol{\Phi}_{t\to T}(\mathbf{x}_t) - \mathbf{x}_T^\star\|_2 = \mathcal{O}(\Delta t^p).$$

Hence

$$\mathbb{E}[(\text{IIb})^2] = \mathbb{E}g(\boldsymbol{\Phi}_{t\to T}(\mathbf{x}_t), t) \leq \mathbb{E}g(\mathbf{x}_T^\star, t) + \mathcal{O}(\Delta t^p) \leq \varepsilon + \mathcal{O}(\Delta t^p).$$

Collecting the bounds,

$$\mathbb{E}\|\mathbf{f}_{\boldsymbol{\theta}_{\text{CMT}}}(\mathbf{x}_{t'}, t') - \boldsymbol{\Psi}_{t\to 0}(\mathbf{x}_t)\|_2^2 \leq 3\mathcal{O}(\Delta t^2) + 3\big(\varepsilon + \mathcal{O}(\Delta t^p)\big) + 3\mathcal{O}(\Delta t^{2p}).$$

Thus,

$$\mathbb{E}\|\mathbf{f}_{\boldsymbol{\theta}_{\text{CMT}}}(\mathbf{x}_{t'}, t') - \boldsymbol{\Psi}_{t\to 0}(\mathbf{x}_t)\|_2^2 = \varepsilon + \mathcal{O}(\Delta t^p) + \mathcal{O}(\Delta t^2) \qquad (p \geq 1).$$

**Case 2. Diffusion Model:** Let the CM loss be initialized at the pre-trained diffusion model weights $\boldsymbol{\theta} = \boldsymbol{\theta}_{\text{DM}}$, then we have

$$\|\mathbf{f}_{\boldsymbol{\theta}_{\text{DM}}}(\mathbf{x}_t', t') - \boldsymbol{\Psi}_{t\to 0}(\mathbf{x}_t)\|_2$$
$$\leq \|\mathbf{f}_{\boldsymbol{\theta}_{\text{DM}}}(\mathbf{x}_{t'}, t') - \mathbf{f}_{\boldsymbol{\theta}_{\text{DM}}}(\mathbf{x}_t, t)\|_2 + \|\mathbf{f}_{\boldsymbol{\theta}_{\text{DM}}}(\mathbf{x}_t, t) - \mathbf{x}_0\|_2 + \|\mathbf{x}_0 - \boldsymbol{\Psi}_{t\to 0}(\mathbf{x}_t)\|_2$$
$$\lesssim \|\mathbf{f}_{\boldsymbol{\theta}_{\text{DM}}}(\mathbf{x}_{t'}, t') - \mathbf{f}_{\boldsymbol{\theta}_{\text{DM}}}(\mathbf{x}_t, t)\|_2^2 + \|\mathbf{f}_{\boldsymbol{\theta}_{\text{DM}}}(\mathbf{x}_t, t) - \mathbf{x}_0\|_2^2 + \|\mathbf{x}_0 - \boldsymbol{\Psi}_{t\to 0}(\mathbf{x}_t)\|_2^2$$
$$=: (\text{I}) + (\text{II}) + (\text{III}).$$

For (I) and (II), we have

$$\mathbb{E}_{\boldsymbol{\xi}}\left[\|\mathbf{f}_{\boldsymbol{\theta}_{\mathrm{DM}}}(\mathbf{x}_t,t) - \mathbf{f}_{\boldsymbol{\theta}_{\mathrm{DM}}}(\mathbf{x}_{t-\Delta t},t-\Delta t)\|_2^2\right] + \mathbb{E}_{\boldsymbol{\xi}}\left[\|\mathbf{f}_{\boldsymbol{\theta}_{\mathrm{DM}}}(\mathbf{x}_t,t) - \mathbf{x}_0\|_2^2\right]$$
$$\leq \mathrm{Lip}(\boldsymbol{\theta}_{\mathrm{DM}})\mathbb{E}_{\boldsymbol{\xi}}\left[\|\mathbf{x}_t - \mathbf{x}_t'\|_2^2\right] + \varepsilon$$
$$= \mathcal{O}(\Delta t^2 + \varepsilon),$$

following the similar argument in CMT's case.

However, using the pre-trained diffusion model's weight as an initialization induces an additional discrepancy between the data $\mathbf{x}_0$ and the reverse-time solution of ODE $\boldsymbol{\Psi}_{t\to 0}(\mathbf{x}_t)$, where $\mathbf{x}_t$ is perturbed from $\mathbf{x}_0$. We will obtain a general upper bound of the term (III) with $\mathbb{E}_{\boldsymbol{\xi}}\left[\|\mathbf{x}_0 - \boldsymbol{\Psi}_{t\to 0}(\mathbf{x}_t)\|_2^2\right]$.

Applying Lemma F.1 with $\mathbf{F} = \boldsymbol{\Psi}_{t\to 0}$ and then averaging over $t$, we obtain

$$\mathbb{E}_{\mathbf{x}_0,\boldsymbol{\epsilon},t}\left[\|\mathbf{x}_0 - \boldsymbol{\Psi}_{t\to 0}(\mathbf{x}_t)\|_2^2\right] = \mathbb{E}_{t,\mathbf{x}_t}\left[\mathrm{Tr}\,\mathrm{Var}(\mathbf{x}_0|\mathbf{x}_t,t)\right] + \mathbb{E}_{t,\mathbf{x}_t}\left[\|\boldsymbol{\Psi}_{t\to 0}(\mathbf{x}_t) - \mathbb{E}[\mathbf{x}_0|\mathbf{x}_t]\|_2^2\right].$$

where the second term means the extra MSE we pay because the flow map is not necessary the Bayes–optimal estimator (the posterior mean).

Below we compute the upper bound for $\mathbb{E}_{t,\mathbf{x}_t}\left[\mathrm{Tr}\,\mathrm{Var}(\mathbf{x}_0|\mathbf{x}_t)\right]$. Given an observation $\mathbf{x}_t$ with $t$ fixed, the minimum mean–squared error (mmse) is

$$\mathsf{mmse}(t) := \inf_{\mathbf{f}} \mathbb{E}\left[\|\mathbf{x}_0 - \mathbf{f}_{\boldsymbol{\theta}_{\mathrm{CMT}}}(\mathbf{x}_t)\|_2^2\right],$$

where the infimum is over all measurable $\mathbf{f}$ with finite second moment. The minimizer is the posterior mean $\mathbb{E}[\mathbf{x}_0|\mathbf{x}_t]$, and

$$\mathsf{mmse}(t) = \mathbb{E}\left[\|\mathbf{x}_0 - \mathbb{E}[\mathbf{x}_0|\mathbf{x}_t]\|_2^2\right] = \mathbb{E}_{\mathbf{x}_t}\left[\mathrm{Tr}\,\mathrm{Var}(\mathbf{x}_0|\mathbf{x}_t)\right].$$

Since $\mathsf{mmse}(t)$ is the minimum risk over all estimators, its value is bounded above by the risk of any specific estimator. Take the linear estimator $\mathbf{f}_{\boldsymbol{\theta}_{\mathrm{CMT}}}(\mathbf{x}_t) = (1/\alpha_t)\mathbf{x}_t$. Using $\mathbf{x}_t = \alpha_t\mathbf{x}_0 + \sigma_t\boldsymbol{\epsilon}$ and the independence of $\mathbf{x}_0$ and $\boldsymbol{\epsilon}$,

$$\mathbb{E}_{\mathbf{x}_t}\left[\mathrm{Tr}\,\mathrm{Var}(\mathbf{x}_0|\mathbf{x}_t)\right] \leq \mathbb{E}\left\|\mathbf{x}_0 - \tfrac{1}{\alpha_t}\mathbf{x}_t\right\|_2^2 = \mathbb{E}\left\|\mathbf{x}_0 - \tfrac{1}{\alpha_t}(\alpha_t\mathbf{x}_0+\sigma_t\boldsymbol{\epsilon})\right\|_2^2 = \mathbb{E}\left\|-\tfrac{\sigma_t}{\alpha_t}\boldsymbol{\epsilon}\right\|_2^2 = \tfrac{\sigma_t^2}{\alpha_t^2}\mathbb{E}\|\boldsymbol{\epsilon}\|_2^2 = \tfrac{\sigma_t^2}{\alpha_t^2}D.$$

Hence $\mathsf{mmse}(t) \leq (\sigma_t^2/\alpha_t^2)D$. Averaging over $t$ gives the bound:

$$\mathbb{E}_{\mathbf{x}_0,\boldsymbol{\epsilon},t}\left[\|\mathbf{x}_0 - \boldsymbol{\Psi}_{t\to 0}(\mathbf{x}_t)\|_2^2\right] \leq D\mathbb{E}_t\left[\tfrac{\sigma_t^2}{\alpha_t^2}\right] + \mathbb{E}_{\mathbf{x}_t,t}\left[\|\boldsymbol{\Psi}_{t\to 0}(\mathbf{x}_t) - \mathbb{E}[\mathbf{x}_0|\mathbf{x}_t]\|_2^2\right]$$

Therefore, to summarize, we have

$$\mathbb{E}_{\boldsymbol{\xi}}\|\mathbf{f}_{\boldsymbol{\theta}_{\mathrm{DM}}}(\mathbf{x}_t',t') - \boldsymbol{\Psi}_{t\to 0}(\mathbf{x}_t)\|_2^2 = \mathcal{O}\left(\Delta t^2 + \varepsilon + \mathbb{E}_t\left[\tfrac{\sigma_t^2}{\alpha_t^2}\right]\right) + \mathbb{E}_{\mathbf{x}_t,t}\left[\|\boldsymbol{\Psi}_{t\to 0}(\mathbf{x}_t) - \mathbb{E}[\mathbf{x}_0|\mathbf{x}_t]\|_2^2\right]$$

**Case 3. General Consistency Distillation:** Let $\boldsymbol{\Phi}_{t\to s}(\mathbf{x}_t) := \mathtt{Solver}_{t\to s}(\mathbf{x}_t)$ be the $p$-th order solver, solving the PF-ODE with the teacher diffusion model's drift.

When initializing at $\boldsymbol{\theta} = \boldsymbol{\theta}_{\mathrm{gCD}}$, we need to additionally assume that:

$$\delta_u := \mathbb{E}_{\mathbf{x}_u \sim p_u}\left\|\mathbf{f}_{\boldsymbol{\theta}_{\mathrm{gCD}}}(\mathbf{x}_u,u) - \boldsymbol{\Psi}_{u\to 0}(\mathbf{x}_u)\right\|^2 < \infty.$$

General CD at a single $u$ does not control the bias to the oracle $\boldsymbol{\Psi}_{u\to 0}$; $\delta_u$ can be arbitrarily large even if the General CD is trained well with small $\varepsilon$. The term $\delta_u$ supplies the necessary anchor at $u$.

We use triangle inequality to obtain

$$\left\|\mathbf{f}_{\boldsymbol{\theta}_{\mathrm{gCD}}}(\mathbf{x}_t',t') - \boldsymbol{\Psi}_{t\to 0}(\mathbf{x}_t)\right\|_2^2$$
$$\lesssim \left\|\mathbf{f}_{\boldsymbol{\theta}_{\mathrm{gCD}}}(\mathbf{x}_{t'},t') - \mathbf{f}_{\boldsymbol{\theta}_{\mathrm{gCD}}}(\mathbf{x}_t,t)\right\|_2^2 + \left\|\mathbf{f}_{\boldsymbol{\theta}_{\mathrm{gCD}}}(\mathbf{x}_t,t) - \mathbf{f}_{\boldsymbol{\theta}_{\mathrm{gCD}}}\left(\boldsymbol{\Phi}_{t\to u}(\mathbf{x}_t),u\right)\right\|_2^2$$
$$+ \left\|\mathbf{f}_{\boldsymbol{\theta}_{\mathrm{gCD}}}\left(\boldsymbol{\Phi}_{t\to u}(\mathbf{x}_t),u\right) - \boldsymbol{\Psi}_{t\to 0}(\mathbf{x}_t)\right\|_2^2$$
$$=:(\mathrm{I}) + (\mathrm{II}) + (\mathrm{III}).$$

For (I),

$$\mathbb{E}_{\boldsymbol{\xi}}[(\text{I})] \lesssim \text{Lip}(\mathbf{f}_{\boldsymbol{\theta}_{\text{gCD}}}) \left( \|\mathbf{x}_{t'} - \mathbf{x}_t\|_2^2 + \Delta t^2 \right).$$

From $\mathbf{x}_t = \alpha_t \mathbf{x}_0 + \sigma_t \boldsymbol{\epsilon}$, independence of $\mathbf{x}_0$ and $\boldsymbol{\epsilon}$, we follow the similar derivation as the previous CMT's cases: Hence

$$\mathbb{E}[(\text{I})] = \mathcal{O}(\Delta t^2).$$

For (II), by the hypothesis $\mathcal{L}_{\text{gCD}}(\boldsymbol{\theta}; u) \leq \varepsilon$,

$$\mathbb{E}[(\text{II})] \leq \varepsilon.$$

For (III), we need to bridge from $u$ to $0$. For notational simplicity, we denote $\mathbf{z}_u := \boldsymbol{\Phi}_{t \to u}(\mathbf{x}_t)$, $\mathbf{y}_u := \boldsymbol{\Psi}_{t \to u}(\mathbf{x}_t)$. Inserting and subtracting the teacher at $u$, we will get:

$$\|\mathbf{f}_{\boldsymbol{\theta}_{\text{gCD}}}(\mathbf{z}_u, u) - \boldsymbol{\Psi}_{t \to 0}(\mathbf{x}_t)\|$$
$$= \|\mathbf{f}_{\boldsymbol{\theta}_{\text{gCD}}}(\mathbf{z}_u, u) - \boldsymbol{\Psi}_{u \to 0}(\mathbf{y}_u)\|$$
$$\leq \|\mathbf{f}_{\boldsymbol{\theta}_{\text{gCD}}}(\mathbf{z}_u, u) - \mathbf{f}_{\boldsymbol{\theta}_{\text{gCD}}}(\mathbf{y}_u, u)\| + \|\mathbf{f}_{\boldsymbol{\theta}_{\text{gCD}}}(\mathbf{y}_u, u) - \boldsymbol{\Psi}_{u \to 0}(\mathbf{y}_u)\| + \|\boldsymbol{\Psi}_{u \to 0}(\mathbf{y}_u) - \boldsymbol{\Psi}_{u \to 0}(z_u)\|$$
$$=: (\text{IIIa}) + (\text{IIIb}) + (\text{IIIc}).$$

Therefore $(\text{III}) \leq 3 \left( (\text{IIIa})^2 + (\text{IIIb})^2 + (\text{IIIc})^2 \right)$ and, by assumptions

$$\mathbb{E}[(\text{IIIa})^2] \leq \text{Lip}^2(\mathbf{f}_{\boldsymbol{\theta}_{\text{gCD}}}) \mathbb{E}\|\mathbf{z}_u - \mathbf{y}_u\|^2 = \mathcal{O}(\Delta t^{2p}),$$
$$\mathbb{E}[(\text{IIIb})^2] = \delta_u,$$
$$\mathbb{E}[(\text{IIIc})^2] \leq \text{Lip}^2(\boldsymbol{\Psi}) \mathbb{E}\|\mathbf{z}_u - \mathbf{y}_u\|^2 = \mathcal{O}(\Delta t^{2p}).$$

Hence,

$$\mathbb{E}[(\text{III})] = \mathcal{O}(\Delta t^{2p} + \delta_u).$$

We thus conclude that

$$\mathbb{E}_{\boldsymbol{\xi}} \left[ \left\| \mathbf{f}_{\boldsymbol{\theta}_{\text{gCD}}}(\mathbf{x}'_t, t') - \boldsymbol{\Psi}_{t \to 0}(\mathbf{x}_t) \right\|_2^2 \right] = \mathcal{O}(\Delta t^2 + \varepsilon + \Delta t^{2p} + \delta_u) = \mathcal{O}(\Delta t^2 + \varepsilon + \delta_u),$$

as $p \geq 1$.

**Case 4. Random Initialization:**

$$\mathbb{E}_{\boldsymbol{\xi}} \|\mathbf{f}_{\boldsymbol{\theta}_{\text{rand.}}}(\mathbf{x}'_t, t') - \boldsymbol{\Psi}_{t \to 0}(\mathbf{x}_t)\|_2^2$$
$$\lesssim \mathbb{E}_{\boldsymbol{\xi}} \|\mathbf{f}_{\boldsymbol{\theta}_{\text{rand.}}}(\mathbf{x}'_t, t') - \mathbf{f}_{\boldsymbol{\theta}_{\text{rand.}}}(\mathbf{x}_t, t)\|_2^2 + \mathbb{E}_{\boldsymbol{\xi}} \|\mathbf{f}_{\boldsymbol{\theta}_{\text{rand.}}}(\mathbf{x}_t, t)\|_2^2 + \mathbb{E}_{\boldsymbol{\xi}} \|\boldsymbol{\Psi}_{t \to 0}(\mathbf{x}_t)\|_2^2$$
$$= \mathcal{O}(\Delta t^2) + \mathcal{O}(1) = \hspace{8cm} \mathcal{O}(1).$$

$$\square$$

## F.5 ANALYSIS OF GRADIENT VARIANCE

Following the same setup as in Appendix F.4, we focus on the case where the distance is given by

$$d(\mathbf{x}, \mathbf{y}) := \|\mathbf{x} - \mathbf{y}\|_2^2, \qquad w(t) \equiv 1,$$

and the CM flow map is denoted by $\boldsymbol{\Psi}_{t \to 0}$. The general case can be obtained analogously.

For notational simplicity, let

$$\boldsymbol{\xi} := (t, \mathbf{x}_t) \sim \text{Unif}[0, T] \times p_t.$$

The gradient variance with respective to $\boldsymbol{\xi}$ of the expected loss $\ell_{\text{CM}}(\boldsymbol{\theta}; \boldsymbol{\xi})$ is given by

$$\mathcal{V}(\boldsymbol{\theta}) := \text{Var}_{\boldsymbol{\xi}}[\nabla_{\boldsymbol{\theta}} \ell_{\text{CM}}(\boldsymbol{\theta}; \boldsymbol{\xi})] = \mathbb{E}_{\boldsymbol{\xi}} \left[ \left\| \nabla_{\boldsymbol{\theta}} \ell_{\text{CM}}(\boldsymbol{\theta}; \boldsymbol{\xi}) - \mathbb{E}_{\boldsymbol{\xi}}[\nabla_{\boldsymbol{\theta}} \ell_{\text{CM}}(\boldsymbol{\theta}; \boldsymbol{\xi})] \right\|_2^2 \right] = \text{Tr}\left( \text{Cov}_{\boldsymbol{\xi}} \left[ \nabla_{\boldsymbol{\theta}} \ell_{\text{CM}}(\boldsymbol{\theta}; \boldsymbol{\xi}) \right] \right).$$

**Theorem F.2.** *Under the same assumptions as in Theorem F.1. The following upper bounds on the variances hold for different initialization schemes:*

1. **CMT :** $\mathcal{V}(\boldsymbol{\theta}_{\mathrm{CMT}}) = \mathcal{O}(\varepsilon + \Delta t^2)$

2. **Diffusion Model:** $\mathcal{V}(\boldsymbol{\theta}_{\mathrm{DM}}) = \min\{\mathcal{O}(\varepsilon), \mathcal{O}(\Delta t^2)\}$.

3. **General Consistency Distillation:** $\mathcal{V}(\boldsymbol{\theta}_{\mathrm{gCD}}) = \mathcal{O}(\varepsilon + \Delta t^2)$

4. **Random Initialization:** $\mathcal{V}(\boldsymbol{\theta}_{\mathrm{rand.}}) = \mathcal{O}(1)$.

*Proof.* To analyze the variance, we observe that

$$\mathcal{V}(\boldsymbol{\theta}) = \mathbb{E}_{\boldsymbol{\xi}}\left[\|\nabla_{\boldsymbol{\theta}}\ell_{\mathrm{CM}}(\boldsymbol{\theta},\boldsymbol{\xi})\|_2^2\right] - \|\mathbb{E}_{\boldsymbol{\xi}}[\nabla_{\boldsymbol{\theta}}\ell_{\mathrm{CM}}(\boldsymbol{\theta},\boldsymbol{\xi})]\|_2^2 \le \mathbb{E}_{\boldsymbol{\xi}}\left[\|\nabla_{\boldsymbol{\theta}}\ell_{\mathrm{CM}}(\boldsymbol{\theta},\boldsymbol{\xi})\|_2^2\right]$$

We compute the gradient of the loss in the gradient variance formula as:

$$\nabla_{\boldsymbol{\theta}}\ell_{\mathrm{CM}}(\boldsymbol{\theta},\boldsymbol{\xi}) = 2\cdot\mathbf{e}(\boldsymbol{\theta})^\top \cdot \nabla_{\boldsymbol{\theta}}\mathbf{f}_{\boldsymbol{\theta}}(\mathbf{x}_t, t),$$

where we define the error vector:

$$\mathbf{e}(\boldsymbol{\theta}) := \mathbf{f}_{\boldsymbol{\theta}}(\mathbf{x}_t, t) - \mathbf{f}_{\boldsymbol{\theta}^-}(\mathbf{x}_{t-\Delta t}, t - \Delta t) \in \mathbb{R}^D.$$

Now we bound the second moment by using $\|\mathbf{A}^\top\mathbf{u}\|_2 \le \|\mathbf{A}\|_F\|\mathbf{u}\|_2$ with $\|\mathbf{A}\|_F$ denoting the Frobenius norm of the matrix $\mathbf{A}$, we will get

$$\|\nabla_{\boldsymbol{\theta}}\ell_{\mathrm{CM}}(\boldsymbol{\theta};\boldsymbol{\xi})\|_2^2 = 4\|\left(\nabla_{\boldsymbol{\theta}}\mathbf{f}_{\boldsymbol{\theta}}(\mathbf{x}_t, t)\right)^\top\mathbf{e}(\boldsymbol{\theta})\|_2^2 \le 4\|\nabla_{\boldsymbol{\theta}}\mathbf{f}_{\boldsymbol{\theta}}(\mathbf{x}_t, t)\|_F^2\|\mathbf{e}(\boldsymbol{\theta})\|_2^2.$$

Therefore,

$$\mathcal{V}(\boldsymbol{\theta}) \le 4\mathbb{E}_{\boldsymbol{\xi}}\left[\|\nabla_{\boldsymbol{\theta}}\mathbf{f}_{\boldsymbol{\theta}}(\mathbf{x}_t, t)\|_F^2\|\mathbf{e}(\boldsymbol{\theta})\|_2^2\right].$$

From the assumption that $\|\nabla_{\boldsymbol{\theta}}\mathbf{f}_{\boldsymbol{\theta}}(\mathbf{x}_t, t)\|_F \le M$ almost surely, then

$$\mathcal{V}(\boldsymbol{\theta}) \le 4M^2\mathbb{E}_{\boldsymbol{\xi}}\left[\|\mathbf{e}(\boldsymbol{\theta})\|_2^2\right].$$

We now bound $\mathbb{E}_{\boldsymbol{\xi}}[\|\mathbf{e}(\boldsymbol{\theta})\|_2^2]$ under the four different initializations.

**Case 1. CMT :** Let $\boldsymbol{\Phi}_{t\to s}$ be a $p$-th order solver for the PF-ODE built from a fixed drift, and define the forward–backward (round-trip) map

$$\tilde{\mathbf{x}}_t := \boldsymbol{\Phi}_{T\to t}\left(\boldsymbol{\Psi}_{t\to T}(\mathbf{x}_t)\right), \qquad \tilde{\mathbf{x}}_{t-\Delta t} := \boldsymbol{\Phi}_{T\to t-\Delta t}\left(\boldsymbol{\Psi}_{t-\Delta t\to T}(\mathbf{x}_{t-\Delta t})\right).$$

Insert solver's round trips in $\mathbf{e}_{\boldsymbol{\theta}_{\mathrm{CMT}}}$:

$$\mathbf{e}_{\boldsymbol{\theta}_{\mathrm{CMT}}} = \left(\mathbf{f}_{\boldsymbol{\theta}_{\mathrm{CMT}}}(\mathbf{x}_t, t) - \mathbf{f}_{\boldsymbol{\theta}_{\mathrm{CMT}}}(\tilde{\mathbf{x}}_t, t)\right) - \left(\mathbf{f}_{\boldsymbol{\theta}_{\mathrm{CMT}}}(\mathbf{x}_{t-\Delta t}, t - \Delta t) - \mathbf{f}_{\boldsymbol{\theta}_{\mathrm{CMT}}}(\tilde{\mathbf{x}}_{t-\Delta t}, t - \Delta t)\right)$$
$$+ \left(\mathbf{f}_{\boldsymbol{\theta}_{\mathrm{CMT}}}(\tilde{\mathbf{x}}_t, t) - \mathbf{f}_{\boldsymbol{\theta}_{\mathrm{CMT}}}(\tilde{\mathbf{x}}_{t-\Delta t}, t - \Delta t)\right).$$

Thus, we have

$$\mathbb{E}\|\mathbf{e}_{\boldsymbol{\theta}_{\mathrm{CMT}}}\|_2^2 \lesssim \mathrm{Lip}(\mathbf{f}_{\boldsymbol{\theta}_{\mathrm{CMT}}})^2\left(\|\mathbf{x}_t - \tilde{\mathbf{x}}_t\|_2^2 + \|\mathbf{x}_{t-\Delta t} - \tilde{\mathbf{x}}_{t-\Delta t}\|_2^2\right) + \mathbb{E}\|C_t\|_2^2, \qquad (14)$$

where $C_t := \mathbf{f}_{\boldsymbol{\theta}_{\mathrm{CMT}}}(\tilde{\mathbf{x}}_t, t) - \mathbf{f}_{\boldsymbol{\theta}_{\mathrm{CMT}}}(\tilde{\mathbf{x}}_{t-\Delta t}, t - \Delta t)$.

To address $C_t$ term, we anchor $C_t$ to the solver of teacher. Set $\mathbf{S}_t(\mathbf{x}) := \boldsymbol{\Phi}_{T\to 0}(\boldsymbol{\Psi}_{t\to T}(\mathbf{x}))$. Then $\mathbf{S}_t(\mathbf{x}_t) := \boldsymbol{\Phi}_{T\to 0}(\boldsymbol{\Psi}_{t\to T}(\mathbf{x}_t))$, and $\mathbf{S}_{t-\Delta t}(\mathbf{x}_{t-\Delta t}) := \boldsymbol{\Phi}_{T\to 0}(\boldsymbol{\Psi}_{t-\Delta t\to T}(\mathbf{x}_{t-\Delta t}))$. We decompose $C_t$ as the following:

$$C_t = \underbrace{\left(\mathbf{f}_{\boldsymbol{\theta}_{\mathrm{CMT}}}(\tilde{\mathbf{x}}_t, t) - \mathbf{S}_t(\mathbf{x}_t)\right)}_{(a)} - \underbrace{\left(\mathbf{f}_{\boldsymbol{\theta}_{\mathrm{CMT}}}(\tilde{\mathbf{x}}_{t-\Delta t}, t - \Delta t) - \mathbf{S}_{t-\Delta t}(\mathbf{x}_{t-\Delta t})\right)}_{(b)} + \underbrace{\left(\mathbf{S}_t(\mathbf{x}_t) - \mathbf{S}_{t-\Delta t}(\mathbf{x}_{t-\Delta t})\right)}_{(c)}.$$

Because $p_T = p_{\mathrm{prior}}$, $\tilde{\mathbf{x}}_t = \boldsymbol{\Phi}_{T\to t}(\mathbf{x}_T)$ with $\mathbf{x}_T \sim p_{\mathrm{prior}}$, so

$$\mathbb{E}\|(a)\|_2^2 \le \varepsilon, \qquad \mathbb{E}\|(b)\|_2^2 \le \varepsilon.$$

Now, we control the teacher drift.

$$\|(c)\|_2 = \|\mathbf{S}_t(\mathbf{x}_t) - \mathbf{S}_{t-\Delta t}(\mathbf{x}_{t-\Delta t})\|_2$$
$$\le \mathrm{Lip}(\boldsymbol{\Phi})\|\boldsymbol{\Psi}_{t\to T}(\mathbf{x}_t)) - \boldsymbol{\Psi}_{t-\Delta t\to T}(\mathbf{x}_{t-\Delta t}))\|_2,$$
$$\le \mathrm{Lip}(\boldsymbol{\Phi})\mathrm{Lip}(\boldsymbol{\Psi})\left(\|\mathbf{x}_t - \mathbf{x}_{t-\Delta t}\|_2 + |\Delta t|\right),$$

so we have $\mathbb{E}\|(\mathrm{c})\|_2^2 = \mathcal{O}(\Delta t^2)$.

Combining the above bounds, we conclude:

$$\mathbb{E}_{t,\mathbf{x}_0,\boldsymbol{\epsilon}}\big[\|\mathbf{e}_{\boldsymbol{\theta}_{\mathrm{CMT}}}\|_2^2\big] = \mathcal{O}\left(\varepsilon + \Delta t^{2p} + \Delta t^2\right).$$

**Case 2. Diffusion Model:**

**Bound I: Training–Error Only; No Smoothness.** Write

$$\mathbf{e}_{\boldsymbol{\theta}_{\mathrm{DM}}} = \Big(\mathbf{f}_{\boldsymbol{\theta}_{\mathrm{DM}}}(\mathbf{x}_t, t) - \mathbf{x}_0\Big) - \Big(\mathbf{f}_{\boldsymbol{\theta}_{\mathrm{DM}}}(\mathbf{x}_{t-\Delta t}, t - \Delta t) - \mathbf{x}_0\Big).$$

By $\|u - v\|^2 \le 2\|u\|^2 + 2\|v\|^2$ and taking expectation over $(t, \mathbf{x}_t, \mathbf{x}_{t-\Delta t})$,

$$\mathbb{E}\big[\|\mathbf{e}_{\boldsymbol{\theta}_{\mathrm{DM}}}\|_2^2\big] \le 2\mathbb{E}\big[\|\mathbf{f}_{\boldsymbol{\theta}_{\mathrm{DM}}}(\mathbf{x}_t, t) - \mathbf{x}_0\|_2^2\big] + 2\mathbb{E}\big[\|\mathbf{f}_{\boldsymbol{\theta}_{\mathrm{DM}}}(\mathbf{x}_{t-\Delta t}, t - \Delta t) - \mathbf{x}_0\|_2^2\big] \le 4\varepsilon,$$

where the last inequality uses the same training distribution for $(t, \mathbf{x}_t)$ and $(t - \Delta t, \mathbf{x}_{t-\Delta t})$ (e.g., $t$ uniform on $[\Delta t, 1]$). Thus

$$\mathbb{E}\big[\|\mathbf{e}_{\boldsymbol{\theta}_{\mathrm{DM}}}\|_2^2\big] \le 4\varepsilon.$$

**Bound II: Lipschitz Smoothness; $\Delta t$–Sensitive.** Assume $\mathbf{f}_{\boldsymbol{\theta}_{\mathrm{DM}}}$ is Lipschitz in state and time:

$$\|\mathbf{f}_{\boldsymbol{\theta}_{\mathrm{DM}}}(\mathbf{x}, t) - \mathbf{f}_{\boldsymbol{\theta}_{\mathrm{DM}}}(\mathbf{y}, s)\|_2 \le \mathrm{Lip}(\mathbf{f}_{\boldsymbol{\theta}_{\mathrm{DM}}})\left(\|\mathbf{x} - \mathbf{y}\|_2 + |t - s|\right).$$

Then

$$\|\mathbf{e}_{\boldsymbol{\theta}_{\mathrm{DM}}}\|_2 \le \|\mathbf{f}_{\boldsymbol{\theta}_{\mathrm{DM}}}(\mathbf{x}_t, t) - \mathbf{f}_{\boldsymbol{\theta}_{\mathrm{DM}}}(\mathbf{x}_{t-\Delta t}, t)\|_2 + \|\mathbf{f}_{\boldsymbol{\theta}_{\mathrm{CMT}}}(\mathbf{x}_{t-\Delta t}, t) - \mathbf{f}_{\boldsymbol{\theta}_{\mathrm{CMT}}}(\mathbf{x}_{t-\Delta t}, t - \Delta t)\|_2$$
$$\le \mathrm{Lip}(\mathbf{f}_{\boldsymbol{\theta}_{\mathrm{DM}}})\left(\|\mathbf{x}_t - \mathbf{x}_{t-\Delta t}\|_2 + |\Delta t|\right),$$

hence by $(a + b)^2 \le 2a^2 + 2b^2$,

$$\|\mathbf{e}_{\boldsymbol{\theta}_{\mathrm{DM}}}\|_2^2 \lesssim \mathrm{Lip}^2(\mathbf{f}_{\boldsymbol{\theta}_{\mathrm{DM}}})\left(\|\mathbf{x}_t - \mathbf{x}_{t-\Delta t}\|_2^2 + \Delta t^2\right).$$

Taking expectation and using the coupled forward process,

$$\mathbf{x}_t - \mathbf{x}_{t-\Delta t} = (a_t - a_{t-\Delta t})\mathbf{x}_0 + (b_t - b_{t-\Delta t})\boldsymbol{\epsilon},$$

so with $m_2 := \mathbb{E}\|\mathbf{x}_0\|_2^2$ and $\mathbb{E}\|\boldsymbol{\epsilon}\|_2^2 = D$ (and $\mathbb{E}[\mathbf{x}_0^\top \boldsymbol{\epsilon}] = 0$),

$$\mathbb{E}\|\mathbf{x}_t - \mathbf{x}_{t-\Delta t}\|_2^2 = \mathcal{O}(\Delta t^2).$$

Therefore,

$$\mathbb{E}\big[\|\mathbf{e}_{\boldsymbol{\theta}_{\mathrm{DM}}}\|_2^2\big] = \mathcal{O}(\Delta t^2).$$

Taking the better of the two regimes yields

$$\mathbb{E}_{t,\mathbf{x}_0,\boldsymbol{\epsilon}}\big[\|\mathbf{e}_{\boldsymbol{\theta}_{\mathrm{DM}}}\|_2^2\big] \lesssim \min\left\{\varepsilon,\ \Delta t^2\right\}.$$

(If one averages over $t$, insert $\mathbb{E}_t[\cdot]$ on the second term's bracket; if one prefers a uniform-in-$t$ bound, replace the bracket by its $\sup_t$.)

**Case 3. General Consistency Distillation:** With

$$\mathbf{e}_{\boldsymbol{\theta}_{\mathrm{gCD}}} = \underbrace{[\mathbf{f}_{\boldsymbol{\theta}_{\mathrm{CMT}}}(\mathbf{x}_t, t) - \mathbf{f}_{\boldsymbol{\theta}_{\mathrm{CMT}}}(\boldsymbol{\Phi}_{t\to u}(\mathbf{x}_t), u)]}_{A_t} - \underbrace{[\mathbf{f}_{\boldsymbol{\theta}_{\mathrm{CMT}}}(\mathbf{x}_{t-\Delta t}, t - \Delta t) - \mathbf{f}_{\boldsymbol{\theta}_{\mathrm{CMT}}}(\boldsymbol{\Phi}_{t-\Delta t\to u}(\mathbf{x}_{t-\Delta t}), u)]}_{B_{t-\Delta t}}$$
$$+ \underbrace{[\mathbf{f}_{\boldsymbol{\theta}_{\mathrm{CMT}}}(\boldsymbol{\Phi}_{t\to u}(\mathbf{x}_t), u) - \mathbf{f}_{\boldsymbol{\theta}_{\mathrm{CMT}}}(\boldsymbol{\Phi}_{t-\Delta t\to u}(\mathbf{x}_{t-\Delta t}), u)]}_{C_{t,u}}$$

by the gCD assumption, we have

$$\mathbb{E}\|A_t\|^2 = \mathcal{L}_{\mathrm{gCD}}(\boldsymbol{\theta}_{\mathrm{gCD}}; u) < \varepsilon, \qquad \mathbb{E}\|B_{t-\Delta t}\|^2 \le \varepsilon,$$

so that

$$3\mathbb{E}\|A_t\|^2 + 3\mathbb{E}\|B_{t-\Delta t}\|^2 \le 6\varepsilon.$$

For $C_{t,u}$, using the Lipschitz properties,

$$\|C_{t,u}\| \leq \mathrm{Lip}(\mathbf{f}_{\boldsymbol{\theta}_{\mathrm{gCD}}})\big(\|\boldsymbol{\Phi}_{t\to u}(\mathbf{x}_t) - \boldsymbol{\Phi}_{t\to u}(\mathbf{x}_{t-\Delta t})\| + \|\boldsymbol{\Phi}_{t\to u}(\mathbf{x}_{t-\Delta t}) - \boldsymbol{\Phi}_{t-\Delta t\to u}(\mathbf{x}_{t-\Delta t})\|\big)$$

$$\leq \mathrm{Lip}(\mathbf{f}_{\boldsymbol{\theta}_{\mathrm{gCD}}})\mathrm{Lip}(\boldsymbol{\Phi})\big(\|\mathbf{x}_t - \mathbf{x}_{t-\Delta t}\| + \Delta t\big),$$

hence

$$\mathbb{E}\|C_{t,u}\|^2 \leq 2\mathrm{Lip}^2(\mathbf{f}_{\boldsymbol{\theta}_{\mathrm{gCD}}})\mathrm{Lip}^2(\boldsymbol{\Phi})\Big(\mathbb{E}\|\mathbf{x}_t - \mathbf{x}_{t-\Delta t}\|^2 + \Delta t^2\Big) = \mathcal{O}\big(\Delta t^2\big).$$

Combining the pieces,

$$\mathbb{E}_{t,\mathbf{x}_0,\boldsymbol{\epsilon}}\big[\|\mathbf{e}_{\boldsymbol{\theta}_{\mathrm{gCD}}}\|_2^2\big] = \mathcal{O}\big(\varepsilon + \Delta t^2\big).$$

**Case 4. Random Initialization:** This is a straightforward derivation from the assumption:

$$\mathbb{E}\|\mathbf{e}(\boldsymbol{\theta}_{\mathrm{rand.}})\|_2^2 \lesssim \mathbb{E}\|\mathbf{f}_{\boldsymbol{\theta}_{\mathrm{rand.}}}(\mathbf{x}_t,t)\|_2^2 + \mathbb{E}\|\mathbf{f}_{\boldsymbol{\theta}_{\mathrm{rand.}}}(\mathbf{x}_{t-\Delta t},t-\Delta t)\|_2^2 \lesssim 2R.$$

$\square$

### F.6 BIAS–VARIANCE DECOMPOSITION

For the squared *gradient bias* and the CM's flow map gradient variance

$$\mathcal{B}(\boldsymbol{\theta}) := \big\|\nabla_{\boldsymbol{\theta}}\bar{\ell}_{\mathrm{oracle}}(\boldsymbol{\theta}) - \nabla_{\boldsymbol{\theta}}\bar{\ell}_{\mathrm{CM}}(\boldsymbol{\theta})\big\|_2^2 \quad \mathcal{V}(\boldsymbol{\theta}) := \mathrm{Var}_{\boldsymbol{\xi}}\big[\nabla_{\boldsymbol{\theta}}\ell_{\mathrm{CM}}(\boldsymbol{\theta};\boldsymbol{\xi})\big].$$

Consider the oracle–relative mean–squared error (MSE) of a CM gradient:

$$\mathcal{E}(\boldsymbol{\theta}) := \mathbb{E}_{\boldsymbol{\xi}}\Big[\big\|\nabla_{\boldsymbol{\theta}}\ell_{\mathrm{CM}}(\boldsymbol{\theta};\boldsymbol{\xi}) - \nabla_{\boldsymbol{\theta}}\bar{\ell}_{\mathrm{oracle}}(\boldsymbol{\theta})\big\|_2^2\Big].$$

Then we have the following mean-squared errors comparison under the four different initializations:

**Corollary F.3.** *Under the same assumptions as in Theorem F.1, the following CM gradient MSE bounds hold for the four initialization schemes ($\boldsymbol{\theta} = \boldsymbol{\theta}_{\mathrm{CMT}}, \boldsymbol{\theta}_{\mathrm{DM}}, \boldsymbol{\theta}_{\mathrm{gCD}}, \boldsymbol{\theta}_{\mathrm{rand}}$).*

  *(i)* **CMT** *:*

$$\mathcal{E}(\boldsymbol{\theta}_{\mathrm{CMT}}) = \mathcal{O}\left(\varepsilon + \Delta t^2 + \Delta t^p\right).$$

  *(ii)* **Diffusion Model:**

$$\mathcal{E}(\boldsymbol{\theta}_{\mathrm{DM}}) = \mathcal{O}\left(\varepsilon + \Delta t^2 + \mathbb{E}_t\left[\frac{\sigma_t^2}{\alpha_t^2}\right]\right) + \mathbb{E}_{\mathbf{x}_t,t}\left[\big\|\boldsymbol{\Psi}_{t\to 0}(\mathbf{x}_t) - \mathbb{E}[\mathbf{x}_0|\mathbf{x}_t]\big\|_2^2\right].$$

  *(iii)* **General Consistency Distillation:** *For a fixed $u \in [0,T]$, assume in addition that*

$$\delta_u := \mathbb{E}_{\mathbf{x}_u \sim p_u}\big\|\mathbf{f}_{\boldsymbol{\theta}_{\mathrm{gCD}}}(\mathbf{x}_u,u) - \boldsymbol{\Psi}_{u\to 0}(\mathbf{x}_u)\big\|^2 < \infty.$$

  *Then*

$$\mathcal{E}(\boldsymbol{\theta}_{\mathrm{gCD}}) = \mathcal{O}\left(\varepsilon + \Delta t^2 + \delta_u\right).$$

  *(iv)* **Random Initialization:**

$$\mathcal{E}(\boldsymbol{\theta}_{\mathrm{rand.}}) = \mathcal{O}(1).$$

*Proof.* Write the (vector) bias as

$$\mathbf{b}(\boldsymbol{\theta}) := \nabla_{\boldsymbol{\theta}}\bar{\ell}_{\mathrm{CM}}(\boldsymbol{\theta}) - \nabla_{\boldsymbol{\theta}}\bar{\ell}_{\mathrm{oracle}}(\boldsymbol{\theta}).$$

Then

$$\mathcal{E}(\boldsymbol{\theta}) = \mathbb{E}_{\boldsymbol{\xi}}\Big[\big\|\nabla_{\boldsymbol{\theta}}\ell_{\mathrm{CM}}(\boldsymbol{\theta};\boldsymbol{\xi}) - \nabla_{\boldsymbol{\theta}}\bar{\ell}_{\mathrm{CM}}(\boldsymbol{\theta}) + \mathbf{b}(\boldsymbol{\theta})\big\|_2^2\Big]$$

$$= \mathbb{E}_{\boldsymbol{\xi}}\Big[\big\|\nabla_{\boldsymbol{\theta}}\ell_{\mathrm{CM}}(\boldsymbol{\theta};\boldsymbol{\xi}) - \nabla_{\boldsymbol{\theta}}\bar{\ell}_{\mathrm{CM}}(\boldsymbol{\theta})\big\|_2^2\Big] + \underbrace{\|\mathbf{b}(\boldsymbol{\theta})\|_2^2}_{=\,\mathcal{B}(\boldsymbol{\theta})} + 2\mathbb{E}_{\boldsymbol{\xi}}\Big[\big\langle\nabla_{\boldsymbol{\theta}}\ell_{\mathrm{CM}}(\boldsymbol{\theta};\boldsymbol{\xi}) - \nabla_{\boldsymbol{\theta}}\bar{\ell}_{\mathrm{CM}}(\boldsymbol{\theta}), \mathbf{b}(\boldsymbol{\theta})\big\rangle\Big].$$

The cross term vanishes because $\mathbb{E}_{\boldsymbol{\xi}}\big[\nabla_{\boldsymbol{\theta}}\ell_{\mathrm{CM}}(\boldsymbol{\theta};\boldsymbol{\xi}) - \nabla_{\boldsymbol{\theta}}\bar{\ell}_{\mathrm{CM}}(\boldsymbol{\theta})\big] = \mathbf{0}$, hence

$$\mathcal{E}(\boldsymbol{\theta}) = \underbrace{\mathrm{Tr}\Big(\mathrm{Cov}_{\boldsymbol{\xi}}\big[\nabla_{\boldsymbol{\theta}}\ell_{\mathrm{CM}}(\boldsymbol{\theta};\boldsymbol{\xi})\big]\Big)}_{=\,\mathcal{V}(\boldsymbol{\theta})} + \underbrace{\big\|\nabla_{\boldsymbol{\theta}}\bar{\ell}_{\mathrm{oracle}}(\boldsymbol{\theta}) - \nabla_{\boldsymbol{\theta}}\bar{\ell}_{\mathrm{CM}}(\boldsymbol{\theta})\big\|_2^2}_{=\,\mathcal{B}(\boldsymbol{\theta})}.$$

The remaining steps follow directly by combining the results of Theorems F.1 and F.2. $\square$

F.7 COMPARISON ON OPTIMIZATION DYNAMICS

We consider plain SGD on the oracle objective using CM gradients. The iteration is

$$\boldsymbol{\theta}_{k+1} = \boldsymbol{\theta}_k - \eta \nabla_{\boldsymbol{\theta}} \ell_{\mathrm{CM}}(\boldsymbol{\theta}_k; \boldsymbol{\xi}_k), \qquad \boldsymbol{\xi}_k \sim \mathrm{Unif}[0, T] \times p_t \text{ i.i.d.},$$

with constant stepsize $\eta > 0$. The expected oracle loss $\bar{\ell}_{\mathrm{oracle}}$ is assumed $L$–smooth and to satisfy a Polyak–Łojasiewicz (PL) condition on the level set visited by the iterates:

$$\|\nabla_{\boldsymbol{\theta}} \bar{\ell}_{\mathrm{oracle}}(\boldsymbol{\theta}) - \nabla_{\boldsymbol{\theta}} \bar{\ell}_{\mathrm{oracle}}(\boldsymbol{\theta}')\|_2 \le L\|\boldsymbol{\theta} - \boldsymbol{\theta}'\|_2, \qquad \frac{1}{2}\|\nabla_{\boldsymbol{\theta}} \bar{\ell}_{\mathrm{oracle}}(\boldsymbol{\theta})\|_2^2 \ge \mu(\bar{\ell}_{\mathrm{oracle}}(\boldsymbol{\theta}) - \bar{\ell}^*)$$

for some $\mu > 0$. Here,

$$\bar{\ell}^* := \min_{\boldsymbol{\theta}} \bar{\ell}_{\mathrm{oracle}}(\boldsymbol{\theta}) = \min_{\boldsymbol{\theta}} \left[ \mathbb{E}_{t, \mathbf{x}_t} \|\mathbf{f}_{\boldsymbol{\theta}}(\mathbf{x}_t, t) - \boldsymbol{\Psi}_{t \to 0}(\mathbf{x}_t)\|_2^2 \right].$$

We use the bias $\mathcal{B}(\boldsymbol{\theta})$, variance $\mathcal{V}(\boldsymbol{\theta})$, and MSE

$$\mathcal{E}(\boldsymbol{\theta}) := \mathbb{E}_{\boldsymbol{\xi}}\left[ \left\| \nabla_{\boldsymbol{\theta}} \ell_{\mathrm{CM}}(\boldsymbol{\theta}; \boldsymbol{\xi}) - \nabla_{\boldsymbol{\theta}} \bar{\ell}_{\mathrm{oracle}}(\boldsymbol{\theta}) \right\|_2^2 \right] = \mathcal{B}(\boldsymbol{\theta}) + \mathcal{V}(\boldsymbol{\theta})$$

as established above. We assume the stepsize satisfies $\eta \le 1/(4L)$.

**Theorem F.4** (SGD Analysis with Scheme-Specific Initializations). *Assume the conditions of Theorem F.1, and further assume that $\bar{\ell}_{\mathrm{oracle}}$ is $L$–smooth, that the PL($\mu$) condition holds, that the stepsize satisfies $\eta \le 1/(4L)$, and that $\mathcal{E}$ is Lipschitz with constant $\mathrm{Lip}(\mathcal{E})$ on the level set visited by SGD. Let $p \ge 1$ be the global order of the ODE solver used in pretraining/teacher flows. For each initialization scheme, define*

$$A_0 := \bar{\ell}_{\mathrm{oracle}}(\boldsymbol{\theta}_0) - \bar{\ell}^*, \qquad M_0 := \mathcal{E}(\boldsymbol{\theta}_0).$$

*Then, for any $K \ge 1$,*

$$\mathbb{E}[\bar{\ell}_{\mathrm{oracle}}(\boldsymbol{\theta}_K) - \bar{\ell}^*] \le (1 - \mu\eta)^K A_0 + \frac{5}{\mu} \mathrm{Lip}(\mathcal{E}) \sqrt{\eta K A_0} + \frac{5}{2\mu} M_0 + \frac{35}{4\mu} \mathrm{Lip}(\mathcal{E})^2 \eta^2 K^2. \quad (15)$$

*Let $C(\eta, K) := \frac{35}{4\mu} \mathrm{Lip}(\mathcal{E})^2 \eta^2 K^2$. Then the initialization lemma (Lemma F.4) and the bias–variance/MSE-at-init bounds assumed in Theorem F.1 imply the following scheme-specific orders:*

- **CMT :**
$$A_0 = \mathcal{O}(\varepsilon + \Delta t^{2p}), \qquad M_0 = \mathcal{O}(\varepsilon + \Delta t^2 + \Delta t^p).$$
*Thus,*
$$\mathbb{E}[\bar{\ell}_{\mathrm{oracle}}(\boldsymbol{\theta}_K) - \bar{\ell}^*] \le (1 - \mu\eta)^K \mathcal{O}(\varepsilon + \Delta t^{2p}) + \sqrt{\eta K} \mathcal{O}((\varepsilon + \Delta t^{2p})^{1/2})$$
$$+ \mathcal{O}(\varepsilon + \Delta t^2 + \Delta t^p) + C(\eta, K).$$

- **Diffusion Model (DM):** *We denote*
$$\mathcal{M}_{\mathrm{DM}} := \mathbb{E}_{t, \mathbf{x}_t} \|\boldsymbol{\Psi}_{t \to 0}(\mathbf{x}_t) - \mathbb{E}[\mathbf{x}_0 | \mathbf{x}_t]\|_2^2,$$
*the deterministic–map versus posterior–mean mismatch. Then*
$$A_0 = \mathcal{O}(\varepsilon) + \mathcal{M}_{\mathrm{DM}}, \qquad M_0 = \mathcal{O}\left(\varepsilon + \Delta t^2 + \mathbb{E}_t\left[\frac{\sigma_t^2}{\alpha_t^2}\right]\right) + \mathcal{M}_{\mathrm{DM}}.$$
*Thus,*
$$\mathbb{E}[\bar{\ell}_{\mathrm{oracle}}(\boldsymbol{\theta}_K) - \bar{\ell}^*] \le (1 - \mu\eta)^K \left(\mathcal{O}(\varepsilon) + \mathcal{M}_{\mathrm{DM}}\right) + \sqrt{\eta K} \left(\mathcal{O}(\varepsilon) + \mathcal{M}_{\mathrm{DM}}\right)^{1/2}$$
$$+ \left(\mathcal{O}\left(\varepsilon + \Delta t^2 + \mathbb{E}_t\left[\frac{\sigma_t^2}{\alpha_t^2}\right]\right) + \mathcal{M}_{\mathrm{DM}}\right) + C(\eta, K).$$

- **General Consistency Distillation (gCD):**
$$A_0 = \mathcal{O}(\varepsilon + \delta_u + \Delta t^{2p}), \qquad M_0 = \mathcal{O}(\varepsilon + \Delta t^2 + \delta_u).$$
*Thus,*
$$\mathbb{E}[\bar{\ell}_{\mathrm{oracle}}(\boldsymbol{\theta}_K) - \bar{\ell}^*] \le (1 - \mu\eta)^K \mathcal{O}(\varepsilon + \delta_u + \Delta t^{2p}) + \sqrt{\eta K} \mathcal{O}((\varepsilon + \delta_u + \Delta t^{2p})^{1/2})$$
$$+ \mathcal{O}(\varepsilon + \Delta t^2 + \delta_u) + C(\eta, K).$$

- ***Random initialization:***

$$A_0 = \mathcal{O}(1), \qquad M_0 = \mathcal{O}(1).$$

*Thus,*

$$\mathbb{E}\big[\bar{\ell}_{\text{oracle}}(\boldsymbol{\theta}_K) - \bar{\ell}^*\big] \leq (1 - \mu\eta)^K \, \mathcal{O}(1) + \sqrt{\eta K} \, \mathcal{O}(1)$$
$$+ \mathcal{O}(1) + C(\eta, K).$$

*All big-$\mathcal{O}$ constants are independent of $\Delta t$, $\varepsilon$, and $K$.*

All schemes enjoy the same geometric contraction factor $(1 - \mu\eta)^K$; differences arise solely through the initialization terms $A_0$ and $M_0$. Among them, CMT achieves

$$\mathbb{E}\big[\bar{\ell}_{\text{oracle}}(\boldsymbol{\theta}_K) - \bar{\ell}^*\big] \leq (1 - \mu\eta)^K \, \mathcal{O}(\varepsilon + \Delta t^{2p}) + \sqrt{\eta K} \, \mathcal{O}\big((\varepsilon + \Delta t^{2p})^{1/2}\big) + \mathcal{O}(\varepsilon + \Delta t^2 + \Delta t^p) + C(\eta, K),$$

which contains no extra irreducible terms (such as $\mathcal{M}_{\text{DM}}$ or $\delta_u$). Consequently, while the asymptotic rate is identical across schemes, CMT attains the smallest excess risk (the tightest bound and lowest floor) for any $K$, up to the common term $C(\eta, K)$.

The bound on $M_0$ in Equation (15) for each initialization scheme follows directly from Theorem F.1. To obtain a complete upper bound needed for the proof of Theorem F.4, however, we also require bounds on $A_0$ for the four initialization schemes in Equation (15). In Lemma F.4, we establish such bounds for each $A_0$. We then return to finalize the proof of Theorem F.4.

**Initialization Excess Oracle Risk.** We bound the initial oracle excess risk $\bar{\ell}_{\text{oracle}}(\boldsymbol{\theta}_0) - \bar{\ell}^*$ for each of the four initialization schemes. We now state and prove the initialization bounds.

**Lemma F.4.** *Under Assumptions B and C and following the notations therein, there exist constants $C_1, C_2 < \infty$ that do not depend on $\Delta t$ or $\varepsilon$ such that*

$$\bar{\ell}_{\text{oracle}}(\boldsymbol{\theta}_{\text{CMT}}) - \bar{\ell}^* \leq 2\,\varepsilon + C_1 \, \Delta t^{\,2p},$$
$$\bar{\ell}_{\text{oracle}}(\boldsymbol{\theta}_{\text{DM}}) - \bar{\ell}^* \leq 2\,\varepsilon + 2\,\mathcal{M}_{\text{DM}},$$
$$\bar{\ell}_{\text{oracle}}(\boldsymbol{\theta}_{\text{gCD}}) - \bar{\ell}^* \leq 2\,\varepsilon + 9\,\delta_u + C_2 \, \Delta t^{\,2p},$$
$$\bar{\ell}_{\text{oracle}}(\boldsymbol{\theta}_{\text{rand.}}) - \bar{\ell}^* \leq 2R + 2C_{\boldsymbol{\Psi}}.$$

*Here any fixed choice suffices since they are absorbed in $C_2$ in the final rates. In the realizable case $\bar{\ell}^* = 0$, these are direct bounds on the initialization oracle loss.*

*Proof.* **CMT.** Let $\mathbf{x}_T \sim p_{\text{prior}}$, $\mathbf{x}_t := \boldsymbol{\Psi}_{T \to t}(\mathbf{x}_T)$, $\tilde{\mathbf{x}}_t := \texttt{Solver}_{T \to t}(\mathbf{x}_T)$, and $\tilde{\mathbf{x}}_0 := \texttt{Solver}_{T \to 0}(\mathbf{x}_T)$. For fixed $t$,

$$\big\|\mathbf{f}_{\boldsymbol{\theta}_{\text{CMT}}}(\mathbf{x}_t, t) - \boldsymbol{\Psi}_{t \to 0}(\mathbf{x}_t)\big\|_2 \leq \underbrace{\big\|\mathbf{f}_{\boldsymbol{\theta}_{\text{CMT}}}(\mathbf{x}_t, t) - \mathbf{f}_{\boldsymbol{\theta}_{\text{CMT}}}(\tilde{\mathbf{x}}_t, t)\big\|_2}_{A_1} + \underbrace{\big\|\mathbf{f}_{\boldsymbol{\theta}_{\text{CMT}}}(\tilde{\mathbf{x}}_t, t) - \tilde{\mathbf{x}}_0\big\|_2}_{A_2}$$
$$+ \underbrace{\big\|\tilde{\mathbf{x}}_0 - \boldsymbol{\Psi}_{t \to 0}(\tilde{\mathbf{x}}_t)\big\|_2}_{A_3} + \underbrace{\big\|\boldsymbol{\Psi}_{t \to 0}(\tilde{\mathbf{x}}_t) - \boldsymbol{\Psi}_{t \to 0}(\mathbf{x}_t)\big\|_2}_{A_4}.$$

By Assumption B, $A_1 \leq \text{Lip}(\mathbf{f}_{\boldsymbol{\theta}_{\text{CMT}}})\|\tilde{\mathbf{x}}_t - \mathbf{x}_t\|_2$. Using the semigroup $\boldsymbol{\Psi}_{t \to 0} \circ \boldsymbol{\Psi}_{T \to t} = \boldsymbol{\Psi}_{T \to 0}$ and triangle inequality, $A_3 \leq \|\tilde{\mathbf{x}}_0 - \boldsymbol{\Psi}_{T \to 0}(\mathbf{x}_T)\|_2 + \|\boldsymbol{\Psi}_{t \to 0}(\tilde{\mathbf{x}}_t) - \boldsymbol{\Psi}_{t \to 0}(\mathbf{x}_t)\|_2$. By Assumption C, $A_4 \leq \text{Lip}(\boldsymbol{\Psi})\|\tilde{\mathbf{x}}_t - \mathbf{x}_t\|_2$ and the second term in $A_3$ is also $\leq \text{Lip}(\boldsymbol{\Psi})\|\tilde{\mathbf{x}}_t - \mathbf{x}_t\|_2$. Now apply $(a + b)^2 \leq 2a^2 + 2b^2$ to split $(A_1 + A_2 + A_3 + A_4)^2$ into $2A_2^2 + 2(A_1 + A_3 + A_4)^2$, then again $(u + v + w)^2 \leq 3(u^2 + v^2 + w^2)$ on the second group to obtain

$$\bar{\ell}_{\text{oracle}}(\boldsymbol{\theta}_{\text{CMT}}) \leq 2\, \mathbb{E}_{t, \mathbf{x}_T}\Big[\underbrace{\big\|\mathbf{f}_{\boldsymbol{\theta}_{\text{CMT}}}(\tilde{\mathbf{x}}_t, t) - \tilde{\mathbf{x}}_0\big\|_2^2}_{\leq\, \varepsilon} + C'\,\|\tilde{\mathbf{x}}_t - \mathbf{x}_t\|_2^2 + \big\|\tilde{\mathbf{x}}_0 - \boldsymbol{\Psi}_{T \to 0}(\mathbf{x}_T)\big\|_2^2\Big],$$

with $C' := 3(\text{Lip}^2(\mathbf{f}_{\boldsymbol{\theta}_{\text{CMT}}}) + 2\text{Lip}^2(\boldsymbol{\Psi}))$. Since the solver is of order $p$, the last two expectations are $O(\Delta t^{\,2p})$. Therefore

$$\bar{\ell}_{\text{oracle}}(\boldsymbol{\theta}_{\text{CMT}}) \lesssim 2\,\varepsilon + 2\,(C' + 1)\,\Delta t^{\,2p}.$$

Absorbing constants into $C_1$ and subtracting $\bar{\ell}^*$ gives the claim.

**Diffusion Model (DM).** By the tower property, for any $\mathbf{h}(\mathbf{x}_t)$,

$$\mathbb{E}\|\mathbf{x}_0 - \mathbf{h}(\mathbf{x}_t)\|_2^2 = \mathbb{E}\|\mathbf{x}_0 - \mathbb{E}[\mathbf{x}_0|\mathbf{x}_t]\|_2^2 + \mathbb{E}\|\mathbf{h}(\mathbf{x}_t) - \mathbb{E}[\mathbf{x}_0|\mathbf{x}_t]\|_2^2.$$

With $\mathbf{h} = \mathbf{D}_{\boldsymbol{\theta}_{\mathrm{DM}}}$ and $\mathcal{L}_{\mathrm{DM}}(\boldsymbol{\theta}_{\mathrm{DM}}) < \varepsilon$, $\mathbb{E}\|\mathbf{D}_{\boldsymbol{\theta}_{\mathrm{DM}}} - \mathbb{E}[\mathbf{x}_0|\mathbf{x}_t]\|_2^2 \leq \varepsilon$. Thus by $\|a - b\|^2 \leq 2\|a - c\|^2 + 2\|c - b\|^2$ with $c = \mathbb{E}[\mathbf{x}_0|\mathbf{x}_t]$,

$$\bar{\ell}_{\mathrm{oracle}}(\boldsymbol{\theta}_{\mathrm{DM}}) = \mathbb{E}\big\|\mathbf{D}_{\boldsymbol{\theta}_{\mathrm{DM}}}(\mathbf{x}_t, t) - \boldsymbol{\Psi}_{t\to 0}(\mathbf{x}_t)\big\|_2^2 \leq 2\,\varepsilon + 2\,\mathcal{M}_{\mathrm{DM}},$$

and subtracting $\bar{\ell}^*$ yields the stated bound.

**General Consistency Distillation (gCD).** Fix $u \in [0, T]$. Let $\tilde{\mathbf{x}}_u := \texttt{Solver}_{t\to u}(\mathbf{x}_t)$ and $\mathbf{x}_u := \boldsymbol{\Psi}_{t\to u}(\mathbf{x}_t)$. Define $\mathbf{F}_u(\mathbf{z}) := \mathbf{f}_{\boldsymbol{\theta}_{\mathrm{gCD}}}(\mathbf{z}, u) - \boldsymbol{\Psi}_{u\to 0}(\mathbf{z})$, which is $(\mathrm{Lip}(\mathbf{f}_{\boldsymbol{\theta}_{\mathrm{gCD}}}) + \mathrm{Lip}(\boldsymbol{\Psi}))$–Lipschitz by Assumption C and Assumption B. Then for fixed $t$,

$$\big\|\mathbf{f}_{\boldsymbol{\theta}_{\mathrm{gCD}}}(\mathbf{x}_t, t) - \boldsymbol{\Psi}_{t\to 0}(\mathbf{x}_t)\big\|_2 \leq \underbrace{\big\|\mathbf{f}_{\boldsymbol{\theta}_{\mathrm{gCD}}}(\mathbf{x}_t, t) - \mathbf{f}_{\boldsymbol{\theta}_{\mathrm{gCD}}}(\tilde{\mathbf{x}}_u, u)\big\|_2}_{B_1} + \underbrace{\big\|\mathbf{F}_u(\tilde{\mathbf{x}}_u)\big\|_2}_{B_2}$$
$$+ \underbrace{\big\|\boldsymbol{\Psi}_{u\to 0}(\tilde{\mathbf{x}}_u) - \boldsymbol{\Psi}_{u\to 0}(\mathbf{x}_u)\big\|_2}_{B_3},$$

using the semigroup $\boldsymbol{\Psi}_{t\to 0} = \boldsymbol{\Psi}_{u\to 0} \circ \boldsymbol{\Psi}_{t\to u}$. By $(a+b)^2 \leq 2a^2 + 2b^2$ and then $(b+c)^2 \leq 2b^2 + 2c^2$,

$$\bar{\ell}_{\mathrm{oracle}}(\boldsymbol{\theta}_{\mathrm{gCD}}) \leq 2\,\mathbb{E}B_1^2 + 4\,\mathbb{E}B_2^2 + 4\,\mathbb{E}B_3^2.$$

The first term is controlled by the training loss: $\mathbb{E}B_1^2 = \mathcal{L}_{\mathrm{gCD}}(\boldsymbol{\theta}_{\mathrm{gCD}}; u) < \varepsilon$. For $B_2$, by $\|a + b\|^2 \leq (1 + \rho)\|a\|^2 + (1 + 1/\rho)\|b\|^2$ with $a = \mathbf{F}_u(\mathbf{x}_u)$, $b = \mathbf{F}_u(\tilde{\mathbf{x}}_u) - \mathbf{F}_u(\mathbf{x}_u)$,

$$\mathbb{E}B_2^2 \leq (1 + \rho)\,\mathbb{E}\|F_u(\mathbf{x}_u)\|_2^2 + (1 + 1/\rho)\,(\mathrm{Lip}(\mathbf{f}_{\boldsymbol{\theta}_{\mathrm{gCD}}}) + \mathrm{Lip}(\boldsymbol{\Psi}))^2\,\mathbb{E}\|\tilde{\mathbf{x}}_u - \mathbf{x}_u\|_2^2.$$

Choosing, e.g., $\rho = 1$ and recalling $\delta_u := \mathbb{E}_{\mathbf{x}_u \sim p_u}\|\mathbf{F}_u(\mathbf{x}_u)\|_2^2$, $p$-th order solver (see Assumption C) gives

$$\mathbb{E}B_2^2 \lesssim 2\,\delta_u + 2\,(\mathrm{Lip}(\mathbf{f}_{\boldsymbol{\theta}_{\mathrm{gCD}}}) + \mathrm{Lip}(\boldsymbol{\Psi}))^2\,\Delta t^{\,2p}.$$

For $B_3$, by Assumption C and Assumption C, $\mathbb{E}B_3^2 \lesssim \mathrm{Lip}^2(\boldsymbol{\Psi})\,\Delta t^{\,2p}$. Combining,

$$\bar{\ell}_{\mathrm{oracle}}(\boldsymbol{\theta}_{\mathrm{gCD}}) \lesssim 2\,\varepsilon + 8\,\delta_u + \underbrace{4\Big(2(\mathrm{Lip}(\mathbf{f}_{\boldsymbol{\theta}_{\mathrm{gCD}}}) + \mathrm{Lip}^2(\boldsymbol{\Psi})) + \mathrm{Lip}^2(\boldsymbol{\Psi})\Big)}_{C_2}\,\Delta t^{\,2p}.$$

**Random Initialization.** By $\|a - b\|^2 \leq 2\|a\|^2 + 2\|b\|^2$ and Assumption C,

$$\bar{\ell}_{\mathrm{oracle}}(\boldsymbol{\theta}_{\mathrm{rand.}}) = \mathbb{E}\big\|\mathbf{f}_{\boldsymbol{\theta}_{\mathrm{rand.}}}(\mathbf{x}_t, t) - \boldsymbol{\Psi}_{t\to 0}(\mathbf{x}_t)\big\|_2^2 \leq 2\,\mathbb{E}\|\mathbf{f}_{\boldsymbol{\theta}_{\mathrm{rand.}}}(\mathbf{x}_t, t)\|_2^2 + 2\,\mathbb{E}\|\boldsymbol{\Psi}_{t\to 0}(\mathbf{x}_t)\|_2^2 \leq 2R + 2C_{\boldsymbol{\Psi}}.$$

Subtracting $\bar{\ell}^*$ in each case yields the claims. $\qquad\square$

**Proof of Theorem F.4.**

*Proof.* **Linear Contraction to an MSE floor.** By the descent lemma for $L$–smooth $\bar{\ell}_{\mathrm{oracle}}$,

$$\bar{\ell}_{\mathrm{oracle}}(\boldsymbol{\theta}_{k+1}) \leq \bar{\ell}_{\mathrm{oracle}}(\boldsymbol{\theta}_k) - \eta\langle\nabla\bar{\ell}_{\mathrm{oracle}}(\boldsymbol{\theta}_k), \nabla_{\boldsymbol{\theta}}\ell_{\mathrm{CM}}(\boldsymbol{\theta}_k; \boldsymbol{\xi}_k)\rangle + \frac{L\eta^2}{2}\big\|\nabla_{\boldsymbol{\theta}}\ell_{\mathrm{CM}}(\boldsymbol{\theta}_k; \boldsymbol{\xi}_k)\big\|_2^2.$$

Taking conditional expectation w.r.t. $\boldsymbol{\xi}_k$, using $\mathbb{E}_{\boldsymbol{\xi}_k}\nabla_{\boldsymbol{\theta}}\ell_{\mathrm{CM}}(\boldsymbol{\theta}_k; \boldsymbol{\xi}_k) = \nabla_{\boldsymbol{\theta}}\bar{\ell}_{\mathrm{CM}}(\boldsymbol{\theta}_k)$ and $\mathbb{E}_{\boldsymbol{\xi}_k}\|\nabla_{\boldsymbol{\theta}}\ell_{\mathrm{CM}}(\boldsymbol{\theta}_k; \boldsymbol{\xi}_k)\|_2^2 = \|\nabla_{\boldsymbol{\theta}}\bar{\ell}_{\mathrm{CM}}(\boldsymbol{\theta}_k)\|_2^2 + \mathrm{Tr}\,\mathrm{Cov}_{\boldsymbol{\xi}_k}[\nabla_{\boldsymbol{\theta}}\ell_{\mathrm{CM}}(\boldsymbol{\theta}_k; \boldsymbol{\xi}_k)]$, together with

$$\big\langle\nabla\bar{\ell}_{\mathrm{oracle}}, \nabla\bar{\ell}_{\mathrm{CM}}\big\rangle \geq \|\nabla\bar{\ell}_{\mathrm{oracle}}\|_2^2 - \|\nabla\bar{\ell}_{\mathrm{oracle}}\|_2\big\|\nabla\bar{\ell}_{\mathrm{CM}} - \nabla\bar{\ell}_{\mathrm{oracle}}\big\|_2,$$

and Young's inequality, we obtain

$$\mathbb{E}\big[\bar{\ell}_{\mathrm{oracle}}(\boldsymbol{\theta}_{k+1}) \mid \boldsymbol{\theta}_k\big] \leq \bar{\ell}_{\mathrm{oracle}}(\boldsymbol{\theta}_k) - \frac{\eta}{2}\|\nabla\bar{\ell}_{\mathrm{oracle}}(\boldsymbol{\theta}_k)\|_2^2 + \frac{5}{4}\eta\mathcal{E}(\boldsymbol{\theta}_k),$$

where we also used $\eta \leq 1/(4L)$ to absorb the $L\eta^2$ terms into the constants.

Applying the PL inequality to eliminate the gradient norm yields

$$\mathbb{E}\big[\bar{\ell}_{\text{oracle}}(\boldsymbol{\theta}_{k+1}) - \bar{\ell}^* \mid \boldsymbol{\theta}_k\big] \le \big(1 - \mu\eta\big)\big(\bar{\ell}_{\text{oracle}}(\boldsymbol{\theta}_k) - \bar{\ell}^*\big) + \frac{5}{4}\eta\mathcal{E}(\boldsymbol{\theta}_k).$$

Taking total expectation and unrolling the recursion gives, for any $K \ge 1$,

$$\mathbb{E}\big[\bar{\ell}_{\text{oracle}}(\boldsymbol{\theta}_K) - \bar{\ell}^*\big] \le (1 - \mu\eta)^K\big(\bar{\ell}_{\text{oracle}}(\boldsymbol{\theta}_0) - \bar{\ell}^*\big) + \frac{5}{4}\eta \sum_{k=0}^{K-1}(1 - \mu\eta)^{K-1-k}\mathbb{E}\big[\mathcal{E}(\boldsymbol{\theta}_k)\big]. \quad (16)$$

The bounds in Corollary F.3 provide $\mathcal{E}(\boldsymbol{\theta})$ only at initialization for each scenario. Thus, we may need more additional assumptions.

First, a *localized MSE stability* assumption: there exists a neighborhood $\mathcal{N}$ of the initialization in which the same order bound holds for $\mathcal{E}(\boldsymbol{\theta})$, and the SGD trajectory remains in $\mathcal{N}$ under $\eta \le 1/(4L)$. Then $\sup_{0 \le k \le K-1} \mathbb{E}[\mathcal{E}(\boldsymbol{\theta}_k)] \le \overline{\mathcal{E}}$ with $\overline{\mathcal{E}}$ of the same order as the initialization, which recovers the floor bound

$$\mathbb{E}\big[\bar{\ell}_{\text{oracle}}(\boldsymbol{\theta}_K) - \bar{\ell}^*\big] \le (1 - \mu\eta)^K\big(\bar{\ell}_{\text{oracle}}(\boldsymbol{\theta}_0) - \bar{\ell}^*\big) + \frac{5}{4}\frac{\overline{\mathcal{E}}}{\mu}.$$

In this case, similar bounds can be obtained by applying Lemma F.4 to different initialization schemes of $\bar{\ell}_{\text{oracle}}(\boldsymbol{\theta}_0) - \bar{\ell}^*$.

Second, a *mild continuity control*: suppose $\mathcal{E}$ is Lipschitz on the level set visited by the iterates, i.e.,

$$\big|\mathcal{E}(\boldsymbol{\theta}) - \mathcal{E}(\boldsymbol{\theta}')\big| \le \text{Lip}(\mathcal{E})\|\boldsymbol{\theta} - \boldsymbol{\theta}'\|_2.$$

If, in addition, the step size ensures a bounded path length $\sum_{k=0}^{K-1}\mathbb{E}\|\boldsymbol{\theta}_{k+1} - \boldsymbol{\theta}_k\|_2 \le R$ (which follows from $\mathbb{E}\|\boldsymbol{\theta}_{k+1} - \boldsymbol{\theta}_k\|_2 = \eta\mathbb{E}\|\nabla_{\boldsymbol{\theta}}\ell_{\text{CM}}(\boldsymbol{\theta}_k; \boldsymbol{\xi}_k)\|_2$ and the same descent argument that bounds the average oracle gradient norm), then

$$\sup_{0 \le k \le K-1} \mathbb{E}\big[\mathcal{E}(\boldsymbol{\theta}_k)\big] \le \mathcal{E}(\boldsymbol{\theta}_0) + \text{Lip}(\mathcal{E})R.$$

Inserting this into Equation (16) gives a data–dependent version in terms of the initialization MSE plus a controllable growth term.

**Proof of the Mild Continuity Control.** Assume that $\mathcal{E}$ is Lipschitz on the level set visited by $\{\boldsymbol{\theta}_k\}$:

$$\big|\mathcal{E}(\boldsymbol{\theta}) - \mathcal{E}(\boldsymbol{\theta}')\big| \le \text{Lip}(\mathcal{E})\|\boldsymbol{\theta} - \boldsymbol{\theta}'\|_2.$$

Fix a horizon $K \ge 1$. By a telescoping argument and Jensen's inequality,

$$\sup_{0 \le k \le K-1} \mathbb{E}\big[\mathcal{E}(\boldsymbol{\theta}_k)\big] \le \mathcal{E}(\boldsymbol{\theta}_0) + \text{Lip}(\mathcal{E})\mathbb{E}\Big[\sum_{j=0}^{K-1}\|\boldsymbol{\theta}_{j+1} - \boldsymbol{\theta}_j\|_2\Big] \le \mathcal{E}(\boldsymbol{\theta}_0) + \text{Lip}(\mathcal{E})\sum_{j=0}^{K-1}\mathbb{E}\big[\|\boldsymbol{\theta}_{j+1} - \boldsymbol{\theta}_j\|_2\big].$$

Since $\boldsymbol{\theta}_{j+1} - \boldsymbol{\theta}_j = -\eta\nabla_{\boldsymbol{\theta}}\ell_{\text{CM}}(\boldsymbol{\theta}_j; \boldsymbol{\xi}_j)$,

$$\mathbb{E}\big[\|\boldsymbol{\theta}_{j+1} - \boldsymbol{\theta}_j\|_2\big] = \eta\mathbb{E}\big[\|\nabla_{\boldsymbol{\theta}}\ell_{\text{CM}}(\boldsymbol{\theta}_j; \boldsymbol{\xi}_j)\|_2\big] \le \eta\sqrt{\mathbb{E}\big[\|\nabla_{\boldsymbol{\theta}}\ell_{\text{CM}}(\boldsymbol{\theta}_j; \boldsymbol{\xi}_j)\|_2^2\big]}.$$

Using $\mathbb{E}\|\nabla_{\boldsymbol{\theta}}\ell_{\text{CM}}\|_2^2 = \|\nabla_{\boldsymbol{\theta}}\bar{\ell}_{\text{CM}}\|_2^2 + \text{Tr}\,\Sigma(\boldsymbol{\theta}_j)$ and

$$\|\nabla_{\boldsymbol{\theta}}\bar{\ell}_{\text{CM}}\|_2^2 \le 2\|\nabla_{\boldsymbol{\theta}}\bar{\ell}_{\text{oracle}}\|_2^2 + 2\|\nabla_{\boldsymbol{\theta}}\bar{\ell}_{\text{CM}} - \nabla_{\boldsymbol{\theta}}\bar{\ell}_{\text{oracle}}\|_2^2,$$

we get

$$\mathbb{E}\big[\|\nabla_{\boldsymbol{\theta}}\ell_{\text{CM}}(\boldsymbol{\theta}_j; \boldsymbol{\xi}_j)\|_2^2\big]$$
$$\le 2\mathbb{E}\big[\|\nabla_{\boldsymbol{\theta}}\bar{\ell}_{\text{oracle}}(\boldsymbol{\theta}_j)\|_2^2\big] + 2\mathbb{E}\big[\|\nabla_{\boldsymbol{\theta}}\bar{\ell}_{\text{CM}}(\boldsymbol{\theta}_j) - \nabla_{\boldsymbol{\theta}}\bar{\ell}_{\text{oracle}}(\boldsymbol{\theta}_j)\|_2^2\big] + \mathbb{E}\big[\text{Tr}\,\Sigma(\boldsymbol{\theta}_j)\big]$$
$$\le 2\mathbb{E}\big[\|\nabla_{\boldsymbol{\theta}}\bar{\ell}_{\text{oracle}}(\boldsymbol{\theta}_j)\|_2^2\big] + 2\mathbb{E}\big[\mathcal{E}(\boldsymbol{\theta}_j)\big].$$

Therefore, by Cauchy–Schwarz,

$$\sum_{j=0}^{K-1}\mathbb{E}\big[\|\boldsymbol{\theta}_{j+1} - \boldsymbol{\theta}_j\|_2\big] \le \eta\sum_{j=0}^{K-1}\sqrt{2\mathbb{E}\big[\|\nabla_{\boldsymbol{\theta}}\bar{\ell}_{\text{oracle}}(\boldsymbol{\theta}_j)\|_2^2\big] + 2\mathbb{E}\big[\mathcal{E}(\boldsymbol{\theta}_j)\big]}$$

$$\le \eta\sqrt{K}\Bigg(\sum_{j=0}^{K-1}\Big(2\mathbb{E}\big[\|\nabla_{\boldsymbol{\theta}}\bar{\ell}_{\text{oracle}}(\boldsymbol{\theta}_j)\|_2^2\big] + 2\mathbb{E}\big[\mathcal{E}(\boldsymbol{\theta}_j)\big]\Big)\Bigg)^{1/2}. \quad (17)$$

Summing the one–step decrease inequality

$$\mathbb{E}\big[\bar{\ell}_{\mathrm{oracle}}(\boldsymbol{\theta}_j)\big] - \mathbb{E}\big[\bar{\ell}_{\mathrm{oracle}}(\boldsymbol{\theta}_{j+1})\big] \geq \frac{\eta}{2}\mathbb{E}\big[\|\nabla_{\boldsymbol{\theta}}\bar{\ell}_{\mathrm{oracle}}(\boldsymbol{\theta}_j)\|_2^2\big] - \frac{5}{4}\eta\mathbb{E}\big[\mathcal{E}(\boldsymbol{\theta}_j)\big]$$

from $j = 0$ to $K-1$ and using $\mathbb{E}[\bar{\ell}_{\mathrm{oracle}}(\boldsymbol{\theta}_K)] \geq \bar{\ell}^*$ yields

$$\sum_{j=0}^{K-1} \mathbb{E}\big[\|\nabla_{\boldsymbol{\theta}}\bar{\ell}_{\mathrm{oracle}}(\boldsymbol{\theta}_j)\|_2^2\big] \leq \frac{2}{\eta}\big(\bar{\ell}_{\mathrm{oracle}}(\boldsymbol{\theta}_0) - \bar{\ell}^*\big) + \frac{5}{2}\sum_{j=0}^{K-1} \mathbb{E}\big[\mathcal{E}(\boldsymbol{\theta}_j)\big].$$

Let $\overline{\mathcal{E}}_K := \sup_{0 \leq j \leq K-1} \mathbb{E}[\mathcal{E}(\boldsymbol{\theta}_j)]$. Then

$$\sum_{j=0}^{K-1} \mathbb{E}\big[\|\nabla_{\boldsymbol{\theta}}\bar{\ell}_{\mathrm{oracle}}(\boldsymbol{\theta}_j)\|_2^2\big] \leq \frac{2}{\eta}\big(\bar{\ell}_{\mathrm{oracle}}(\boldsymbol{\theta}_0) - \bar{\ell}^*\big) + \frac{5}{2}K\overline{\mathcal{E}}_K.$$

Substituting this and $\sum_j \mathbb{E}[\mathcal{E}(\boldsymbol{\theta}_j)] \leq K\overline{\mathcal{E}}_K$ into Equation (17) gives the path-length bound

$$\sum_{j=0}^{K-1} \mathbb{E}\big[\|\boldsymbol{\theta}_{j+1} - \boldsymbol{\theta}_j\|_2\big] \leq \eta\sqrt{K}\Big(\tfrac{4}{\eta}\big(\bar{\ell}_{\mathrm{oracle}}(\boldsymbol{\theta}_0) - \bar{\ell}^*\big) + 7K\overline{\mathcal{E}}_K\Big)^{1/2} \tag{18}$$

$$\leq 2\sqrt{\eta K\big(\bar{\ell}_{\mathrm{oracle}}(\boldsymbol{\theta}_0) - \bar{\ell}^*\big)} + \sqrt{7}\eta K\overline{\mathcal{E}}_K^{1/2},$$

where the last inequality uses $\sqrt{a+b} \leq \sqrt{a} + \sqrt{b}$.

Combining Lipschitzness of $\mathcal{E}$ and Equation (18) yields

$$\overline{\mathcal{E}}_K \leq \mathcal{E}(\boldsymbol{\theta}_0) + \mathrm{Lip}(\mathcal{E})\Big(2\sqrt{\eta K\big(\bar{\ell}_{\mathrm{oracle}}(\boldsymbol{\theta}_0) - \bar{\ell}^*\big)} + \sqrt{7}\eta K\overline{\mathcal{E}}_K^{1/2}\Big).$$

This is of the form $s \leq u + v\sqrt{s}$ with $s = \overline{\mathcal{E}}_K$, $u = \mathcal{E}(\boldsymbol{\theta}_0) + 2\mathrm{Lip}(\mathcal{E})\sqrt{\eta K(\bar{\ell}_{\mathrm{oracle}}(\boldsymbol{\theta}_0) - \bar{\ell}^*)}$, and $v = \sqrt{7}\mathrm{Lip}(\mathcal{E})\eta K$. The inequality $s \leq u + v\sqrt{s}$ implies $s \leq 2u + v^2$ (complete-the-square argument). Hence

$$\overline{\mathcal{E}}_K \leq 2\mathcal{E}(\boldsymbol{\theta}_0) + 4\mathrm{Lip}(\mathcal{E})\sqrt{\eta K\big(\bar{\ell}_{\mathrm{oracle}}(\boldsymbol{\theta}_0) - \bar{\ell}^*\big)} + 7\mathrm{Lip}(\mathcal{E})^2\eta^2 K^2. \tag{19}$$

Plugging Equation (19) into the pathwise contraction yields

$$\mathbb{E}\big[\bar{\ell}_{\mathrm{oracle}}(\boldsymbol{\theta}_K) - \bar{\ell}^*\big]$$

$$\leq (1-\mu\eta)^K\big(\bar{\ell}_{\mathrm{oracle}}(\boldsymbol{\theta}_0) - \bar{\ell}^*\big) + \frac{5}{4\mu}\Big(2\mathcal{E}(\boldsymbol{\theta}_0) + 4\mathrm{Lip}(\mathcal{E})\sqrt{\eta K\big(\bar{\ell}_{\mathrm{oracle}}(\boldsymbol{\theta}_0) - \bar{\ell}^*\big)} + 7\mathrm{Lip}(\mathcal{E})^2\eta^2 K^2\Big)$$

$$= (1-\mu\eta)^K\big(\bar{\ell}_{\mathrm{oracle}}(\boldsymbol{\theta}_0) - \bar{\ell}^*\big) + \frac{5}{\mu}\mathrm{Lip}(\mathcal{E})\sqrt{\eta K\big(\bar{\ell}_{\mathrm{oracle}}(\boldsymbol{\theta}_0) - \bar{\ell}^*\big)} + \frac{5}{2\mu}\mathcal{E}(\boldsymbol{\theta}_0) + \frac{35}{4\mu}\mathrm{Lip}(\mathcal{E})^2\eta^2 K^2.$$

This is a data–dependent bound in terms of the initialization MSE $\mathcal{E}(\boldsymbol{\theta}_0)$, the loss gap $\bar{\ell}_{\mathrm{oracle}}(\boldsymbol{\theta}_0) - \bar{\ell}^*$, and the Lipschitz constant $\mathrm{Lip}(\mathcal{E})$.

$\square$

