# OpenReview forum: "CMT: Mid-Training for Efficient Learning of Consistency, Mean Flow, and Flow-Map Models"
_ICLR.cc/2026/Conference — ICLR 2026 Poster_

### Official Review · Reviewer_BkLe · 2025-10-21

**Soundness:** 3
**Presentation:** 3
**Contribution:** 3
**Rating:** 4
**Confidence:** 3

**Summary:**

Flow map models such as Consistency Models (CM) and Mean Flow (MF) enable few-step generation by learning the long jump of the ODE solution of diffusion models, yet training remains unstable, sensitive to hyperparameters, and costly. Initializing from a pre-trained diffusion model helps, but still requires converting infinitesimal steps into a long-jump map, leaving instability unresolved. This paper introduces mid-training, the first concept and practical method that inserts a lightweight intermediate stage between the (diffusion) pre-training and the final flow map training (i.e., post-training) for vision generation. Concretely, Consistency Mid-Training (CMT) is a compact and principled stage that trains a model to map points along a solver trajectory from a pre-trained model, starting from a prior sample, directly to the solver-generated clean sample. It yields a trajectory-consistent and stable initialization. This initializer outperforms random and diffusion-based baselines and enables fast, robust convergence without heuristics. Initializing post-training with CMT weights further simplifies flow map learning. Empirically, CMT achieves state of the art two step FIDs: 1.97 on CIFAR-10, 1.32 on ImageNet 64×64, and 1.84 on ImageNet 512×512, while using up to 98% less training data and GPU time, compared to CMs. On ImageNet 256×256, CMT reaches 1-step FID 3.34 while cutting total training time by about 50% compared to MF from scratch (FID 3.43). This establishes CMT as a principled, efficient, and general framework for training flow map models.

**Strengths:**

1. Mitigates issues of training instability, hyperparameter sensitivity, and high training costs that plague models like Consistency Models (CM) and Mean Flow (MF)

2. Proposes a compact, principled intermediate stage (Consistency Mid-Training, CMT) between diffusion pre-training and final flow map post-training, providing trajectory-consistent and stable initialization—filling a critical gap in existing training pipelines.

3. The CMT initializer outperforms random and diffusion-based baselines, enabling fast, robust convergence without relying on heuristics.

**Weaknesses:**

1. Lack of large-scale experiments, e.g., text-to-image, text-to-video. This reduces the credibility of the pipeline.

2. The contribution is quite incremental.

3. I have doubts that the performance gain stems from the unfair experimental setup, since the latent baselines do not use LatentLPIPS in training.

**Questions:**

n/a

---

> ### Author Response · Authors · 2025-11-20
> **Reply to Reviewer BkLe**
>
> We would like to thank Reviewer BkLe for your thoughtful review and the insights you shared. The answers to your questions are summarized below.
>
> ## W1. Regarding large-scale experiments.
>
> ## R1.
> We appreciate the reviewer’s suggestion to further validate our pipeline at larger scales such as text-to-image (T2I) or text-to-video (T2V). We have taken concrete steps in this direction and also clarify the scope and feasibility constraints below.
>
> **New T2I results in the revised manuscript.** In the revised version, we additionally evaluate CMT on the MSCOCO T2I benchmark (please refer to Sec 4.2 of the revised manuscript). To be more precisely on this additional T2I experiments, we evaluate three MF configurations on MS-COCO: (1) **MF (Rand. Init.)**: MF  trained only in the post-training stage from random initialization; (2) **MF (SiT init.)**: SiT pre-training followed by MF post-training initialized from SiT; (3) **CMT**: SiT pre-training, CMT mid-training using SiT as the teacher to generate ODE trajectories for an MF student, then MF post-training from this mid-trained initializer. The table shows that CMT performs strongly on T2I with substantially faster convergence than other initialization schemes: it achieves the best FID while reducing total training time by about 47%. CMT’s 2-step FID (5.01) is also close to the teacher SiT’s 50-step FID (4.73). This additional experiment show that CM can yield strong performance even in T2I, highlighting the generality and effectiveness of CMT across more challenging datasets and architectures.
>
>
>
>
> | Method          | Pre-Training | Mid-Training | Post-Training | Total Time (↓) | 1/2-Step FID (↓)   |
> |----------------|--------------|--------------|---------------|----------------|--------------------|
> | MF (Rand. Init.)   | 0            | 0            | 60 hours      | 60 hours       | 15.97 / 5.42       |
> | MF (SiT Init.) | 22 hours     | 0            | 38 hours      | 60 hours       | 15.55 / 5.26       |
> | **CMT**        | 22 hours     | 1 hour       | 9 hours       | **32 hours**   | **15.12 / 5.01**   |
>
>
>
>
> **Our current scale is already comparable to, or larger than, prior CM-family and MeanFlow works.** CMT is evaluated on ImageNet at 256×256 and 512×512 resolutions, as well as on MSCOCO T2I, which are representative large-scale benchmarks in the literature. To the best of our knowledge, publicly reported CM-family methods whose main novelty lies in principled training schemes (as opposed to large-scale engineering extensions) have been evaluated up to ImageNet 512×512, and MeanFlow up to ImageNet 256×256. Our experiments therefore match or exceed the scale of these prior principled flow map approaches, while consistently improving over them. We believe this provides strong evidence for the effectiveness and generality of CMT at the scale typically used in academic work.
>
> **On truly massive SOTA T2I/T2V models during rebuttal.** We agree that demonstrating performance large-scale T2I/T2V systems would be  interesting. However, we would like to emphasize that the dominant factors behind current SOTA T2I/T2V quality are (i) access to very large, high-quality, proprietary datasets and (ii) extremely large-scale compute (often many GPU-years), together with non-public training and filtering pipelines. More concretely, Table 6 in [1] indicates that large-scale models such as SANA and FLUX are trained on non-open datasets. These resource constraints are beyond what can be reasonably reproduced within a short rebuttal period, and in many cases they make a fair, apples-to-apples comparison impossible. That said, we are currently running additional larger scale T2I experiments (e.g., up to 512×512 on BLIP3o dataset (https://arxiv.org/abs/2505.09568) with larger MeanFlow training configurations (2B parameters)), and we will include any completed results in the updated manuscript during the rebuttal period if feasible, or otherwise in the camera-ready version.
>
>
> In summary, we have expanded our evaluation to MSCOCO T2I and already demonstrate strong performance on large-scale academic benchmarks such as ImageNet 256×256, ImageNet 512×512, and T2I MSCOCO. Given that these settings are on par with or larger than those used in prior CM-family and MeanFlow works, we respectfully believe that the current experimental evidence is sufficient to establish the credibility and practicality of CMT as a principled and scalable pipeline, while leaving truly industrial-scale T2I/T2V deployment as valuable future exploration.
>
>
>
> [1] Tang, B., Zheng, B., Paul, S., & Xie, S. (2025). Exploring the Deep Fusion of Large Language Models and Diffusion Transformers for Text-to-Image Synthesis. In Proceedings of the Computer Vision and Pattern Recognition Conference (pp. 28586-28595).
>
> [2] Wang, F. Y., Huang, Z., Bergman, A., Shen, D., Gao, P., Lingelbach, M., ... & Li, H. (2024). Phased consistency models. Advances in neural information processing systems, 37, 83951-84009.

---

> ### Author Response · Authors · 2025-11-20
> **Reply to Reviewer BkLe**
>
> ## W2. Regarding the contribution.
>
> ## R2.
> We respectfully believe that our contributions are more substantial than incremental for the following reasons.
>
> First, to the best of our knowledge, this is the first work that explicitly introduces and studies a *mid-training* stage for flow–map–based generative models in computer vision (beyond language modeling): a lightweight, low-heuristic, fast-converging training phase that significantly alleviates the training bottleneck. This mid-training stage allows us to rapidly upgrade a given teacher sampler into a much stronger initialization for flow-map learning with very few additional iterations, thereby directly targeting one of the main practical limitations of flow-map models.
>
>
> Second, Theorem 5.1 and Theorem F provide, to the best of our knowledge, the first systematic theoretical analysis of several plausible mid-training strategies for flow map initialization. Our analysis compares different ways of constructing mid-training objectives and shows that the proposed CMT objective enjoys smaller bias with respect to the oracle gradients. These theoretical predictions are supported by the empirical studies (originally in App. C and now moved to Sec. 4.3), where CMT emerges as the simplest and most lightweight choice that attains the best performance among the considered alternatives.
>
> Third, we validate the proposed mid-training pipeline and CMT objective across two representative families of flow map models (CM-family and Mean Flow), including a text-to-image setup. In all cases, we observe consistent gains in performance and training efficiency, indicating that the method is generally applicable rather than tailored to a single architecture or dataset.
>
> Taken together, we believe these conceptual, theoretical, and empirical advances go beyond a minor refinement, and we kindly ask the reviewer to reconsider the assessment that the contribution is merely incremental.

---

> ### Author Response · Authors · 2025-11-20
> **Reply to Reviewer BkLe**
>
> ## W3. Regarding the usage of ELatentLPIPS.
>
> ## R3.
> We first clarify that ELatentLPIPS is used only in the ImageNet 512x512 experiment during the CMT mid-training stage, and nowhere else. The motivation for this design choice and the exact loss have been discussed in Appendix E of the original manuscript or Appendix E of the revised version. It shows that only *CMT's mid-training stage* can benefit from the strong human–perceptual alignment of ELatentLPIPS, whereas applying ELatentLPIPS to CM baselines in post-training stage merely increases training instability and degrades performance. Below, we provide a more detailed analysis of the effect of ELatentLPIPS in the CMT mid-training stage and in the post-training stage of CM methods (specifically ECT).
>
>
> **Mid-training stage with CMT.** To isolate the effect of ELatentLPIPS, we additionally train CMT on ImageNet 512x512 using a purely squared $\ell_2$ loss for the mid-training stage, while keeping all other mid-training hyperparameters unchanged. We then run ECD post-training for 20M images (slightly larger than the original 16M images used with ELatentLPIPS in mid-training stage) and obtain 1/2-step FIDs of 3.32 / 1.86, which are essentially comparable to the CMT results with ELatentLPIPS (3.38 / 1.84).
>
> According to the results, we still obtain a SOTA 2-step FID. The total training cost increases only modestly from 28.8M to 32.8M images, which remains far below the budgets of the CM baselines (AYF: 102.4M, sCT and ECT: 204.8M, ECD and sCD: 409.6M). These results show that CMT already achieves strong performance and a favorable compute–performance trade-off without ELatentLPIPS; the perceptual term mainly provides a mild acceleration of convergence rather than being the source of the performance gain.
>
> **Post-training stage for CM baselines.** We also investigate whether post-training of CM can benefit from ELatentLPIPS in a comparable way. Unlike CMT’s mid-training, which regresses to fixed, high-quality teacher trajectories, the post-training stages of CM methods (such as ECT and ECD) rely on self-generated pseudo-targets from a still-evolving model and therefore use a robust Huber loss to mitigate the instability caused by inaccurate stop-gradient targets.
>
> When we replace the Huber loss with ELatentLPIPS in ECT and ECD on ImageNet 512x512, we observe both degraded performance and increased training instability. With Huber, ECT and ECD achieve 1/2-step FIDs of 9.98/6.28 and 8.47/3.38, respectively. With ELatentLPIPS instead, their FIDs become 10.37/6.39 (ECT) and 8.58/3.41 (ECD), and training becomes more prone to divergence and requires additional tuning to remain stable. Similar observations about the ineffectiveness of ELatentLPIPS for CM-style self-distillation are also reported in https://arxiv.org/abs/2502.01441.
>
>
> Taken together, these ablations indicate that
> - CMT’s improvements do not rely on an “unfair” choice of loss, since CMT remains competitive even with a simple $\ell_2$ mid-training loss under comparable or lower compute.
> - Simply adding ELatentLPIPS to CM post-training stage does not help and can even harm their training dynamics.
>
> We therefore believe the experimental setup is fair, and that the observed performance gains primarily stem from the CMT initialization mechanism rather than from the use of ELatentLPIPS.

---

> ### Author Response · Authors · 2025-11-26
> **A Kind Reminder for Comments**
>
> We sincerely thank Reviewer BkLe for the thorough and constructive feedback. We believe our rebuttal addresses the main concerns through the following concrete clarifications and revisions:
>
> - **Experimental scale and additional T2I evidence:** We clarify that our ImageNet-scale experiments are already comparable to prior CM and MF works under standard academic settings. To further address concerns about text-to-image (T2I), we additionally include new T2I results on MS-COCO. We note that substantially larger-scale T2I/T2V training during rebuttal is difficult to conduct fairly due to limited access to non-public datasets and training pipelines; nevertheless, we are actively working to also report results on *BLIP3o* by the end of rebuttal or include them in the camera-ready version.
> - **Clarified contributions:** We clarify our contributions as:
>   1) introducing the first *mid-training* methodology for flow-map models in vision generation that substantially reduces training budget/time relative to baselines;
>   2) providing the first *systematic theoretical analysis* of how initialization schemes affect flow-map training;
>   3) proposing a *simple*, *general recipe* applicable to two representative flow-map settings: CM-family (learning $t\to0$ jumps) and MF (learning $t\to s$ jumps).
> - **Fairness of CMT Loss:** We clarify that CMT’s gains do *NOT* mainly rely on an "unfair" loss choice. ELatentLPIPS is used *ONLY* in the *ImageNet 512×512 CMT mid-training* stage. Removing ELatentLPIPS and using a plain $L_2$ mid-training loss yields essentially similar FIDs as the original CMT setting, with only a modest increase in training budget (*28.8M → 32.8M images*), still far efficient than typical CM baselines. On the other hand, CM baseline (ECD) is sensitive to loss choice and becomes more unstable and worse if ELatentLPIPS is used.
>
> We appreciate the reviewer’s time and would be happy to address any remaining questions, time permitting.

---

> > ### Author Response · Authors · 2025-12-01
> > **Reply to Reviewer BkLe**
> >
> > We have added **completed** MS-COCO T2I results in the revision, where CMT achieves the best FID while reducing total training time by 47%, and approaches the SiT teacher 50-step (100 NFEs due to CFG in SiT) quality with only 2 steps/NFEs. We also attempted to further extend to a substantially larger-scale T2I setting (e.g., BLIP3o at higher resolution with larger MeanFlow configurations). However, this extension is currently blocked by dataset/model scale explained below.
> >
> > - **Why even larger-scale T2I/T2V is challenging during rebuttal:** Although CMT remains lightweight, efficient, and effective, and CMT  applies to general T2I tasks such as MS-COCO as previously demonstrated, the main challenge lies in the substantially larger training corpus and richer conditioning. Specifically, the 32 million BLIP3o T2I samples (compared with only 1.4 million ImageNet samples), together with their more complex text conditions (as opposed to simple class labels in ImageNet), require significantly longer training to fully absorb the information. In addition, achieving strong performance under these conditions necessitates a larger DiT model. Below, we provide the detailed settings for the T2I experiments.
> >
> > - **Additional experiments and compute constraints.** We followed the REPA-E setup (3B DiT, 32M BLIP3o samples at 512×512) to scale CMT to T2I. However, the required teacher FM pre-training is clearly infeasible within the rebuttal window:
> >
> >   - T2I flow-matching pretraining with REPA-E + DiT on BLIP3o 512×512 needs ≈460–470 H100 GPU days (≈1,400 hours on 8×H100) to obtain a state-of-the-art teacher.
> >   - On ImageNet 256×256, REPA-E + DiT flow matching converges in ≈5 H100 GPU days, and our MF test case already costs ≈30 H100 GPU days.
> >   - The analogous BLIP3o T2I setup is ≈100× more expensive, i.e., ≈3,000 H100 GPU days, which is prohibitive under our current compute budget and time limit.
> >
> >   Given this, we cannot complete full 512×512 BLIP3o T2I within rebuttal, but we plan to downsample BLIP3o to 256×256 and aim to report a 3B DiT + 32M-sample T2I result by the camera-ready deadline.
> >
> >     - **Orthogonal bottleneck that motivates CMT.** This bottleneck is entirely in dataset/model scale, not in CMT itself. In fact, we also observe that training flow-map models (e.g., MeanFlow) from scratch, without a strong FM teacher, can diverge due to unstable early gradients, reinforcing the need for principled mid-training/initialization once a viable teacher exists.
> >
> > Given these constraints, we report the completed MS-COCO T2I results, and treat BLIP3o-scale T2I as ongoing work. Nevertheless, together with our competitive results on ImageNet at 256×256 and 512×512 and the additional MS-COCO T2I evaluation, we believe the current evidence already demonstrates the effectiveness of the CMT pipeline.

---

### Official Review · Reviewer_12GL · 2025-10-26

**Soundness:** 4
**Presentation:** 3
**Contribution:** 4
**Rating:** 8
**Confidence:** 5

**Summary:**

This paper proposes a mid-training stage to stabilize the consistency trajectory model training. The proposed method can achieve competitive sample quality while reducing training cost.

**Strengths:**

* Evaluation is done on multiple datasets, including a high-resolution dataset (ImageNet 512x512).
* The method rivals 2-step FID sCD with a 14x reduction in training cost.
* The idea is simple and easy to implement.

**Weaknesses:**

One weakness is that 1-step FID on ImageNet 512 is worse than sCD by a non-trivial margin (2.28 vs 3.38). It is understandable as CMT uses fewer resources, but can it match the 1-step FID of sCD if given more compute?

Theoretical analysis (Sec. 5) does not offer significant insight. It says that if we train a model to predict the PF-ODE solutions, it will better approximate the PF-ODE solutions, which is straightforward. I suggest moving this part to the appendix and instead using that space to include contents from the appendix (as it contains some valuable insights and experimental details).

Some experimental setups are not clear. See questions.

**Questions:**

Did you use Eq. 6 or Eq. 7 in the experiments?

Did you train ECD with general flow map post-training or special flow map post-training? ECD originally only supports the special flow map.

Isn't Eq. 6 the same as ( L_{\text{var}}^{(1)} ) in L1108? (the slow CMT)

> However, the Slow CMT’s mid-training stage costs 3x more GPU time since generating the regression target in the Slow version is more costly due to the failure to use intermediate points by the ODE-solver.

Do you collect more trajectories for the slow version? Otherwise, the GPU hours should be the same. If we collect the same number of trajectories to match their GPU hours, does the slow version underperform CMT?

( L_{\text{var}}^{(1)} ) (or eq 6) seems the same as 2-rectified flow loss. If we apply improved techniques for 2-rectified flow from [1], can it improve CMT's performance? It seems that [1]'s FID on CIFAR-10 is already pretty similar to CMT's FID, so it may offer even better initialization for the post training.

Typo: in line 1116; it should be ( L_{\text{var}}^{(2)} )?

[1] Lee, Sangyun, Zinan Lin, and Giulia Fanti. "Improving the training of rectified flows."

---

> ### Author Response · Authors · 2025-11-20
> **Reply to Reviewer 12GL**
>
> We would like to thank Reviewer 12GL for your thoughtful review and the insights you shared. The answers to your questions are summarized below.
>
> ## W1. Regarding the 1-step FID gap between sCD and CMT on ImageNet 512.
>
> ## R1.
> **On 1-step FID gap at ImageNet 512 and fair comparison.** We thank the reviewer for highlighting this point. The 1-step ImageNet 512x512 results in Table~3 use ECD [1] as the underlying flow-map backbone for CMT, whereas sCD [2] is built on a continuous-time CM formulation with several carefully tuned heuristics for stability and performance. Thus, the observed FID gap reflects not only differences in training cost, but also differences in the base flow map backbone and training recipe. In addition, the teacher models are not matched: sCD reports results using its own EDM [3] teacher trained with a dedicated recipe, while our CMT experiments rely on a standard open-sourced EDM teacher that has been widely used and validated for reproducibility. For these reasons, we believe that the comparison is not strictly apples-to-apples.
>
>
>
> **CMT as a general mid-training pipeline.** In principle, CMT is orthogonal to the choice of flow map model: one could also insert a CMT mid-training stage on top of the sCD-style flow map, and we expect this would generally improve its 1-step FID as well. However, since there is currently no reliable, official implementation of sCD available, we were not able to reproduce its exact setup, and a direct "same-backbone + more compute" comparison would not be fair. For these reasons, we believe the reported numbers should be interpreted as evidence that CMT can substantially boost a given flow-map backbone at low cost compared to other initialization schemes (e.g., random or pretrained-teacher initialization), rather than as indicating any fundamental limitation that would prevent CMT from matching the 1-step FID of sCD under the same backbone and compute budget.
>
>
> **Empirical takeaway.** Even in this disadvantageous setting, where we use ECD as a weaker flow-map backbone (1/2-step FID 8.47 / 3.38) with a less heavily tuned recipe than sCD (the “E” in ECD stands for “Easy”), inserting a relatively cheap CMT mid-training stage still substantially boosts performance. CMT with an ECD backbone achieves a 2-step FID of 1.84, which is better than sCD (1.88), and a 1-step FID that remains competitive despite the weaker base model (Table 3). CMT+ECD’s 1/2-step FIDs (3.38 / 1.84) are also significantly better than vanilla ECD’s (8.47 / 3.38).
> We therefore view these results as evidence that CMT is an efficient, general pipeline for improving flow map models, rather than as indicating that CMT is inherently unable to match sCD-level 1-step FID when paired with an equally strong backbone and comparable compute.
>
> [1] Geng, Z., Pokle, A., Luo, W., Lin, J., & Kolter, J. Z. Consistency Models Made Easy. In The Thirteenth International Conference on Learning Representations 2025.
>
>
> [2] Lu, C., & Song, Y. Simplifying, Stabilizing and Scaling Continuous-time Consistency Models. In The Thirteenth International Conference on Learning Representations 2025.
>
>
> [3] Karras, T., Aittala, M., Aila, T., & Laine, S. (2022, October). Elucidating the Design Space of Diffusion-Based Generative Models. In Advances in Neural Information Processing Systems.
>
> ## W2. Regarding the organization of theoretical results and supplementary empirical analyses.
>
> ## R2.
> We thank the reviewer for this feedback. We have followed the suggestion by moving more experimental details from the appendix into the main paper and by shortening the theoretical section in Sec. 5. At the same time, we would like to retain a brief presentation of the main theoretical result in the main body, as it provides, to our knowledge, the first systematic analysis that (i) quantifies the irreducible deviation in diffusion-based initialization and (ii) extends to consistency-distillation–based initialization, in a way that aligns with our empirical findings.
>
> If the reviewers find this acceptable, we would like to retain this minimal description of the theoretical analysis in the main body in the proposed revision, as this more concise version strikes a better balance between theory and practice. We are, of course, happy to further condense or relocate material if the reviewers feel that additional changes are necessary.

---

> ### Author Response · Authors · 2025-11-20
> **Reply to Reviewer 12GL**
>
> ## Q1. Regarding Eq. (6) and Eq. (7) (which become Eq. (7) and Eq. (8) in the revised manuscript).
>
>
> ## R1.
> We apologize for the potential confusion and thank the reviewer for raising this point. In all our experiments, Eqs. (7) and (8) (Eqs. (6) and (7) in the original manuscript) are used in Stage 2 of the pipeline, where we train CMT to obtain trajectory-aware weights that directly match the teacher sampler’s outputs. Specifically, Eq. (7) is used for the CM-based CMT objective and Eq. (8) for the MF-based CMT objective. To emphasize, we do not introduce any additional weighting, rescaling, or alternative time-distribution sampling beyond what is written in these equations; the implementation literally minimizes Eqs. (7) and (8) as stated in the paper.
>
> In the revised Sec. 3.1, we now explicitly reference these losses by adding "see Eqs. (7) and (8) below" next to the description of the Stage-2 objective to clarify their usage. We also include algorithm boxes in Appendix B.1 that  summarize the training pipelines for special flow-map post-training (e.g., the CM flow map uses Eq. (7) in mid-training) and general flow-map post-training (e.g., the MeanFlow flow map uses Eq. (8) in mid-training). We hope this clarification resolves the concern, and we would be happy to further refine the exposition if the reviewer has additional suggestions.
>
> ## Q2. Regarding the clarification of flow map post-training.
>
> ## R2.
> We thank the reviewer for pointing out this potential source of confusion and are happy to clarify how ECD is used in our pipeline.
>
> Learning the special flow map (instantiated via ECT/ECD) and learning the general flow map (instantiated via MeanFlow) are two independent and separate tasks.
> In all experiments where ECD appears, we follow the original paper, which aims to learn the special flow map $\Psi_{t \to 0}$, i.e., a jump from any noisy state to a clean sample. That is, ECD is used as a post-training stage for the function that maps any
> $(\mathbf{x}_t, t)$  directly to $\mathbf{x}_0$.
>
> This is exactly as in the original ECD setting, without any modification to support a general $\Psi_{t \to s}$ flow map. When we consider the general flow map $\Psi_{t \to s}$, we instead use MeanFlow (MF) as the post-training instantiation, as described in Section 3.3.
>
> CMT is a general and principled recipe that inserts a mid-training stage to provide a trajectory-aligned initializer, which is then refined either by CM-style post-training (ECT/ECD) for the special flow map $\Psi_{t \to 0}$ or by MF post-training for the general flow map $\Psi_{t \to s}$, depending on the experiment.
>
> For additional clarity, Appendix B.1 includes an algorithm box that summarize the complete training pipeline for both cases: (i) CMT followed by CM-style post-training for the special flow map $\Psi_{t \to 0}$, and (ii) CMT followed by MF post-training for the general flow map $\Psi_{t \to s}$. We will make this distinction more explicit in the revised manuscript to avoid any confusion between the special and general flow-map post-training setups.

---

> ### Author Response · Authors · 2025-11-20
> **Reply to Reviewer 12GL**
>
> ## Q3. Regarding the CMT loss and Slow CMT variant.
>
> ## R3.
> Mathematically, both the CMT loss
>
> $\mathcal{L}_{\text{CMT-CM}}$
>
> in Eq. (7) and the Slow CMT variant
>
> $\mathcal{L}_{\mathrm{var}}^{(1)}$
>
> are discrete-time approximations to the same oracle loss
>
> $\mathcal{L}_{\text{oracle-CM}}$
>
> in Eq. (2). However, they differ crucially in how the teacher-generated supervision targets are constructed. Intuitively, CMT is *data-efficient in teacher calls*: each teacher rollout supervises many student updates, whereas Slow CMT spends a full teacher rollout per update, which quickly becomes impractical as $M$ grows. More precisely,
>
>
> **How the supervision targets are computed.** In CMT (Eq. (7)), we first draw a prior sample, run the teacher ODE solver once from $T$ down to $0$ to obtain a full trajectory, and then use the single endpoint as a shared target for all intermediate states on that trajectory. Thus, one teacher solve yields $M$ supervised pairs.
>
>  In contrast, in Slow CMT (based on $\mathcal{L}_{\mathrm{var}}^{(1)}$), we sample $(\mathbf{x}_t, t)$ and, for each such pair, independently run the teacher solver from $\mathbf{x}_t$ to $t=0$ to obtain $\hat{\mathbf{x}}_0(\mathbf{x}_t)$ with $M$ multiple teacher calls. Each supervised pair $(\mathbf{x}_t, t) \mapsto \hat{\mathbf{x}}_0(\mathbf{x}_t)$ therefore requires its own teacher integration. The two objectives coincide in the continuous-time limit, but their discrete implementations use teacher trajectories in fundamentally different ways.
>
>
> **Computational cost and practicality.**  Because of this difference, the mid-training cost of Slow CMT is significantly higher for a fixed number of supervised pairs. Let $M$ denote the number of discretization steps on $[0,T]$, and let $\mathrm{NFEs}_{\text{traj}}$ be the teacher function evaluations required for one trajectory. In CMT, a single teacher trajectory is amortized over $M$ pairs, so the teacher cost per training pair is
>
> $\mathrm{NFEs}_{\text{traj}}/M = \mathcal{O}(1)$.
>
> In Slow CMT, each training pair requires a fresh teacher solve, so the teacher cost per pair is
>
> $\mathcal{O}(\mathrm{NFEs}_{\text{traj}})$.
>
> Consequently, to train on the same number of supervised pairs, Slow CMT would require roughly a factor of $M$ more teacher evaluations (and thus GPU time) than CMT; conversely, under a fixed teacher-compute budget, Slow CMT can only afford about $1/M$ as many pairs. In our experimental configuration, this leads to an overhead of roughly $3\times$ in wall-clock GPU time, which is what we refer to in the paper when stating that the Slow CMT mid-training stage is about three times more expensive. We did not perform an exhaustive sweep that precisely equalizes GPU hours between the two variants during the review period, but the existing experiments already show that CMT achieves better FID at substantially lower teacher cost, which is exactly the mid-training efficiency advantage we aim to demonstrate. We have clarified the distinction between
>
> $\mathcal{L}_{\text{CMT-CM}}$
>
> and
>
> $\mathcal{L}_{\mathrm{var}}^{(1)}$
>
> and included the above discussion in Appendix C.2 of the revised manuscript.

---

> ### Author Response · Authors · 2025-11-20
> **Reply to Reviewer 12GL**
>
> ## Q4. Regarding the relationship between CMT and 2-Rectified Flow.
>
> ## R4.
> We thank the reviewer for raising this connection. We clarify that $\mathcal{L}_{\mathrm{var}}^{(1)}$ (and current Eq. (7) for CMT) are *not* equivalent to the 2-Rectified Flow loss, for two reasons: (i) what is parameterized, and (ii) what is used as supervision.
>
> In our setting, let
>
> $\mathtt{Solver}_{t\to s}$
>
> denote the proxy of the oracle flow map
>
> $\Psi_{t\to s}$
>
> induced by the pre-trained diffusion teacher (i.e., integration of the PF-ODE).  Both Slow CMT and CMT mathematically have the same goal to train a flow map model
>
>  $\mathbf{f}_\theta(\cdot,t)$
>
>  to match the teacher sampler map $\mathtt{Solver}_{t\to 0}$.
>
> By contrast, 2-Rectified Flow keeps a velocity-field parameterization $\mathbf{v}_\theta(\mathbf{x}, t)$ and learns a new instantaneous drift such that the induced ODE follows a linearly interpolated path between
>
> $(\hat{\mathbf{x}}_0(\mathbf{x}_T), \mathbf{x}_T)$,
>
> where
>
> $\hat{\mathbf{x}}_0(\mathbf{x}_T)$ :=
>
> $\mathtt{Solver}_{T\to 0}(\mathbf{x}_t)$,
>
> obtained by running the teacher ODE solver from the prior to time zero. Its objective is essentially a diffusion-style (flow-matching) loss on the velocity field, not a regression loss on the integrated map $\mathtt{Solver}_{t\to 0}$. Thus, even though both methods may use a teacher, 2-Rectified Flow optimizes a velocity field, whereas CMT optimizes a flow map; the corresponding objectives act on different quantities and are not interchangeable.
>
> Regarding the suggestion to apply improved 2-Rectified Flow techniques to CMT: doing so would first require switching from our flow map model to a diffusion/flow-matching model $\mathbf{v}_\theta$. In theory, one could then insert a 2-Rectified Flow stage into our pipeline, i.e., train a 2-Rectified Flow model as the Stage 2 model of mid-training in Sec. 3.1 and use it as an improved diffusion-style initialization for the flow map post-training in Stage 3. However, our theoretical analysis (Theorem 5.1) shows that any diffusion-based initialization, including this one, inevitably incurs a non-zero discrepancy term of the form
>
> $\mathbb{E}_{t,\mathbf{x}_t}$
>
> $\big[|\Psi_{t\to 0}(\mathbf{x}_t) - \mathbb{E}[\mathbf{x}_0 | \mathbf{x}_t]|_2^2\big]$.
>
> This arises from a fundamental mismatch: a flow map must approximate large integrated jumps along the PF-ODE trajectory, whereas diffusion/flow-matching models control only the *infinitesimal* velocity field. As a result, diffusion-based initialization is inherently limited and cannot remove the fundamental deviation we quantify.
>
> ## Q5. Regarding typo in line 1116.
>
> ## R5.
> We thank the reviewer for pointing out the typo, and we have corrected it in the revised manuscript.
>
>
> [1] Lee, Sangyun, Zinan Lin, and Giulia Fanti. "Improving the training of rectified flows."

---

> > ### Author Response · Authors · 2025-11-26
> > **A Kind Reminder for Comments**
> >
> > We sincerely thank Reviewer 12GL for the thorough and constructive feedback. We believe our rebuttal addresses the main concerns through the following concrete clarifications and revisions:
> >
> > - **Fairness of the sCD vs. CMT comparison (ImageNet 512, 1-step FID):** We clarify that the reported gap is not evidence of a limitation of CMT mid-training, but largely stems from *non-matched configurations* (different teacher models and different flow-map backbones). In addition, the lack of reliable, reproducible open-source sCD code makes a strictly controlled comparison difficult.
> > - **Better balance between empirics and theory:** We moved additional empirical results from the appendix to the main paper and tightened/polished the theoretical section to improve readability and emphasize the key empirical takeaways.
> > - **Pipeline clarity (losses + algorithm):** We clarified how the losses (Eq. (7) and Eq. (8) in the revised manuscript) are used in mid-training and in flow-map post-training, and added end-to-end pseudocode in *Algorithm 1* for the proposed full pipeline.
> > - **CMT vs. Slow CMT (cost + scaling):** We clarified the CMT loss and the Slow CMT variant. Empirically, Slow CMT requires 3× more GPU training time than CMT. We further provide a theoretical runtime comparison showing that Slow CMT scales as $\mathcal{O}(M)$ with the number of time discretization steps $M$, and is therefore inherently slower at larger $M$.
> > - **CMT vs. Rectified Flow (objective distinction):** We clarified the fundamental difference in objectives and goals: CMT defines a *flow-map (integrated) loss* to approximate trajectories, whereas Rectified Flow uses a *velocity-level (instantaneous) loss* for data-noise coupling refinement.
> >
> > We appreciate the reviewer’s time and would be happy to address any remaining questions, time permitting.

---

### Official Review · Reviewer_GbKZ · 2025-10-26

**Soundness:** 3
**Presentation:** 3
**Contribution:** 3
**Rating:** 8
**Confidence:** 4

**Summary:**

This paper proposed a mid-training strategy. The motivation is to alleviate instability in training for the few-step trajectory distillation method, such as the Consistency model (CM) and MeanFlow. Such unstable training is inherent in approximation errors for the trajectory from $t->s$. This paper proposed a CMT loss to first provide a sub-optimal trajectory from $t->s$ with lower approximation errors, which significantly simplifies post-training trajectory distillation.  The experimental results demonstrate that such a mid-training strategy can: 1) achieve the few-step generation built up on the trajectory distillation method, such as CM, 2) boostraps the post-training for CM and MeanFlow.

**Strengths:**

1. This paper solves an important challenge about how to train a trajectory-based distillation method in a stable way and at a low training cost.  These distillation methods rely heavily on theory, which is sensitive to approximation errors. The mid-training strategy is a way to remedy this situation.

2. The proposed method is interesting and novel. Such a mid-training strategy provides a good starting point for the trajectory-based distillation method. The experimental results also reveal that the training cost during mid-training remains low compared to that of the direct training of the trajectory-based distillation method.

3. The experimental results demonstrate that CMT with CM can achieve the one-step generation and reduce the training cost.

**Weaknesses:**

I think this paper does not have significant concerns. I only have minor concerns.

1.  While the author claims that $x_{T}$ is the terminal state. I refer to the code but find that $x_{T}$ actually comes from GT + noise. In this case, to directly claim that $x_{T}$ is the terminal state which differs from "a fresh data point $x_{0}$ and a noise vector $\epsilon$" (205 lines) may mislead the reader.

2. For the choice of the discretization time step. Is there any way to quantify the approximation errors that will be generated in different discretization schedulers?

3. I think this paper is related to the progress distillation [1]. Although the targets of the two methods differ, the author should slightly distinguish them using CMT.

[1] PROGRESSIVE DISTILLATION FOR FAST SAMPLING OF DIFFUSION MODELS. Tim Salimans et al. ICLR 2022.

**Questions:**

1. A minor question: Can this method be used for a large-scale model such as SD 3.5 and Flux?

To sum up, this paper proposed a novel mid-training strategy to mitigate the training cost for few-step trajectory-based distillation methods. I have carefully checked the theory proofs and related code. I think there are no major problems. The overall process is novel and sound. The experimental results clearly demonstrate that CMT can reduce the training cost and improve the stability of trajectory-based distillation methods. Therefore, I rate it as accept.

---

> ### Author Response · Authors · 2025-11-20
> **Reply to Reviewer GbKZ**
>
> We would like to thank Reviewer GbKZ for your thoughtful review and the insights you shared. The answers to your questions are summarized below.
>
> ## W1. Regarding the implementation of $\mathbf{x}_T$.
>
> ## R1.
> We thank the reviewer for carefully checking our implementation and for pointing out this potential source of confusion. In the code snapshot released for the review, we indeed construct \$\mathbf{x}_T\$ via a forward perturbation of a data $\mathbf{x}_0$, as the reviewer observed. In our latest codebase, we instead draw directly from the prior.
>
> Indeed, these two choices are well aligned. In EDM-style setups, one typically has $p_T \approx p_{\text{prior}}$ for the conventional terminal time $T$, and in flow-matching formulations one has $p_T = p_{\text{prior}}$ by design. As a result, sampling from the forward-perturbed or directly from prior is (up to a negligible approximation error) equivalent in distribution. Empirically, we did not observe any statistically significant difference in performance between these two implementations.
>
> We fully agree that our original wording may mislead readers into thinking that $\mathbf{x}_T$ is constructed in a way that is inconsistent with the implementation. We have therefore revised the manuscript to (i) explicitly state that this can be taken either from forward-perturbation or directly from prior, and (ii) clarify how this is implemented in practice. We will also release our latest code, which consistently uses prior and includes comments that match the revised description, in order to avoid any further confusion. To be more precise, in the updated code we modify line 128 of "training/loss.py" from
> ```python
> x = y + eps_T
> ```
> to
> ```python
> x = eps_T
> ```
>
> ## W2. Regarding the discretization timestep used by the solvers.
>
> ## R2.
> We thank the reviewer for raising this question about the choice of discretization time steps and how to quantify the associated approximation error. In all our experiments, we follow a standard DPM-Solver sampling setting: we reparameterize time by the log-SNR variable $\lambda_t := \log(\alpha_t / \sigma_t)$, where $\alpha_t$ and $\sigma_t$ are the forward noise schedules, and choose time steps that are uniformly spaced in $\lambda$ (the `skip_type="logSNR"` setting in DPM-Solver). We keep all settings as simple as possible as this scheduler is default in DPM-Solver.
>
>
> Regarding the possibility to quantify the errors, let {$\lambda_{t_i}$}$_{i=0}^M$ be the resulting time grid and denote
>
> $h := \max_{1 \le i \le M} |\lambda_{t_i} - \lambda_{t_{i-1}}|$.
>
> For a $k$-th order solver, the analysis of DPM-Solver (e.g., Theorem 3.2 in [1]) shows that the global discretization error scales as $\mathcal{O}(h^k)$. In other words, different discretization schedulers correspond to different step-size patterns in $\lambda$-space, and their approximation error is quantitatively controlled by this $\mathcal{O}(h^k)$ bound.
>
>
> In our work, we intentionally keep the setup simple and fair, and adopt this well-studied log-SNR schedule for sampling rather than introducing an additional axis of improvement on the solver side. Empirically, we observe that moderate variations of the schedule (under the same number of steps $M$ and solver order $k$) do not materially affect sample quality. A systematic optimization of the time grid or scheduler is an interesting and largely orthogonal direction; in the revised manuscript, we now discuss and cite works that specifically study this aspect (e.g., [2, 3]). However, we view such schedule optimization as outside the main scope of CMT, whose focus is on a principled mid-training scheme for flow map models. Motivated by the reviewer’s comment, we also clarify this choice in the revised manuscript.
>
>
> [1]  Lu, C., Zhou, Y., Bao, F., Chen, J., Li, C., & Zhu, J. (2022). Dpm-solver: A fast ode solver for diffusion probabilistic model sampling in around 10 steps. Advances in neural information processing systems, 35, 5775-5787.
>
> [2] Sabour, A., Fidler, S., & Kreis, K. (2024, July). Align Your Steps: Optimizing Sampling Schedules in Diffusion Models. In International Conference on Machine Learning (pp. 42947-42975). PMLR.
>
> [3] Nguyen, B., Nguyen, B., & Nguyen, V. A. Bellman Optimal Stepsize Straightening of Flow-Matching Models. In The Twelfth International Conference on Learning Representations 2024.

---

> ### Author Response · Authors · 2025-11-20
> **Reply to Reviewer GbKZ**
>
> ## W3. Regarding the relationship of CMT and Progressive Distillation.
>
> ## R3.
> **Differences in methodology and goal.** We first clarify the fundamental difference between Progressive Distillation (PD) and our CMT in how they learn a proxy of the special flow map $\Psi_{t \to 0}$, where $\Psi_{t\to s}$ denotes the oracle flow map from time $t$ to $s$. Conceptually, PD trains a student model
>
> $
> \mathbf{G}_\theta(\cdot, t, t-2\Delta t)
> $
>
> to approximate a *local (in time)*  two–step flow $\Psi_{t\to t-2\Delta t}$ by matching the composition of two one–step teacher solver updates:
>
> $$
> \mathbf{G}_\theta(\cdot, t, t-2\Delta t) \approx \Psi_{t\to t-2\Delta t} = \Psi_{t-\Delta t\to t-2\Delta t} \circ \Psi_{t\to t-\Delta t} \approx  \mathrm{Solver}_{t-\Delta t\to t-2\Delta t} \circ \mathrm{Solver}_{t\to t-\Delta t},$$
>
> and then iteratively repeats this procedure to distill a sequence of students that halve the number of steps at each stage (e.g., $1024 \to 512\to 256 \cdots \to 16\to 8\to 4\to 2\to 1$).
>
> In contrast, the mid-training stage of CMT learns a one-shot, *global* anytime-to-origin mapping, rather than relying on an iterative refinement process as in PD. Given a sample from prior distribution, we first obtain a discrete reference trajectory by running a numerical ODE solver with a pre–trained diffusion model as the teacher sampler. The goal of CMT is then to train a single student such that it can map any intermediate state back to the clean origin, so that the student model learns a direct approximation of the flow $\Psi_{t_i\to 0}$ for *all* intermediate times, rather than only a two–step local map.
> Moreover, by construction the learning objective of CMT is exactly aligned with the flow-map objective in Eq. (2): it supervises entire PF-ODE trajectories and thus yields a trajectory-consistent, numerically stable initialization of the flow-map model. In contrast, PD fundamentally cannot serve as a trajectory-aware initialization in the same sense.
>
> **Further differences in practice.**  We also emphasize several practical differences between PD and CMT. First, PD is computationally demanding: one must repeatedly run a relatively slow teacher sampler and train a sequence of students at multiple stages, which substantially increases the total training cost. This makes PD not suitable as a lightweight mid–training refinement method, whereas CMT converges quickly with a single student model and minimal additional heuristics. Second, each PD stage only approximates a composed teacher mapping over an increasingly long effective step (e.g., $2\Delta t$, $4\Delta t$, etc.). As the number of steps is aggressively compressed, approximation errors from each stage can accumulate, which in practice tends to degrade sample quality and diversity at very small NFE (e.g., $1$–$2$ steps). Third, the distilled student in PD is typically specialized to a particular teacher parameterization, noise schedule, and ODE sampler (in prior work, mainly a DDIM-like sampler), so changing these components generally requires  substantial adaptation. In contrast, CMT is formulated to be solver–agnostic: it directly learns an anytime–to–origin mapping and is compatible with a wide range of ODE solvers, with especially strong benefits when paired with higher–order or multi–step integrators.

---

> ### Author Response · Authors · 2025-11-20
> **Reply to Reviewer GbKZ**
>
> ## Q1. Regarding large-scale experiments.
>
> ## R1.
> CMT is a generic and principled pipeline: it only requires (i) a pre-trained teacher sampler (such as SD~3.5 or Flux) and (ii) a few-step flow-map backbone; CMT then provides a trajectory-aware, numerically stable initialization for learning that flow map. The mid-training stage itself is stable and minimally reliant on ad-hoc techniques or hyperparameters, since it reduces to supervised regression with fixed, high-quality labels, and it uses reliable, well-tested teacher samplers (e.g., diffusion ODE solvers). Our theoretical results (e.g., Theorem 5.1 and Theorem F.1) further formalize the advantages of CMT initialization over other plausible choices. In this sense, there is no conceptual obstacle to applying CMT to large-scale systems at the SD 3.5/Flux scale, once a well-developed few-step flow-map backbone is available in those systems.
>
>
> In the current work, we already evaluate CMT on large-scale academic benchmarks (ImageNet 256x256, ImageNet 512x512) and, in the revised manuscript, additionally on MSCOCO T2I (please refer to Table 5 in the revised manuscript or to the attached table). To be more precisely on the additional T2I experiments, we evaluate three MF configurations on MS-COCO: (1) **MF (Rand. Init.)**: MF trained only in the post-training stage from random initialization; (2) **MF (SiT init.)**: SiT pre-training followed by MF post-training initialized from SiT; (3) **CMT**: SiT pre-training, CMT mid-training using SiT as the teacher to generate ODE trajectories for an MF student, then MF post-training from this mid-trained initializer. The table shows that CMT performs strongly on T2I with substantially faster convergence than other initialization schemes: it achieves the best FID while reducing total training time by about 47%. CMT’s 2-step FID (5.01) is also close to the teacher SiT’s 50-step FID (4.73). This additional experiment show that CM can yield strong performance even in T2I, highlighting the generality and effectiveness of CMT across more challenging datasets and architectures.
>
> | Method          | Pre-Training | Mid-Training | Post-Training | Total Time (↓) | 1/2-Step FID (↓)   |
> |----------------|--------------|--------------|---------------|----------------|--------------------|
> | MF (Rand. Init.)   | 0            | 0            | 60 hours      | 60 hours       | 15.97 / 5.42       |
> | MF (SiT Init.) | 22 hours     | 0            | 38 hours      | 60 hours       | 15.55 / 5.26       |
> | **CMT**        | 22 hours     | 1 hour       | 9 hours       | **32 hours**   | **15.12 / 5.01**   |
>
> These settings match or exceed the scale used in prior CM-family and MeanFlow papers that focus on principled training schemes, and CMT consistently improves over these baselines.
>
>
> Adapting CMT to truly industrial-scale T2I models would additionally require access to high-quality proprietary datasets, non-public training code, and very large compute budgets, which is beyond what can be realistically achieved within the rebuttal period. We therefore view deployment on SD 3.5/Flux-level systems as an exciting direction for future work, rather than a prerequisite for establishing the practicality and scalability of CMT at the scales typically considered in the literature.

---

> > ### Comment · Reviewer_GbKZ · 2025-11-23
> >
> > Thanks to the author`s rebuttal. I have carefully checked all the contents. The author's further clarification addresses my concerns about the code, timestep setting, and the related work. I think the initial score 'accept' is proper, so I keep my initial score.

---

> > > ### Author Response · Authors · 2025-11-26
> > >
> > > We sincerely thank the reviewer for the thorough and constructive feedback.

---

### Official Review · Reviewer_3ZTM · 2025-10-30

**Soundness:** 3
**Presentation:** 4
**Contribution:** 3
**Rating:** 8
**Confidence:** 4

**Summary:**

The main contribution of this paper is the introduction of an intermediate training stage for accelerating the training of fast inference generative models for image generation in continuous signals.

Current state-of-the-art methods for distilling flow and diffusion models without GAN losses such as ECD [1], and MF [2] typically employ a two-stage training pipeline. In Stage 1 (pre-training), a diffusion model is trained with an x-prediction parameterization [3] or a flow matching model with a velocity-prediction parameterization [4]. In stage 2 (post-training),  the trained diffusion/flow model is distilled by approximating the flow map it defines. However, optimizing the exact objective function for the flow map is intractable, so instead an auxiliary loss is used, commonly involving a stopgrad instance of the current student model which can lead to unstable training.

This paper proposes a third stage, mid-training, for distillation of flow and diffusion models. The proposed mid-training loss regresses the flow map defined by the pretrained model using pre-sampled trajectories. Finally, the effectiveness of the mid-training stage is validated on several benchmark.

[1] Geng, Zhengyang, et al. "Consistency models made easy." arXiv preprint arXiv:2406.14548 (2024).

[2] Geng, Zhengyang, et al. "Consistency models made easy." arXiv preprint arXiv:2406.14548 (2024).

[3] Karras, Tero, et al. "Elucidating the design space of diffusion-based generative models." Advances in neural information processing systems 35 (2022): 26565-26577.

[4] Lipman, Yaron, et al. "Flow matching for generative modeling." arXiv preprint arXiv:2210.02747 (2022).

**Strengths:**

1. The paper is well written and the method presented clearly.

2. The paper goal - accelerating distillation of flow/diffusion models,  is aligned with current most relevant problems considered in deep learning, potentially making large impact.

3. Introducing a new training stage is a popular method for improving training of neural networks and can easily integrate with existing training pipelines.

4. The empirical validation is comprehensive.

**Weaknesses:**

1. Regressing a peace wise linear version of the pre-trained model's trajectories was done in previous work e.g., [5], and might be worth mentioning.

2. While the preliminary section covers number of previous work, the main paper does not include a previous work section (instead it is in the appendix) which is somewhat unorthodox for a research paper..

[5] Nguyen, Bao, Binh Nguyen, and Viet Anh Nguyen. "Bellman optimal stepsize straightening of flow-matching models." arXiv preprint arXiv:2312.16414 (2023).

**Questions:**

1. In line 214 it is written that the discretization number of steps $T$ is fixed for the mid-training. First, did you examine how different choices of $T$ affect the training and the final results? Second, in the post-training stage ( i.e., ECD [1]/MeanFlow [2]) a continuous time is used again?

2. How many different trajectories $\\{x_{t_0}, x_{t_1}, ... ,x_{t_T}\\}$ are used during mid-training and does the time to generate these is taken into account in your mid-training times report?

3. It is claimed in the paper that training of vanilla ECD [1]/MeanFlow [2] remains "unstable, and configuration sensitive", how ever it seems that the authors did train a MeanFlow model with relatively close performance to the CMT (the proposed method). Can the authors provide an estimation of how many training attempts hyper-parameters searches are required to obtain the reported results ECD [1]/MeanFlow [2]?

---

> ### Author Response · Authors · 2025-11-20
> **Reply to Reviewer 3ZTM**
>
> We would like to thank Reviewer 3ZTM for your thoughtful review and the insights you shared. The answers to your questions are summarized below.
>
> ## W1. Regarding the previous work [5].
>
> ## R1.
> We thank the reviewer for pointing out this interesting work. We have added the citation and a brief discussion in the related-work section, and are happy to expand it in the appendix if the reviewer finds it helpful.
>
> ## W2. Regarding the Inclusion of Previous Work in the Main Paper.
>
> ## R2.
> We thank the reviewer for this helpful comment. In the initial submission, we moved the detailed related-work discussion to the appendix due to the strict page limit, and only summarized key prior work in the preliminaries. In the revision, we have reorganized the manuscript and brought the Related Work section into the main paper, following the conventional structure for research articles while keeping it concise enough to respect the page limit.
>
>
> [5] Nguyen, Bao, Binh Nguyen, and Viet Anh Nguyen. "Bellman optimal stepsize straightening of flow-matching models." arXiv preprint arXiv:2312.16414 (2023).
>
> ## Q1. Regarding the treatment of time in mid-training and post-training.
>
> ## R1.
> Regarding the number of discretization steps in mid-training, our notation may have been confusing: we use $T$ to denote the terminal time of the diffusion process (as in conventional settings, e.g., $T=80$ in EDM and $T=1$ in flow matching), while $M$ denotes the number of mid-training solver steps. In all main experiments we fix $M=16$ for the teacher ODE solver to control computational cost. To further address the reviewer’s concern, we additionally provide an ablation on the choice of $M$, as detailed below.
>
>
>
> **Ablation on $M$ in mid-training and their effect on post-training.** To assess how the number of discretization steps $M$ in CMT mid-training affects the quality of the post-trained flow map, we conduct an ablation on CIFAR-10 while keeping all other settings fixed. We evaluate $M \in$ {8, 16, 32}, where $M=16$ is the default choice in our main results.
>
>
> Using $M=16$ yields the best 1/2-step FIDs (2.74 / 1.97), while $M=8$ (2.84 / 2.02) and $M=32$ (2.85 / 2.03) are slightly worse but still clearly outperform vanilla ECT (3.60 / 2.11). This confirms that our default setting $M=16$ is near-optimal, and also shows that the post-trained flow map is reasonably robust to the choice of $M$: smaller $M$ converges slightly faster at the cost of a modest FID degradation, whereas larger $M$ does not bring further gains.
>
> These trends are consistent with the behavior of the teacher solver.
> If $M$ is too small, the teacher trajectories deteriorate because the solver uses too few steps; this degrades mid-training quality since the supervision signal becomes less reliable.
> Conversely, if $M$ is too large, training slows down because the multistep DPM-Solver cannot be trivially parallelized and becomes the computational bottleneck. Under a fixed training-data budget (measured in kimgs), increasing $M$ also reduces the number of distinct trajectories seen during training, since kimgs = traj$\times M$. This can slightly harm mid-training quality, while the benefit from further improving teacher samples is marginal, because DPM-Solver already produces high-quality samples with about $10$–$20$ steps.
>
> The original DPM-Solver work likewise recommends on the order of $10$–$20$ steps for strong sample quality at modest cost. We therefore adopt $M=16$ as a reasonable default that keeps CMT both succinct and efficient, and our ablations support that this choice is a robust operating point.
>
> **Time parametrization in post-training stage.** Regarding the post-training stage of the flow map model, we follow the original time parameterization of each method in the post-training stage.
> - For CM-based models (in particular ECT/ECD), the objective in Eq. (3) is defined in continuous time, and we specify a small step size $\Delta t$ for Eq. (3); in practice we use fixed $\Delta t \in$ {$1/256$, $1/512$, $1/1024$}.
> - For MeanFlow, we adopt the continuous-time formulation from the original work and sample $t$ from the prescribed log-normal distribution on $[0,T]$.
>
> During sampling, both ECT and MeanFlow are treated as continuous-time flow map models on $[0,T]$. Thus, the post-training stage consistently uses the (continuous-time) setup of the underlying methods.

---

> ### Author Response · Authors · 2025-11-20
> **Reply to Reviewer 3ZTM**
>
> ## Q2. Regarding trajectory counts and overall runtime.
>
> ## R2.
> **Counts of trajectories.** The number of distinct trajectories used during mid-training can be computed from the hyperparameters reported in Appendix B. Specifically, let the CMT mid-training batch size be $B$, the number of discretization steps be $M$, and the total number of training iterations be $\texttt{iter}$. At each iteration, we observe $B/M$ trajectories, and each trajectory provides $M$ steps, so the total CMT batch size is $B/M \times M = B$. Therefore, the total number of distinct trajectories is $B/M\times \texttt{iter}$. For instance, in CIFAR10 \& AFHQ \& FFHQ, $B=128$, $M=16$, and $\texttt{iter}=$ 300K (300,000). So, the total number is 2.4Million.
>
> **Fair runtime comparison.**  Yes, the time required to generate teacher trajectories is fully included in our reported training-time measurements for a fair comparison. For example, on ImageNet 512x512, CMT achieves state-of-the-art FIDs while using up to 98% less training data and GPU time compared to CM baselines. On ImageNet 256x256, CMT further reduces the total training time by about 50% compared to MF.
>
> ## Q3. Regarding hyperparameter searches for vanilla ECD and MeanFlow.
>
> ## R3.
> We train a MeanFlow model that achieves performance relatively close to CMT, while CMT reduces the training time by 50%. The hyperparameter robustness of CMT is evident in its stability with respect to sensitive parameters such as ECD’s $\Delta t$ and MeanFlow’s $t \neq s$ ratio.
>
> **About the vanilla ECT/ECD.** Vanilla ECT is highly sensitive to the choice of the step size $\Delta t$. In practice, stable training requires annealing $\Delta t$ from a relatively large value (e.g., $1/2$) to a much smaller value (e.g., $1/256$) over training iteration. A larger $\Delta t$ yields higher discretization bias but improves numerical stability, whereas a smaller $\Delta t$ reduces this bias but leads to high-variance updates and severe instability. Consequently, vanilla ECD requires a carefully engineered $\Delta t$ schedule: in our experiments, using a small $\Delta t$ from the beginning consistently causes divergence.
>
> By contrast, when we initialize with CMT and then apply ECD (CMT+ECD), we can train with a very small, fixed step size (e.g., $\Delta t = 1/1024$) from the very beginning, without any instability. This indicates that CMT provides a trajectory-aware, numerically well-conditioned initialization that substantially enlarges the stable hyperparameter region. Practically, this both simplifies hyperparameter tuning (no annealing schedule is needed) and accelerates convergence, since we can operate in a low-bias regime throughout training.
>
> This effect is particularly pronounced on ImageNet 512x512. Under comparable compute, CMT+ECD with a fixed tiny step size significantly outperforms vanilla ECD, even when the latter uses a carefully tuned annealing schedule for $\Delta t$. Moreover, vanilla ECD with a small $\Delta t$ applied from the beginning diverges, whereas CMT+ECD remains stable and converges to a substantially better solution.
>
>
> **About Mean Flow.** Empirically, we observe a similar stability phenomenon for MF. In the standard MF implementation, the loss is explicitly designed so that each training update decomposes, with a pre-assigned ratio $\rho$, into (i) a standard flow-matching term for $t = s$, which is numerically stable, and (ii) a pure MF term for $t \neq s$, which is substantially harder to optimize and prone to instability. Increasing the $t \neq s$ ratio $\rho$ has a clear algorithmic benefit for few-step MF models: it allocates a larger portion of the loss to learning the long-range mappings from $t$ to $s$ that are required for aggressive step reduction, rather than to the local flow matching term at $t = s$.
>
>
> However, vanilla MF is highly sensitive to the proportion $\rho$ of $t \neq s$ samples in the training mixture: in our experiments, stable training requires $\rho \lesssim 0.25$, and increasing $\rho$ beyond this range consistently leads to divergence.
>
>
> By contrast, when we initialize with CMT and then train MF (CMT+MF), we can safely increase the fraction of $t \neq s$ samples up to about $\rho \approx 0.75$ without observing instability.
> This indicates that CMT again enlarges the stable hyperparameter region: vanilla MF only tolerates relatively small $t \neq s$ ratios, whereas CMT+MF remains stable even when the majority of samples satisfy $t \neq s$. In practice, this leads to faster and more effective convergence for CMT+MF compared to vanilla MF under similar compute and model capacity (see Table 4).

---

> > ### Author Response · Authors · 2025-11-26
> > **A Kind Reminder for Comments**
> >
> > We sincerely thank Reviewer 3ZTM for the thorough and constructive feedback. We believe our rebuttal addresses the main concerns through the following concrete revisions and additional evidence:
> >
> > - **Related work:** Reorganized and expanded the related-work discussion, including additional coverage of [5].
> > - **Time discretization and its effect on post-training:** Added new ablations on the discrete time grid size $M$ in mid-training. Results show that post-training (flow-map) initialized from CMT mid-trained weights is *robust to $M$*, and the default **$M=16$** provides a strong performance–runtime trade-off.
> > - **Trajectory counts + fair runtime accounting:** Clarified trajectory counts and our fair comparison of overall training cost. Consistent with the original submission, CMT reduces overall training runtime (including all pre-/mid-/post-training stages) by up to 98% vs. CMs and 50% vs. MF-from-scratch, while achieving state-of-the-art FIDs.
> > - **Hyperparameter robustness:** Clarified the hyperparameter search for vanilla ECD and MeanFlow, and added evidence that *CMT remains robust* under these variations.
> >
> > We appreciate the reviewer’s time and would be happy to address any remaining questions, time permitting.
> >
> > [5] Nguyen, Bao, Binh Nguyen, and Viet Anh Nguyen. "Bellman optimal stepsize straightening of flow-matching models." arXiv preprint arXiv:2312.16414 (2023).

---

> > > ### Comment · Reviewer_3ZTM · 2025-11-27
> > >
> > > I would like to thank authors for there detailed and clear answers. In particular, the information provided in answer to question 3 highlights the advantage of the proposed method.

---

### Author Response · Authors · 2025-11-20
**Reply to All Reviewers**

# List of major revisions
We sincerely thank the reviewers for their helpful and constructive feedback. We have revised and clarified the manuscript in response to these comments. The main changes are summarized below, and all modified parts are highlighted in "blue" in the revised version.


1. We relocate and expand the related work section into the main paper as Section 2.3, providing a more complete and better discussion of prior approaches on solvers.

2. We clarify the proposed CMT pipeline and summarize it as full pseudocode in Algorithm 1 in Appendix B.1, making the three-stage procedure (pre-training, CMT mid-training, and flow-map post-training) explicit, including the two independent scenarios of special flow map (CM) and general flow map (MF) learning.

3. We add new T2I experiments on the MS-COCO dataset and show that CMT significantly reduces total training time by about 47% compared to other initialization schemes (random initialization or a pretrained diffusion model), while achieving the best FID.

4. We reorganize the ablation on alternative initialization schemes and move it to Section 4.3 for better coherence with the main empirical results. We also extend the discussion of the theoretical runtime counts of these initialization variants in Appendix C.2.

5. We shorten and streamline the theoretical analysis in Section 5 to strike a better balance between empirical insights and theoretical takeaways, while retaining the key results and intuitions.


In the replies below, we also apologize if some mathematical formulas are not displayed properly due to Markdown constraints. We would be very happy to clarify any expressions that are hard to read or unclear.

---

### Author Response · Authors · 2025-12-01
**Rebuttal summary: reviewer-by-reviewer updates & actions**

### Reviewer 3ZTM
- **Related work:** Reorganized/expanded related work.
- **Time discretization in mid-training:** Added ablations on discrete grid size $M$. Post-training (flow-map) initialized from CMT mid-trained weights is *robust to $M$*; default *$M=16$* is a strong speed–quality trade-off.
- **Trajectories + fair runtime:** Clarified trajectory counts and *end-to-end training cost* accounting. As we claimed in the original submission, CMT reduces *overall runtime* (pre-/mid-/post-training) by up to *98% vs. CMs* and *50% vs. MF-from-scratch*, while achieving SOTA FIDs.
- **Hyperparameter robustness:** Clarified hyperparameter search for vanilla ECD and MeanFlow; added evidence CMT remains robust under these variations.

Reviewer appears satisfied with our replies.

---

### Reviewer GbKZ
We clarified / added:
- **Prior sampling:** $\mathbf{x}_T$ is sampled from the prior distribution.
- **Solver discretization:** Clarified that our solver timesteps follow the conventional DPM-Solver timestep selection.
- **Related work:** Clarified relationship between CMT and Progressive Distillation.
- **Extra T2I experiments:** Added MS-COCO T2I experiments.

Reviewer appears satisfied with our replies.

---

### Reviewer 12GL
- **Fairness (sCD vs. CMT, ImageNet 512, 1-step FID):** Clarified that the observed gap mainly reflects configuration differences in the available comparison (different teacher samplers and flow-map backbones), not a limitation of CMT. Also noted that without reliable, reproducible open-source sCD code, a fully controlled apples-to-apples comparison is not feasible. However, CMT can already bootstrap the relatively weak ECD flow-map backbone into a SOTA 2-step generator.
- **Empirics vs. theory balance:** Moved key empirical results from appendix to main; tightened/polished theory for readability and to emphasize takeaways.
- **Pipeline clarity:** Clarified how Eq. (7)/(8) are used in mid-training vs. post-training; added *end-to-end pseudocode (Algorithm 1)*.
- **CMT vs. Slow CMT:** Clarified loss variants; empirically Slow CMT costs ~3× GPU time. Added theory showing Slow CMT scales as $\mathcal{O}(M)$ in the number of time steps $M$, so it becomes inherently slower as $M$ increases. In contrast, the CMT used in the main paper/results scales as $\mathcal{O}(1)$ and remains efficient.
- **CMT vs. Rectified Flow:** Clarified objective distinction: CMT is *flow-map / integrated trajectory* supervision for trajectory-aware initializations; Rectified Flow is *velocity-level / instantaneous* supervision for data–noise coupling refinement.


---

---

> ### Author Response · Authors · 2025-12-01
> **(Continued) Rebuttal summary to AC: reviewer-by-reviewer updates & actions**
>
> ### Reviewer BkLe
> - **Scale concerns + added T2I evidence:**
>     - Clarified ImageNet-scale experiments are comparable to prior CM/MF works under standard academic settings.
>     - Added *MS-COCO T2I* results: CMT achieves best FID, reduces *total training time by 47%*, and approaches SiT teacher 50-step (100 NFEs due to CFG in SiT) quality with only 2 steps/NFEs.
> - **Why even larger-scale T2I/T2V is challenging during rebuttal:** Although CMT remains lightweight, efficient, and effective, and CMT  applies to general T2I tasks such as MS-COCO as previously demonstrated, the main challenge lies in the substantially larger training corpus and richer conditioning. Specifically, the 32 million BLIP3o T2I samples (compared with only 1.4 million ImageNet samples), together with their more complex text conditions (as opposed to simple class labels in ImageNet), require significantly longer training to fully absorb the information. In addition, achieving strong performance under these conditions necessitates a larger DiT model. Below, we provide the detailed settings for the T2I experiments.
>
> - **Additional experiments and compute constraints.** We followed the REPA-E setup (3B DiT, 32M BLIP3o samples at 512×512) to scale CMT to T2I. However, the required teacher FM pre-training is clearly infeasible within the rebuttal window:
>
>   - T2I flow-matching pretraining with REPA-E + DiT on BLIP3o 512×512 needs ≈460–470 H100 GPU days (≈1,400 hours on 8×H100) to obtain a state-of-the-art teacher.
>   - On ImageNet 256×256, REPA-E + DiT flow matching converges in ≈5 H100 GPU days, and our MF test case already costs ≈30 H100 GPU days.
>   - The analogous BLIP3o T2I setup is ≈100× more expensive, i.e., ≈3,000 H100 GPU days, which is prohibitive under our current compute budget and time limit.
>
>   Given this, we cannot complete full 512×512 BLIP3o T2I within rebuttal, but we plan to downsample BLIP3o to 256×256 and aim to report a 3B DiT + 32M-sample T2I result by the camera-ready deadline.
>
>     - **Orthogonal bottleneck that motivates CMT.** This bottleneck is entirely in dataset/model scale, not in CMT itself. In fact, we also observe that training flow-map models (e.g., MeanFlow) from scratch, without a strong FM teacher, can diverge due to unstable early gradients, reinforcing the need for principled mid-training/initialization once a viable teacher exists.
>
>
>
> - **Contributions clarified:**
>   - First mid-training method for flow-map models in vision generation that substantially reduces training budget/time while achieving SOTA results.
>   - First systematic theory on how initialization affects flow-map training.
>   - A simple but general recipe applicable to CM-family (t to 0 jumps) and MF (t to s jumps).
> - **Fairness of CMT loss:** Gains are *not* mainly from an “unfair” loss. ELatentLPIPS is used *only* in ImageNet 512 CMT mid-training; removing it (plain $L_2$ mid-training) yields similar FIDs with modest extra budget (28.8M → 32.8M images), still far more efficient than typical CM baselines. In contrast, CM baseline (ECD) is sensitive to loss choice and becomes less stable and performs worse if ELatentLPIPS is used.

---

### Meta-Review · Area_Chair_NLNa · 2026-01-06

**Summary:**

### Reviewer 3ZTM
1. No major concerns.

### Reviewer GbKZ
1. No major conerns.

### Reviewer 12GL
1. 1-step result is worse than sCD
    * **Author replies**: the method uses a different backbone from sCD and the teacher model is also different. So it's not a strict
    apple-to-apple comparison.


### Reviewer BkLe
1. Lack of large-scale experiments, such as text-to-image and text-to-video
    * **Author replies**: The authors add new results on MS-COCO Text-to-Image, the results
    show that the proposed method improves convergence and generation.
2. Contribution is incremental.
    * **Author replies**: The authors reiterate the contributions.
    * **AC comment**: The reviewer does not give reasons why he thinks the contribution is incremental.
    So I think the concern is not well-grounded.
3. Performance gain might be because of no LatentLPIPS in the baseline
    * **Author replies**: The authors provide experiment results to show that CMT's improvement
    does not rely on LatentLPIPS.

**Reviewer Concerns:**

I think the concerns from the reviewers are well addressed.

**Reviewer Scores:**

Reviewer 3ZTM: maintain the score, 8

Reviewer GbKZ: maintain the score, 8

Reviewer 12GL: maintain the score, 8

Reviewer BkLe: increase the score from 4 to 6

---

### Decision · Program_Chairs · 2026-01-26

Accept (Poster)